# Towards Atoms of Large Language Models

## Abstract

The fundamental units of internal representations in large language models (LLMs) remain undefined, limiting further understanding of their mechanisms. Neurons or features are often regarded as such units, yet neurons suffer from polysemy, while features face concerns of unreliable reconstruction and instability. To address this issue, we propose the *Atoms Theory*, which defines such units as atoms. We introduce the atomic inner product (AIP) to correct representation shifting, formally define atoms, and prove the conditions that atoms satisfy the Restricted Isometry Property (RIP), ensuring stable sparse representations over atom set and linking to compressed sensing. Under stronger conditions, we further establish the uniqueness and exact $\ell_1$ recoverability of the sparse representations, and provide guarantees that single-layer sparse autoencoders (SAEs) with threshold activations can reliably identify the atoms. To validate the Atoms Theory, we train threshold-activated SAEs on Gemma2-2B, Gemma2-9B, and Llama3.1-8B, achieving 99.9% sparse reconstruction across layers on average, and more than 99.8% of atoms satisfy the uniqueness condition, compared to 0.5% for neurons and 68.2% for features, showing that atoms more faithfully capture intrinsic representations of LLMs. Scaling experiments further reveal the link between SAEs size and recovery capacity. Overall, this work systematically introduces and validates Atoms Theory of LLMs, providing both a theoretical framework for understanding internal representations and a foundation for mechanistic interpretability.

## 1 Introduction

> By continually dividing matter into smaller parts, one cannot proceed indefinitely
> and must eventually reach indivisible units, termed atoms, meaning "indivisible."
> – Democritus

Large language models (LLMs), trained on vast corpus, exhibit emergent knowledge and reasoning abilities (Petroni et al., 2019; Brown et al., 2020; Achiam et al., 2023). Yet such information is not stored in explicit symbolic structures, but rather implicitly embedded within high-dimensional representations (Nanda et al., 2023; Gurnee et al., 2023; Cunningham et al., 2023). There arises a critical question, reminiscent of Democritus: **Do LLMs contain fundamental representational units—an atomic structure underlying how they encode and compose information?**

Traditionally, neurons have been regarded as fundamental units of neural networks (Olah et al., 2017). However, neurons often exhibit substantial polysemy (Elhage et al., 2022), growing doubt on their validity for analysis. To address polysemy, features decomposed from internal representations (Cunningham et al., 2023) have been proposed as such units (Olah et al., 2020). Yet this perspective remains controversial: (i) features fail to fully reconstruct the original representations (Bricken et al., 2023), raising fidelity concerns; and (ii) features undergo splitting into finer ones or merging into broader ones under different decomposition settings (Bussmann et al., 2025; Chanin et al., 2025), undermining stability. To date, there is no formal definition of fundamental units of LLMs, limiting theoretical clarity and constraining progress in mechanistic interpretability (Elhage et al., 2021).

In this paper, we present *Atoms Theory*, a rigorous framework for defining and analyzing atoms as fundamental units of LLMs, with strong properties of uniqueness, recoverability, and identifiability. Specifically, we introduce the atomic inner product (AIP) for representational distinguishability, a non-Euclidean metric to correct representation shifting (Figure 2- 3), where the centroid of the angle distribution between representations deviates significantly from 90° due to the Softmax operation in LLMs, thereby distorting the underlying geometry. We validate AIP across all layers of multiple

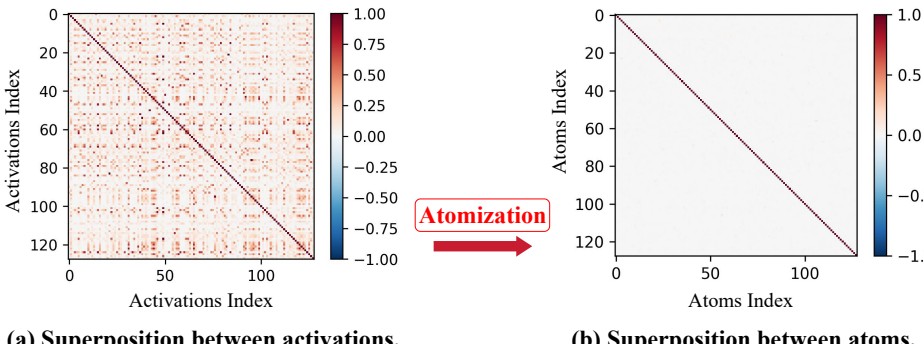

**(a) Superposition between activations.**      **(b) Superposition between atoms.**

Figure 1: Atomization of activations and corresponding superposition. (a) Activations of LLMs exhibit substantial superposition, sharing heavily overlapping structures. (b) After atomization, fundamental units (i.e., atoms) can be extracted, with the extent of superposition significantly reduced.

LLM families. Based on AIP, we formally define atoms and prove that, under specific conditions, atoms satisfy the Restricted Isometry Property (RIP), a sufficient condition for compressed sensing (Donoho, 2006; Candès et al., 2006), thereby ensuring stable embeddings of sparse combinations of atoms. We further show that stronger conditions guarantee uniqueness and exact $\ell_1$ recoverability of sparse representations over atom set, providing theoretical support for stability. In addition, we prove that single-layer sparse autoencoders (SAEs) (Templeton et al., 2024; Cunningham et al., 2023) with threshold activations can effectively identify the target atom set. Collectively, these results establish a comprehensive theory, encompassing modeling, recovery mechanisms, and provable guarantees.

To validate Atoms Theory, we conduct systematic experiments on Gemma2-2B, Gemma2-9B (Team et al., 2024), and Llama3.1-8B (Dubey et al., 2024) through atomization of activations (Figure 1). Single-layer SAEs with threshold activations decompose activations from LLMs with 99.9% sparse reconstruction across layers on average, confirming their recoverability. Building on this, we further examine the uniqueness conditions of sparse representations: on average, more than 99.8% of atoms satisfy these conditions, compared with only 0.5% of neurons and 68.2% of features (Lieberum et al., 2024; He et al., 2024). Finally, scaling experiments across dataset sizes and SAE capacities demonstrate that the reliable recovery occurs only when SAE capacity surpasses a critical threshold.

In summary, the contributions of this paper are as follows:

- **Atoms Theory framework.** We propose a rigorous theoretical framework that introduces atoms as the fundamental units of LLMs. This framework provides the formal definition of atoms, establishes their fundamental properties, and is grounded on atomic inner product, which we further validate across all layers of diverse LLM families.

- **Theoretical guarantees of uniqueness and recoverability.** We prove the conditions under which atoms satisfy the Restricted Isometry Property, and further demonstrate that stronger conditions ensure uniqueness and exact $\ell_1$ recoverability of corresponding sparse representations, providing rigorous guarantees for stability of atoms.

- **Practical method for atom identification.** We prove that single-layer SAEs with threshold activations can effectively recover the target atom set and, in practice, achieve 99.9% sparse reconstruction on average, demonstrating the feasibility of Atoms Theory.

- **Systematic validation and comparative analysis.** We conduct large-scale experiments on Gemma2-2B, Gemma2-9B, and Llama3.1-8B, showing that over 99.8% atoms satisfy uniqueness and recoverability conditions, compared to 0.5% for neurons and 68.2% for features. Scaling studies further reveal how SAE capacity governs recovery performance.

## 2 PRELIMINARY

Given an autoregressive language model parameterized by $\theta$, denoted as $f_\theta : X \mapsto Y$, the input $X$ is a sequence $\boldsymbol{x} = [x_1, x_2, \cdots, x_T]$ composed of tokens from a vocabulary $V$, where $x_i \in V$. The model maps this sequence to a probability distribution $\boldsymbol{y} \in \mathbb{R}^{|V|}$, thereby predicting the next token.

Specifically, an $L$-layer language model first assigns each token $x_i \in V$ an embedding representation $\boldsymbol{h}_i^0 \in \mathbb{R}^H$, which is subsequently updated across layers. At the $l$-th Transformer layer, the hidden state $\boldsymbol{h}_i^{l-1}$ is transformed into $\boldsymbol{h}_i^l$ according to

$$\boldsymbol{h}_i^l = \boldsymbol{h}_i^{l-1} + \boldsymbol{a}_i^l + \boldsymbol{v}_i^l, \tag{2.1}$$

where $\boldsymbol{a}_i^l$ and $\boldsymbol{v}_i^l$ denote the outputs of the attention and MLP modules at layer $l$, respectively. Viewing the computation from the perspective of the residual stream, the model can be expressed as

$$\boldsymbol{h}_i^L = \boldsymbol{h}_i^0 + \sum_{l=1}^L \boldsymbol{a}_i^l + \sum_{l=1}^L \boldsymbol{v}_i^l. \tag{2.2}$$

Finally, the language model maps the last hidden representation $\boldsymbol{h}_T^L$ to a probability distribution $\boldsymbol{y}$ via a Softmax operation to predict the next token as

$$\boldsymbol{y} = \mathrm{Softmax}(W_U^\top \boldsymbol{h}_T^L), \tag{2.3}$$

where $W_U \in \mathbb{R}^{H \times |V|}$ is the unembedding matrix.

## 3 ATOMS THEORY

In language models, all information is embedded into high-dimensional representations. Our objective is to identify the fundamental units of these representations, which we refer to as the "atoms". Formally, we aim to decompose a collection of representations $M = \{\boldsymbol{m}_i\}_{i=1}^{|M|}$, where $\boldsymbol{m}_i \in \mathbb{R}^H$. Each representations can be expressed as $\boldsymbol{m}_i = \sum_j \delta(i,j)\boldsymbol{d}_j$, where $\delta(i,j) \geq 0$ denotes the presence and magnitude of the $j$-th atom in the $i$-th representations. The matrix $D = [\boldsymbol{d}_1, \boldsymbol{d}_2, \cdots, \boldsymbol{d}_{|D|}] \in \mathbb{R}^{H \times |D|}$ consists of columns that form the atom set, and $\{\boldsymbol{d}_1, \boldsymbol{d}_2, \cdots, \boldsymbol{d}_{|D|}\}$ spans the atom space $\mathcal{D}$.

The key question is how to define the atom. A natural criterion is distinguishability, which ensures that each atom can be detected or manipulated without interfering others. In high-dimensional spaces, this translates to orthogonality: distinct atoms occupy mutually orthogonal directions, so their identities can be determined by inner products. The Euclidean inner product is the standard choice, but that need not be the case in language models. Following Park et al. (2023) and Hu et al. (2025), we illustrate how this leads to representation shifting. To see this, consider the following reparameterization of $W_U$ and $\boldsymbol{h}^L$:

$$W_U' \leftarrow A^{-\top} W_U + \boldsymbol{b} \cdot \boldsymbol{1}^\top, \quad \boldsymbol{h}'^L \leftarrow A \boldsymbol{h}^L, \tag{3.1}$$

where $A \in \mathbb{R}^{H \times H}$ is an invertible linear transform, $\boldsymbol{b} \in \mathbb{R}^H$, and $\boldsymbol{1} \in \mathbb{R}^{|V|}$ is the all-ones vector. Owing to the Softmax translation invariance, this reparameterization leaves the distribution unchanged: $\boldsymbol{y} = \mathrm{Softmax}(W_U^\top \boldsymbol{h}^L) = \mathrm{Softmax}(W_U'^\top \boldsymbol{h}'^L)$. See Appendix A.1 for further details.

However, since the training objective depends on representations solely through Softmax probabilities, $\boldsymbol{h}^L$ is identifiable only up to an invertible linear transform $A$. By the linearity of matrix multiplication and the residual-stream architecture, this invariance propagates to all hidden states and any decomposed atoms $\boldsymbol{d}$, which are likewise determined only up to transform $A$.

Under the Euclidean inner product, $\langle \boldsymbol{d}_i, \boldsymbol{d}_j \rangle \neq \langle A\boldsymbol{d}_i, A\boldsymbol{d}_j \rangle$, where $\langle \cdot, \cdot \rangle$ denotes the standard inner product. Hence algebraic operations based on this inner product need not reflect the true geometry of language-model representations. In activations of language models, the Euclidean inner product causes ***representation shifting***, with the angle-distribution centroid deviating markedly from $90°$ (Figure 2), a pattern observed across all layers of language model families such as GPT, Pythia, Llama, and Gemma. By contrast, atomic inner product, which introduced later, corrects representation shifting, keeping the centroid near $90°$ (Figure 3). Full details and results appear in Appendix B.

### 3.1 ATOMIC INNER PRODUCT

To better understand and define the representation of atoms in high-dimensional spaces, we require additional principles to determine the appropriate form of the inner product. To this end, we first introduce the definition of an inner product with the desired property.

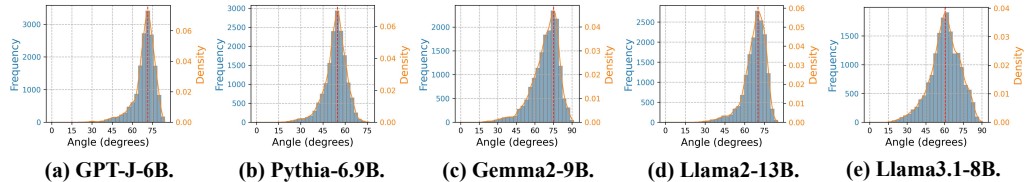

(a) GPT-J-6B.     (b) Pythia-6.9B.     (c) Gemma2-9B.     (d) Llama2-13B.     (e) Llama3.1-8B.

Figure 2: Representation shifting caused by adopting the Euclidean inner product, where the centroid of angles distribution between representations deviates substantially from $90°$.

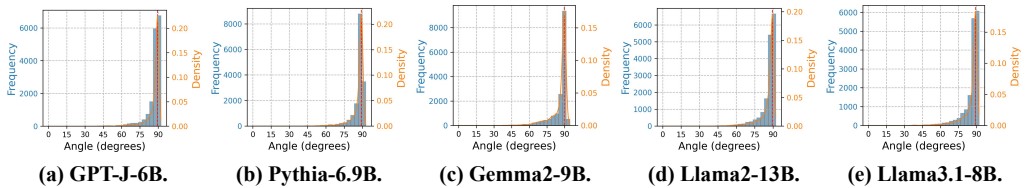

(a) GPT-J-6B.     (b) Pythia-6.9B.     (c) Gemma2-9B.     (d) Llama2-13B.     (e) Llama3.1-8B.

Figure 3: Correcting representation shifting by identifying and adopting the atomic inner product, where the centroid of angle distribution between representations approaches $90°$.

**Definition 1** (Atomic Inner Product; AIP). *In the atom space $\mathcal{D}$, the **atomic inner product** $\langle \cdot, \cdot \rangle_S$ satisfies $\langle \boldsymbol{d}_i, \boldsymbol{d}_j \rangle_S = 0$ for any pair of distinct atoms $\boldsymbol{d}_i, \boldsymbol{d}_j$ with $i \neq j$.*

Atoms are indexed, and any permutation of indices leaves their geometric properties unchanged. Consequently, there is no basis for assigning different scales to different atoms. By this symmetry, we assume a common norm under the chosen inner product, i.e., $\|\boldsymbol{d}_i\|_S = c, \forall i \in [|D|]$ and $c > 0$. It is important to note that the constant $c$ naturally cancels in the subsequent analytical framework.

How can we identify an inner product in the representation space that meets this definition?

**Theorem 2** (Explicit Form of the Atomic Inner Product). *Let $\langle \boldsymbol{d}_i, \boldsymbol{d}_j \rangle_S = \boldsymbol{d}_i^\top S \boldsymbol{d}_j$ be an atomic inner product with $S \in \mathbb{R}^{H \times H}$ symmetric and positive definite. If the columns of $D = [\boldsymbol{d}_1, \boldsymbol{d}_2, \cdots, \boldsymbol{d}_{|D|}]$ form a set of atoms such that $\|\boldsymbol{d}_i\|_S = c > 0$ for all $i$, and $\mathcal{D} \simeq \mathbb{R}^H$, then $S = c^2 (DD^\top)^{-1}$.*

All proofs are provided in Appendix A. To eliminate the effect of $c$ in comparisons and measurements, analogous to cosine similarity, we introduce the normalized atomic inner product.

**Corollary 3** (Normalized Atomic Inner Product; NAIP). *Let the atomic inner product be defined by $\langle \boldsymbol{d}_i, \boldsymbol{d}_j \rangle_S = \boldsymbol{d}_i^\top S \boldsymbol{d}_j$, where $S$ is symmetric and positive definite. Suppose the columns of $D = [\boldsymbol{d}_1, \cdots, \boldsymbol{d}_{|D|}]$ form a set of atoms satisfying $\|\boldsymbol{d}_i\|_S = c > 0$ for all $i$. Then, for any $i, j$,*

$$\rho_S(\boldsymbol{d}_i, \boldsymbol{d}_j) = \frac{\langle \boldsymbol{d}_i, \boldsymbol{d}_j \rangle_S}{\|\boldsymbol{d}_i\|_S \|\boldsymbol{d}_j\|_S} = \boldsymbol{d}_i^\top \tilde{S} \boldsymbol{d}_j, \quad \tilde{S} := \frac{1}{c^2} S = (DD^\top)^{-1}. \tag{3.2}$$

*Consequently, the bilinear form $\langle \boldsymbol{d}_i, \boldsymbol{d}_j \rangle_{\tilde{S}} = \boldsymbol{d}_i^\top \tilde{S} \boldsymbol{d}_j$ defines a **normalized atomic inner product**.*

**Remark.** Define $\tilde{\boldsymbol{d}}_i = \tilde{S}^{\frac{1}{2}} \boldsymbol{d}_i$ and $\tilde{\boldsymbol{d}}_j = \tilde{S}^{\frac{1}{2}} \boldsymbol{d}_j$. Under this transformation, $\langle \boldsymbol{d}_i, \boldsymbol{d}_j \rangle_{\tilde{S}} = \langle \tilde{\boldsymbol{d}}_i, \tilde{\boldsymbol{d}}_j \rangle$, where the right-hand side is the standard Euclidean inner product. Hence, properties of the Euclidean inner product transfer directly to the NAIP; $\tilde{\boldsymbol{d}}_i$ and $\tilde{\boldsymbol{d}}_j$ are accordingly termed **normalized atoms**.

### 3.2 FORMAL DEFINITION OF ATOMS

Although an appropriate inner product has been identified to preserve the geometry of representations (Figure 3), substantial superposition (Elhage et al., 2022) persists (Figure 1 (a)), indicating the necessity of further decomposition to obtain genuine atoms. This section presents a formal definition of atoms, preceded by the sparsity assumption and a justification for approximate orthogonality.

**Assumption 4** (Sparsity). *Let $M = \{\boldsymbol{m}_i\}_{i=1}^{|M|} \subset \mathbb{R}^H$ be a set of representations. Assume that there exist $D = [\boldsymbol{d}_1, \cdots, \boldsymbol{d}_{|D|}] \in \mathbb{R}^{H \times |D|}$ and $\Delta = [\boldsymbol{\delta}_1, \cdots, \boldsymbol{\delta}_{|M|}] \in \mathbb{R}_{\geq 0}^{|D| \times |M|}$ such that, $\forall i \in [|M|]$, $\boldsymbol{m}_i = D\boldsymbol{\delta}_i$, $\|\boldsymbol{\delta}_i\|_0 \leq K \ll |D|$, where $\| \cdot \|_0$ denotes the $\ell_0$ norm, and $K$ is a fixed constant.*

Sparsity permits the number of atoms to substantially exceed the ambient dimension, yielding an overcomplete structure with $|D| \gg H$ that provides the degrees of freedom sufficient to capture the richness of world knowledge, while simultaneously ensuring that mutual interference is minimized.

**Remark.** In the basic setting of § 3.1, where $|D| = H$, one verifies that $\tilde{S} = (DD^\top)^{-1}$ satisfies $D^\top \tilde{S} D = I$. When $|D| \gg H$, the matrix $\tilde{S} = (DD^\top)^{-1}$ remains well defined provided $\text{rank}(D) = H$; however, the matrix $G := D^\top \tilde{S} D$ is a projection operator of rank $H$. Hence exact orthogonality cannot be achieved, which motivates the introduction of approximate orthogonality.

This consideration motivates the following definition of $\epsilon$-approximately orthogonal atoms.

**Definition 5** ($\epsilon$-Approximately Orthogonal Atoms). *Let $\langle \boldsymbol{x}, \boldsymbol{y} \rangle_{\tilde{S}} := \boldsymbol{x}^\top \tilde{S} \boldsymbol{y}$ be normalized atomic inner product, where $\tilde{S} := (DD^\top)^{-1}$. The atom set $\{\boldsymbol{d}_i\}_{i=1}^{|D|}$ is said to be $\epsilon$-**approximately orthogonal** if, $\forall i \neq j$, $|\langle \boldsymbol{d}_i, \boldsymbol{d}_j \rangle_{\tilde{S}}| = |\langle \tilde{\boldsymbol{d}}_i, \tilde{\boldsymbol{d}}_j \rangle| \leq \epsilon$, where $\tilde{\boldsymbol{d}}_i := \tilde{S}^{\frac{1}{2}} \boldsymbol{d}_i$ and $\tilde{\boldsymbol{d}}_j := \tilde{S}^{\frac{1}{2}} \boldsymbol{d}_j$.*

Notably, the constraint $|\langle \boldsymbol{d}_i, \boldsymbol{d}_j \rangle_{\tilde{S}}| = |\langle \tilde{\boldsymbol{d}}_i, \tilde{\boldsymbol{d}}_j \rangle| \leq \epsilon$ does not prevent maintaining $\|\tilde{\boldsymbol{d}}_i\| = \|\tilde{\boldsymbol{d}}_j\| = 1$.

**Remark.** In the ideal setting of exact orthogonality, the random variable $\langle \tilde{\boldsymbol{d}}_i, \tilde{\boldsymbol{d}}_j \rangle$, for all $i \neq j$, would follow a Dirac measure concentrated at the origin. Under the practical $\epsilon$-approximate orthogonality, however, $\langle \tilde{\boldsymbol{d}}_i, \tilde{\boldsymbol{d}}_j \rangle$, for all $i \neq j$, is expected to follow a Gaussian distribution $\mathcal{N}(0, s^2)$ with small variance $s^2$, approaching the Dirac measure as $s \to 0$.

We now introduce a formal definition of atoms suitable for practical models.

**Definition 6** (Atoms). *Let $M = \{\boldsymbol{m}_i\}_{i=1}^{|M|}$ be a collection of representations. Assume that there exists a matrix $D = [\boldsymbol{d}_1, \cdots, \boldsymbol{d}_{|D|}] \in \mathbb{R}^{H \times |D|}$ and a sparse coefficient matrix $\Delta = [\boldsymbol{\delta}_1, \cdots, \boldsymbol{\delta}_{|M|}] \in \mathbb{R}_{\geq 0}^{|D| \times |M|}$ such that, for every $i \in [|M|]$ and a fixed sparsity level $K \in \mathbb{N}$,*

$$\boldsymbol{m}_i = D\boldsymbol{\delta}_i, \quad \|\boldsymbol{\delta}_i\|_0 \leq K. \tag{3.3}$$

*Furthermore, for all $i \neq j$, $|\langle \tilde{\boldsymbol{d}}_i, \tilde{\boldsymbol{d}}_j \rangle| \leq \epsilon$, where $\tilde{\boldsymbol{d}}_i := \tilde{S}^{\frac{1}{2}} \boldsymbol{d}_i$, $\tilde{\boldsymbol{d}}_j := \tilde{S}^{\frac{1}{2}} \boldsymbol{d}_j$ and $\tilde{S} := (DD^\top)^{-1}$. Under these conditions, $\{\boldsymbol{d}_i\}_{i=1}^{|D|}$ is called the **atom set** of $M$, and each $\boldsymbol{d}_i$ is referred to as an **atom**.*

It is worth noting that $\boldsymbol{\delta}_i \in \mathbb{R}^{|D|}$ can be interpreted as the sparse representation of higher-dimensional semantics, which is compressed by the matrix $D \in \mathbb{R}^{H \times |D|}$ to yield the representation $\boldsymbol{m}_i$. Premultiplying both sides of the equation by $\tilde{S}^{\frac{1}{2}}$ yields $\tilde{\boldsymbol{m}}_i = \tilde{D}\boldsymbol{\delta}_i$, where $\tilde{\boldsymbol{m}}_i := \tilde{S}^{\frac{1}{2}} \boldsymbol{m}_i$ and $\tilde{D} := \tilde{S}^{\frac{1}{2}} D = [\tilde{\boldsymbol{d}}_1, \cdots, \tilde{\boldsymbol{d}}_{|D|}]$. This transformation simplifies the derivations of subsequent analysis.

### 3.3 FUNDAMENTAL PROPERTIES OF ATOMS

Having introduced the atomic inner product and a formal definition of atoms, this section characterizes atoms in terms of uniqueness, recoverability, and identifiability, drawing on their close connection to compressed sensing (Donoho, 2006), whose core idea is that a high-dimensional signal sparse in some basis can be recovered from far fewer linear measurements than its ambient dimension, with the Restricted Isometry Property (RIP) providing the essential guarantee.

**Definition 7** (Restricted Isometry Property; RIP). *A matrix $\tilde{D} \in \mathbb{R}^{H \times |D|}$ is said to satisfy the $K$-RIP if there exists a constant $\delta_K \in [0, 1)$ such that, for any $K$-sparse vector $\boldsymbol{\delta} \in \mathbb{R}^{|D|}$ (i.e., $\|\boldsymbol{\delta}\|_0 \leq K$),*

$$(1 - \delta_K)\|\boldsymbol{\delta}\|_2^2 \leq \|\tilde{D}\boldsymbol{\delta}\|_2^2 \leq (1 + \delta_K)\|\boldsymbol{\delta}\|_2^2. \tag{3.4}$$

*Here, $\delta_K$ is called the $K$-RIP constant of $\tilde{D}$.*

Intuitively, projecting a sparse vector into a lower-dimensional space via $\tilde{D}$ preserves its geometric structure, ensuring the possibility of recovery. Direct verification of the RIP is NP-hard; however, the coherence provides a computable upper bound on $\delta_k$.

**Theorem 8** (Coherence–RIP Upper Bound). *Let $\tilde{D} \in \mathbb{R}^{H \times |D|}$ and define the coherence $\mu := \max_{i \neq j} |\langle \tilde{\boldsymbol{d}}_i, \tilde{\boldsymbol{d}}_j \rangle| \leq \varepsilon$. For any $K \in \mathbb{N}$ and any $K$-sparse vector $\boldsymbol{\delta} \in \mathbb{R}^{|D|}$ with $\|\boldsymbol{\delta}\|_0 \leq K$,*

$$\left(1 - (K-1)\mu\right) \|\boldsymbol{\delta}\|_2^2 \leq \|\tilde{D}\boldsymbol{\delta}\|_2^2 \leq \left(1 + (K-1)\mu\right) \|\boldsymbol{\delta}\|_2^2. \tag{3.5}$$

*Hence $\delta_K(\tilde{D}) \leq (K-1)\mu$; in particular, $\tilde{D}$ satisfies the $K$-RIP whenever $(K-1)\mu < 1$.*

In other words, coherence provides a computable criterion for verifying the RIP, ensuring that all $K$-sparse vectors projected through the atom set preserve geometric structure, an essential prerequisite in compressed sensing. Nevertheless, the RIP alone does not preclude non-uniqueness: even if $(K-1)\mu < 1$ holds, the sparse coefficients associated with representations need not be unique.

**Theorem 9** (Uniqueness and Exact $\ell_1$ Recoverability). *Let $\tilde{D} \in \mathbb{R}^{H \times |D|}$ and define the coherence $\mu := \max_{i \neq j} |\langle \tilde{\boldsymbol{d}}_i, \tilde{\boldsymbol{d}}_j \rangle| \leq \varepsilon$. If $\mu < \frac{1}{2K-1}$, then for every $\boldsymbol{\delta} \in \mathbb{R}^{|D|}$ with $\|\boldsymbol{\delta}\|_0 \leq K$, the $K$-sparse representation determined by $\tilde{\boldsymbol{m}} = \tilde{D}\boldsymbol{\delta}$ is unique; that is, no other $K$-sparse vector yields the same $\tilde{\boldsymbol{m}}$. Moreover, $\boldsymbol{\delta}$ is the unique minimizer of the convex program*

$$\min_{\boldsymbol{x} \in \mathbb{R}^{|D|}} \|\boldsymbol{x}\|_1 \quad \text{subject to} \quad \tilde{D}\boldsymbol{x} = \tilde{\boldsymbol{m}}. \tag{3.6}$$

These results furnish theoretical guarantees of uniqueness and recoverability for atoms; nevertheless, the problem remains purely theoretical, namely whether such atoms can be identified and recovered in practice. Given that sparse autoencoders (SAEs) are a standard method for obtaining disentangled representations (Cunningham et al., 2023), we next demonstrate that, under appropriate conditions, SAEs can indeed recover such atoms, thereby rendering the theory practically applicable.

**Theorem 10** (Identifiability of SAEs with Threshold Activation). *Let $M = \{\boldsymbol{m}_i\}_{i=1}^{|M|} \subset \mathbb{R}^H$ with $\boldsymbol{m}_i = D\boldsymbol{\delta}_i$, where $D = [\boldsymbol{d}_1, \cdots, \boldsymbol{d}_{|D|}] \in \mathbb{R}^{H \times |D|}$ and satisfies $|\langle \tilde{\boldsymbol{d}}_i, \tilde{\boldsymbol{d}}_j \rangle| \leq \epsilon$ for all $i \neq j$. Suppose each $\boldsymbol{\delta}_i \in \mathbb{R}^{|D|}$ is $K$-sparse, i.e. $\|\boldsymbol{\delta}_i\|_0 \leq K$. Consider the threshold activation function*

$$\sigma_\tau(x) = \begin{cases} 0 & x < \tau, \\ x & x \geq \tau, \end{cases} \tag{3.7}$$

*with threshold $\tau > 0$. Assume there exist constants $0 < \delta_{\min} \leq \delta_{\max} < \infty$ such that, for each $i$ and support $\mathcal{S}_i = \text{supp}(\boldsymbol{\delta}_i)$, $\min_{j \in \mathcal{S}_i} \delta_{ij} \geq \delta_{\min}$ and $\max_{j \in \mathcal{S}_i} \delta_{ij} \leq \delta_{\max}$. If the amplitude gap and threshold satisfy $\varepsilon K \delta_{\max} < \tau < \delta_{\min} - \varepsilon(K-1)\delta_{\max}$, which is feasible whenever $\delta_{\min} > \varepsilon(2K-1)\delta_{\max}$, then setting $W_{dec} = D$ and $W_{enc} = D^\top \tilde{S}$ yields, in a probabilistic sense,*

$$\forall i, \quad W_{dec}\, \sigma_\tau(W_{enc}\boldsymbol{m}_i) = \boldsymbol{m}_i. \tag{3.8}$$

*Hence, under this parameterization, the SAE can identify the target atom set $D$.*

Thus SAEs with threshold activation can, in principle, achieve effective sparse inference of atoms. In contrast, conventional ReLU (Templeton et al., 2024), lacking a threshold term, fails to satisfy the support-separation condition and is therefore theoretically invalid. Although Top$K$ activation (Gao et al., 2024) is equivalent in some respects, its reliance on a fixed $K$ compromises adaptivity and limits practical use. This also responds to O'Neill et al. (2024): the limitation of SAEs does not arise from their "linear–nonlinear" encoding mechanism, but rather from the absence of threshold activation, which prevents ReLU-based SAEs from achieving effective sparse inference.

**Remark.** Although the theorem is formulated with a uniform scalar threshold $\tau$, it extends directly to a coordinate-wise threshold vector $\boldsymbol{\tau}$, with the squeeze condition and proof remaining unaffected. This generalization enlarges the feasible interval when activation magnitudes differ, thereby strengthening support separation and the robustness of atom identification.

## 4 ATOMS IN LLMS

While proposed Atoms Theory provides a rigorous theoretical framework, it awaits empirical verification. This section presents experiments showing that atoms are pervasive in LLM activations and exhibit the predicted properties. Specifically, experiments comprise the following aspects:

- **Sparse reconstruction.** Training single-layer SAEs with threshold activation on Gemma2-2B, Gemma2-9B, and Llama3.1-8B achieves 99.9% $R^2$ on average, with further analyses confirming that SAEs learn atomic structures, verifying ***identifiability*** and ***recoverability***.

- **Atomicity test.** The learned atoms exhibit pervasive approximate orthogonality, with NAIP distributions closely resembling the Dirac measure as predicted, verifying ***atomicity***.

- **Comparative analysis.** Atoms outperform neurons and features in ***stability***, satisfying the uniqueness condition in 99.8% of cases on average versus 0.5% and 68.2%, respectively.

- **Scaling experiments.** Experiments across varying model scales and dataset sizes reveal how SAE capacity governs recovery performance, exhibiting ***scalability***.

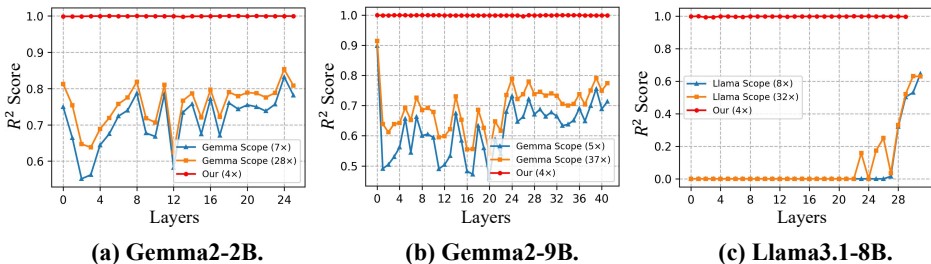

**(a) Gemma2-2B.**  **(b) Gemma2-9B.**  **(c) Llama3.1-8B.**

Figure 4: Sparse reconstruction $R^2$ scores across models. GemmaScope and LlamaScope serve as standard tools for extracting features from representations. The coefficient before $\times$ denotes the ratio of SAE hidden size to representation dimensionality. $R^2$ values below 0 are clipped to 0.

## 4.1 SPARSE RECONSTRUCTION

**Experimental Setup**   We employ single-layer SAEs with threshold activation, denoted as $f : \boldsymbol{x} \mapsto \hat{\boldsymbol{x}} = W_{\text{dec}}\,\sigma(W_{\text{enc}}\boldsymbol{x})$, and train it by minimizing a joint reconstruction–sparsity objective

$$\mathcal{L}(\boldsymbol{x}) = \underbrace{\|\boldsymbol{x} - \hat{\boldsymbol{x}}\|_2^2}_{\mathcal{L}_{\text{reconstruct}}} + \lambda \underbrace{\|\sigma(\boldsymbol{z})\|_1}_{\mathcal{L}_{\text{sparsity}}}, \tag{4.1}$$

where $\boldsymbol{z} = W_{\text{enc}}\boldsymbol{x}$, and $\sigma$ is coordinate-wise JumpReLU activation (Rajamanoharan et al., 2024b),

$$\big(\sigma(\boldsymbol{z})\big)_i = \begin{cases} 0, & z_i < \tau_i, \\ z_i, & z_i \geq \tau_i, \end{cases} \quad \boldsymbol{\tau} = (\tau_i)_i. \tag{4.2}$$

For training, we extract knowledge activations for all entities in the Counterfact dataset (Meng et al., 2022), across all layers of the model, yielding 20,391 activations per layer. The scaling experiments extend this up to 73,728 knowledge activations corresponding to WikiData entities (Vrandečić & Krötzsch, 2014). Comprehensive details of data, training and baselines are provided in Appendix C.

**Evaluation Metrics**   We adopt the coefficient of determination as the evaluation metric,

$$R^2 = 1 - \frac{\sum_i \|\boldsymbol{x}_i - \hat{\boldsymbol{x}}_i\|_2^2}{\sum_i \|\boldsymbol{x}_i - \bar{\boldsymbol{x}}\|_2^2}, \tag{4.3}$$

where $\bar{\boldsymbol{x}}$ denotes the sample mean, and $R^2 = 1$ indicates perfect reconstruction.

**Main Results**   As illustrated in Figure 4, SAEs achieve consistently high reconstruction fidelity across layers of Gemma2-2B, Gemma2-9B, and Llama3.1-8B, showing mean $R^2$ scores of 99.92%, 99.93%, and 99.85%, with sparsity details provided in Appendix C.7. This result is consistent with the identifiability guarantee of Theorem 10 and the recoverability guarantee of Theorem 9.

The reconstruction performance is largely insensitive to hyperparameters, with nearly identical learning curves for $\lambda \in \{0.01, 0.1, 1\}$ (Appendix C.3), indicating that high-fidelity reconstruction reflects the inherent sparsifiability of the representations rather than an artifact of meticulous tuning.

The encoder and decoder of SAEs converge to alignment under atomic inner product, namely parameterization of $W_{dec}=D$ and $W_{enc}=D^\top \tilde{S}$, consistent with Theorem 10. This holds even under random initialization and independent training without weight tying or other constraints, as shown in Figure 5 for Gemma2-2B, with analogous results for Gemma2-9B and Llama3.1-8B in Appendix C.7.

## 4.2 ATOMICITY TEST

By Definition 6, atoms must satisfy approximate orthogonality under the normalized atomic inner product (NAIP), a property referred to as atomicity, ensuring their mutual distinguishability.

The NAIP among all atoms can be computed by directly evaluating the matrix $G = \tilde{D}^\top \tilde{D}$, with a more practical procedure, similar to Corollary 3, given by

$$G = \frac{D^\top S D}{\sqrt{\text{diag}(D^\top S D)} \times \sqrt{\text{diag}(D^\top S D)}}, \tag{4.4}$$

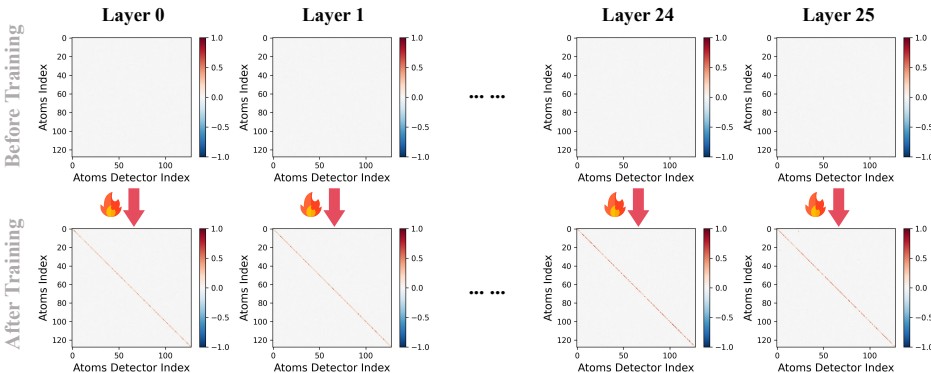

Figure 5: Spontaneous alignment between the encoder and decoder during training on Gemma2-2B.

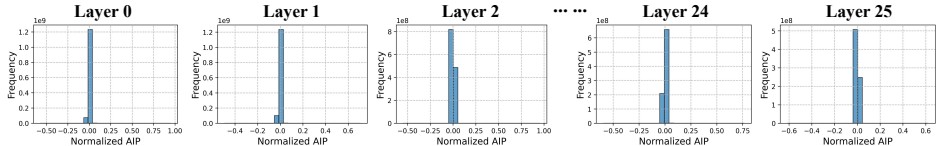

Figure 6: NAIP distribution of atoms on Gemma2-2B, with additional models in Appendix C.7.

where $S = (DD^\top)^{-1}$, $\text{diag}(D^\top S D)$ denotes the diagonal of $D^\top S D$, $\sqrt{\text{diag}(D^\top S D)}$ denotes its element-wise square root, and $\times$ indicates the outer product. If the vectors learned by the SAEs exhibit atomicity, the off-diagonal elements $G_{ij} = \langle \tilde{d}_i, \tilde{d}_j \rangle$ should cluster near zero with very small variance, demonstrating approximate orthogonality, while the diagonal entries are normalized.

As shown in Figure 6, across all layers of Gemma2-2B, Gemma2-9B, and Llama3.1-8B, the matrices $D$ learned by SAEs exhibit strong atomicity: the off-diagonal elements are tightly concentrated near zero, closely matching the theoretical Dirac delta distribution. This accords with Definition 5: although strict orthogonality is unattainable, sparsity and overcompleteness drive convergence to approximately orthogonal atoms. Additionally, case studies of atoms are provided in Appendix C.6.

### 4.3 NEURONS V.S. FEATURES V.S. ATOMS

According to Theorem 9, the evaluation of fundamental units in LLMs reduces to a verifiable criterion: whether representations can be uniquely and stably recovered from such units, which holds whenever $\mu < \frac{1}{2K-1}$, with $\mu$ denoting coherence and $K$ denoting sparsity. Building on this theoretical foundation, we next compare the practical performance of neurons, features, and atoms.

Specifically, we introduce quantile statistics to capture $\mu < \frac{1}{2K-1}$, thereby ensuring robustness. Quantile coherence $\mu_q$ and sparsity $K_q$ (details in Appendix C.5) ensure that whenever $\mu_q < \frac{1}{2K_q-1}$, at least a fraction $q$ of samples satisfy the sufficient conditions for uniqueness and recoverability.

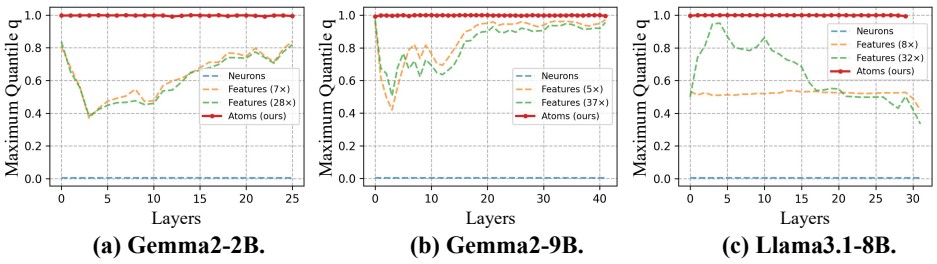

Figure 7: Maximum quantile $q$ per layer satisfying $\mu_q < \frac{1}{2K_q-1}$ for each model.

As demonstrated in Figure 7, the atoms exhibit significant uniqueness and recoverability, achieving average rates of 99.74% on Gemma2-2B, 99.88% on Gemma2-9B, and 99.95% on Llama3.1-8B, compared with 68.25% for features and 0.45% for neurons. This demonstrates that atoms, with superior stability, constitute a more reliable fundamental unit than neurons and features.

### 4.4 SCALING SAEs

Finally, we investigate how varying SAE scales affect recovery performance. Specifically, we train SAEs of varying sizes on Gemma2-2B, gradually increasing capacity to accommodate increasing dataset size. The results in Figure 8 show that as SAE size increases, reconstruction accuracy improves and then stabilizes; corresponding sparsity details are provided in Appendix C.7. However, as the dataset size increases, the optimal SAE size no longer grows linearly, suggesting that the required atomic information is much smaller than the raw number of activations.

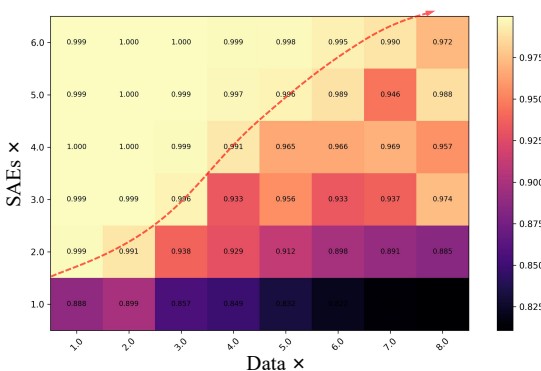

Figure 8: Scaling experiments on Gemma2-2B.

## 5 RELATED WORK

**Neurons.** Early interpretability studies considered neurons as the smallest computational units in neural networks. Bills et al. (2023) attempted to automatically generate functional explanations for neurons in language models, while Geva et al. (2020) analyzed their role in knowledge recall. However, this view faces the polysemy problem, where the same neuron often activates for multiple semantically unrelated patterns (Olah et al., 2020), a phenomenon that Elhage et al. (2022) attributed to superposition. Collectively, these studies suggest that neurons are unsuitable as the fundamental units of neural networks. Thus, Olah et al. (2020) prompted a shift in focus from neurons to features.

**Features.** Although there was no unified formal definition of "feature" at its inception (Elhage et al., 2022), it is commonly understood as a linear direction with specific meaning (Hewitt & Manning, 2019; Park et al., 2023; Gurnee et al., 2023; Chen et al., 2025). Cunningham et al. (2023) introduced sparse autoencoders (SAEs) to learn features in language models, with subsequent work Gao et al. (2024) and Templeton et al. (2024) expanding SAEs to larger scales. Rajamanoharan et al. (2024a) and Rajamanoharan et al. (2024b) optimized the architecture to mitigating feature shrinkage (Wright & Sharkey, 2024). Lieberum et al. (2024) and He et al. (2024) trained SAEs and made them open source, facilitating broader use in the community. However, open issues persist: existing SAEs fail to achieve complete reconstruction, with the unreconstructed component termed "dark matter" (Engels et al., 2024), and instability from feature splitting and merging (Bussmann et al., 2025; Chanin et al., 2025) limits their suitability as basic units. To address these issues, we propose atoms as the fundamental unit for mechanistic interpretability, developing Atoms Theory, which provides principled guarantees through definition, theoretical analysis, and empirical validation.

## 6 CONCLUSION AND FUTURE WORK

This paper introduces and validates the Atoms Theory for characterizing the fundamental units in the high-dimensional representation space of large language models. We provide a formal definition of atoms, demonstrate the conditions under which atoms satisfy the Restricted Isometry Property, and further prove the uniqueness and exact $\ell_1$ recoverability of sparse representations. We also prove that sparse autoencoders with threshold activation can identify and recover these atoms. Empirical results across multiple models confirm that the learned atoms exhibit the predicted properties of atomicity and stability, providing a novel theoretical framework for mechanistic interpretability.

In future work, we aim to further expand Atoms Theory and develop more computationally efficient tools for the widespread identification of atoms. Additionally, we plan to explore the feasibility of using Atoms Theory as a bottom-up approach to further understand large language models.

## ETHICS STATEMENT

All authors have read and comply with the ICLR Code of Ethics. This study uses only publicly available large language models and standard open-access datasets, all of which adhere to established data-privacy, licensing, and usage policies, and does not involve any sensitive personal data. While we acknowledge the broader societal risks associated with language models, such as potential biases, our work does not introduce methods intended for harmful applications. To mitigate risk and ensure transparency, we will release all code, data, and documentation under appropriate open-source licenses and in accordance with relevant legal and ethical standards.

## REPRODUCIBILITY STATEMENT

We have made extensive efforts to ensure full reproducibility of our study. Appendix A provides complete explanations and proofs for the theoretical results of Section 3; Appendix B describes the experimental settings and complete results for the analysis of representation shifting across all layers of multiple language-model families described in Section 3; and Appendix C presents the complete experimental configurations and supplementary results for Section 4. In addition, the supplementary materials include all data, code, documentation, interactive notebooks, and step-by-step reproduction instructions to enable independent verification of all findings in our work.

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

# A PROOFS

## A.1 EXPLANATION FOR EQUATION 3.1

For

$$W_U{}' \leftarrow A^{-\top} W_U + \boldsymbol{b} \cdot \mathbf{1}^\top, \; \boldsymbol{h}'^L \leftarrow A\boldsymbol{h}^L, \tag{3.1}$$

we provide a simple derivation as follows:

$$W_U'{}^\top \boldsymbol{h}' = (A^{-\top} W_U + \boldsymbol{b} \cdot \mathbf{1}^\top)^\top (A\boldsymbol{h}) \tag{A.1}$$

$$= W_U^\top (A^{-1} A)\boldsymbol{h} + \mathbf{1}(\boldsymbol{b}^\top A\boldsymbol{h}) \tag{A.2}$$

$$= W_U^\top \boldsymbol{h} + c(\boldsymbol{h})\mathbf{1}, \tag{A.3}$$

where $c(\boldsymbol{h}) = \boldsymbol{b}^\top A\boldsymbol{h} \in \mathbb{R}$ is a scalar. Using the translation invariance property of softmax, $\text{Softmax}(\boldsymbol{z} + c\mathbf{1}) = \text{Softmax}(\boldsymbol{z})$, the result follows.

It is important to note that this is the only form, meaning that $W_U$ can only be identified up to an invertible transformation plus a bias, and $\boldsymbol{h}$ can only be identified up to an invertible transformation.

## A.2 PROOF OF THEOREM 2

**Theorem 2** (Explicit Form of the Atomic Inner Product). *Let $\langle \boldsymbol{d}_i, \boldsymbol{d}_j \rangle_S = \boldsymbol{d}_i^\top S\boldsymbol{d}_j$ be an atomic inner product with $S \in \mathbb{R}^{H \times H}$ symmetric and positive definite. If the columns of $D = [\boldsymbol{d}_1, \boldsymbol{d}_2, \cdots, \boldsymbol{d}_{|D|}]$ form a set of atoms such that $\|\boldsymbol{d}_i\|_S = c > 0$ for all $i$, and $\mathcal{D} \simeq \mathbb{R}^H$, then $S = c^2(DD^\top)^{-1}$.*

*Proof.* Since $\langle \cdot, \cdot \rangle_S$ is atomic inner product, for any pair of atoms $\boldsymbol{d}_i$ and $\boldsymbol{d}_j$ we have

$$\boldsymbol{d}_i^\top S\boldsymbol{d}_j = \langle \boldsymbol{d}_i, \boldsymbol{d}_j \rangle_S = \begin{cases} 0 & i \neq j, \\ c^2 & i = j. \end{cases} \tag{A.4}$$

Applying this property to the atom set $D = [\boldsymbol{d}_1, \boldsymbol{d}_2, \cdots, \boldsymbol{d}_{|D|}]$ yields

$$c^2 I = D^\top S D. \tag{A.5}$$

Since $\{\boldsymbol{d}_1, \boldsymbol{d}_2, \cdots, \boldsymbol{d}_{|D|}\}$ spans the atomic space $\mathcal{D}$, and $\mathcal{D} \simeq \mathbb{R}^H$, it follows that $\text{rank}(D) = H$. Let $S^{1/2}$ denote the symmetric positive-definite square root of $S$. Then

$$D^\top S D = (S^{1/2} D)^\top (S^{1/2} D), \tag{A.6}$$

which implies $\text{rank}(I) = \text{rank}(D^\top S D) = \text{rank}(S^{1/2} D) = \text{rank}(D)$. Therefore, $\text{rank}(D) = |D| \leq H$. Combining this with the earlier condition gives $|D| = \text{rank}(D) = H$, which shows that $D$ is invertible. Consequently,

$$S = c^2(DD^\top)^{-1}. \tag{A.7}$$

$\square$

## A.3 PROOF OF COROLLARY 3

**Corollary 3** (Normalized Atomic Inner Product; NAIP). *Let the atomic inner product be defined by $\langle \boldsymbol{d}_i, \boldsymbol{d}_j \rangle_S = \boldsymbol{d}_i^\top S\boldsymbol{d}_j$, where $S$ is symmetric and positive definite. Suppose the columns of $D = [\boldsymbol{d}_1, \cdots, \boldsymbol{d}_{|D|}]$ form a set of atoms satisfying $\|\boldsymbol{d}_i\|_S = c > 0$ for all $i$. Then, for any $i, j$,*

$$\rho_S(\boldsymbol{d}_i, \boldsymbol{d}_j) = \frac{\langle \boldsymbol{d}_i, \boldsymbol{d}_j \rangle_S}{\|\boldsymbol{d}_i\|_S \|\boldsymbol{d}_j\|_S} = \boldsymbol{d}_i^\top \tilde{S}\boldsymbol{d}_j, \quad \tilde{S} := \frac{1}{c^2} S = (DD^\top)^{-1}. \tag{3.2}$$

*Consequently, the bilinear form $\langle \boldsymbol{d}_i, \boldsymbol{d}_j \rangle_{\tilde{S}} = \boldsymbol{d}_i^\top \tilde{S}\boldsymbol{d}_j$ defines a **normalized atomic inner product**.*

*Proof.* Since $\langle \boldsymbol{d}_i, \boldsymbol{d}_j \rangle_S = 0$ for $i \neq j$, and $\|\boldsymbol{d}_i\|_S = \|\boldsymbol{d}_j\|_S = c > 0$, it follows that $D^\top S D = c^2 I$. Therefore, $\tilde{S} = \frac{1}{c^2} S$ satisfies $D^\top \tilde{S} D = I$, which implies that the atoms are orthonormal. Since $D$ is invertible, we also have $\tilde{S} = D^{-\top} D^{-1} = (DD^\top)^{-1}$, and thus $\langle \cdot, \cdot \rangle_{\tilde{S}}$ is a symmetric positive-definite inner product. $\square$

## A.4 PROOF OF THEOREM 8

**Theorem 8** (Coherence–RIP Upper Bound). *Let $\tilde{D} \in \mathbb{R}^{H \times |D|}$ and define the coherence $\mu :=$ $\max_{i \neq j} |\langle \tilde{d}_i, \tilde{d}_j \rangle| \leq \varepsilon$. For any $K \in \mathbb{N}$ and any $K$-sparse vector $\boldsymbol{\delta} \in \mathbb{R}^{|D|}$ with $\|\boldsymbol{\delta}\|_0 \leq K$,*

$$\left(1 - (K-1)\mu\right) \|\boldsymbol{\delta}\|_2^2 \leq \|\tilde{D}\boldsymbol{\delta}\|_2^2 \leq \left(1 + (K-1)\mu\right) \|\boldsymbol{\delta}\|_2^2. \tag{3.5}$$

*Hence $\delta_K(\tilde{D}) \leq (K-1)\mu$; in particular, $\tilde{D}$ satisfies the $K$-RIP whenever $(K-1)\mu < 1$.*

*Proof.* Let $\mathrm{supp}(\boldsymbol{\delta}) = \mathcal{S} \subseteq [|D|]$ and $|\mathcal{S}| \leq K$. Then, we have

$$\|\tilde{D}\boldsymbol{\delta}\|_2^2 = \boldsymbol{\delta}^\top (\tilde{D}^\top \tilde{D})\boldsymbol{\delta} = \sum_{i \in \mathcal{S}} \delta_i^2 + 2 \sum_{\substack{i < j \\ i,j \in \mathcal{S}}} \delta_i \delta_j \langle \tilde{d}_i, \tilde{d}_j \rangle. \tag{A.8}$$

By the fact that $|\langle \tilde{d}_i, \tilde{d}_j \rangle| \leq \mu$ and applying the triangle inequality, we obtain:

$$\|\tilde{D}\boldsymbol{\delta}\|_2^2 \geq \sum_{i \in \mathcal{S}} \delta_i^2 - 2\mu \sum_{i < j} |\delta_i \delta_j|, \tag{A.9}$$

$$\|\tilde{D}\boldsymbol{\delta}\|_2^2 \leq \sum_{i \in \mathcal{S}} \delta_i^2 + 2\mu \sum_{i < j} |\delta_i \delta_j|. \tag{A.10}$$

Next, we observe that:

$$\left(\sum_{i \in \mathcal{S}} |\delta_i|\right)^2 = \sum_{i \in \mathcal{S}} \delta_i^2 + 2 \sum_{i < j} |\delta_i \delta_j| \leq |\mathcal{S}| \sum_{i \in \mathcal{S}} \delta_i^2 \leq K \sum_{i \in \mathcal{S}} \delta_i^2. \tag{A.11}$$

Thus, we have $2 \sum_{i < j} |\delta_i \delta_j| \leq (K-1) \sum_{i \in \mathcal{S}} \delta_i^2$. Substituting this back, we conclude the proof. $\square$

## A.5 PROOF OF THEOREM 9

**Theorem 9** (Uniqueness and Exact $\ell_1$ Recoverability). *Let $\tilde{D} \in \mathbb{R}^{H \times |D|}$ and define the coherence $\mu := \max_{i \neq j} |\langle \tilde{d}_i, \tilde{d}_j \rangle| \leq \varepsilon$. If $\mu < \frac{1}{2K-1}$, then for every $\boldsymbol{\delta} \in \mathbb{R}^{|D|}$ with $\|\boldsymbol{\delta}\|_0 \leq K$, the $K$-sparse representation determined by $\tilde{m} = \tilde{D}\boldsymbol{\delta}$ is unique; that is, no other $K$-sparse vector yields the same $\tilde{m}$. Moreover, $\boldsymbol{\delta}$ is the unique minimizer of the convex program*

$$\min_{\boldsymbol{x} \in \mathbb{R}^{|D|}} \|\boldsymbol{x}\|_1 \quad \text{subject to} \quad \tilde{D}\boldsymbol{x} = \tilde{m}. \tag{3.6}$$

*Proof.* We first prove that under the condition $\mu < \frac{1}{2K-1}$, the $K$-sparse representation is unique.

Suppose there exist two distinct $K$-sparse coefficient vectors $\boldsymbol{\delta}, \boldsymbol{\delta}'$ such that $\tilde{D}\boldsymbol{\delta} = \tilde{D}\boldsymbol{\delta}'$. Let $\boldsymbol{h} = \boldsymbol{\delta} - \boldsymbol{\delta}' \neq \mathbf{0}$. Then $\tilde{D}\boldsymbol{h} = 0$ and $\|\boldsymbol{h}\|_0 \leq 2K$. By Theorem 8 (applied with $K$ replaced by $2K$), we have

$$\left(1 - (2K-1)\mu\right) \|\boldsymbol{h}\|_2^2 \leq \|\tilde{D}\boldsymbol{h}\|_2^2 = 0. \tag{A.12}$$

If $\mu < \frac{1}{2K-1}$, then the prefactor on the left is strictly positive, which forces $\|\boldsymbol{h}\|_2 = 0$. This contradicts $\boldsymbol{h} \neq \mathbf{0}$. Hence uniqueness holds.

Next, we prove that under the same condition $\mu < \frac{1}{2K-1}$, the sparse vector $\boldsymbol{\delta}$ is also the unique solution of the convex optimization problem

$$\min_{\boldsymbol{x} \in \mathbb{R}^{|D|}} \|\boldsymbol{x}\|_1 \quad \text{s.t.} \quad \tilde{D}\boldsymbol{x} = \tilde{m}. \tag{A.13}$$

The overall strategy is as follows: (i) show that the null space property of order $K$ ($\mathrm{NSP}_K$) holds under the assumption $\mu < \frac{1}{2K-1}$; (ii) recall the equivalence $\mathrm{NSP}_K \iff$ exact and unique recovery of any $K$-sparse solution via noiseless $\ell_1$-minimization.

Formally, the null space property of order $K$ ($\mathrm{NSP}_K$) is defined as

$$\forall \boldsymbol{h} \in \ker(\tilde{D}) \setminus \{\mathbf{0}\}, \ \forall \mathcal{S} \subseteq [n], \ |\mathcal{S}| \leq K : \quad \boxed{\|\boldsymbol{h}_\mathcal{S}\|_1 < \|\boldsymbol{h}_{\mathcal{S}^c}\|_1}, \tag{A.14}$$

where $\ker(\tilde{D}) = \{\boldsymbol{h} : \tilde{D}\boldsymbol{h} = \mathbf{0}\}$, $\mathcal{S} \subseteq [n]$ is an index set with $[n] = \{1, \dots, n\}$, $\boldsymbol{h}_\mathcal{S}$ denotes the restriction of $\boldsymbol{h}$ to the coordinates in $\mathcal{S}$ (with other entries set to zero), and $\mathcal{S}^c = [n] \setminus \mathcal{S}$ is a complementary set.

**Step (i): Proof of NSP$_K$.** Let $G = \tilde{D}^\top \tilde{D}$ denote the Gram matrix. Since each column of $\tilde{D}$ is normalized, we have $G_{jj} = 1$ and $|G_{ij}| \leq \mu$ for $i \neq j$. Take any $\boldsymbol{h} \in \ker(\tilde{D}) \setminus \{\boldsymbol{0}\}$ and any index set $\mathcal{S}$ with $|\mathcal{S}| = K$.

Since $\tilde{D}\boldsymbol{h} = 0$, we have $G\boldsymbol{h} = 0$. For any $j$,

$$0 = (G\boldsymbol{h})_j = \sum_i G_{ji} h_i = G_{jj} h_j + \sum_{i \neq j} G_{ji} h_i \quad \Rightarrow \quad h_j = -\sum_{i \neq j} G_{ji} h_i. \tag{A.15}$$

Taking absolute values and using $|G_{ji}| \leq \mu$, we obtain

$$|h_j| \leq \mu \sum_{i \neq j} |h_i|. \tag{A.16}$$

Summing over $j \in \mathcal{S}$ gives

$$\sum_{j \in \mathcal{S}} |h_j| \leq \mu \sum_{j \in \mathcal{S}} \sum_{i \neq j} |h_i|. \tag{A.17}$$

The inner summation can be decomposed into contributions from $i \in \mathcal{S} \setminus \{j\}$ and $i \in \mathcal{S}^c$:

$$\sum_{j \in \mathcal{S}} \sum_{i \neq j} |h_i| = \underbrace{\sum_{j \in \mathcal{S}} \sum_{\substack{i \in \mathcal{S} \\ i \neq j}} |h_i|}_{\text{each } i \in \mathcal{S} \text{ counted } K-1 \text{ times}} + \underbrace{\sum_{j \in \mathcal{S}} \sum_{i \in \mathcal{S}^c} |h_i|}_{\text{each } i \in \mathcal{S}^c \text{ counted } K \text{ times}}. \tag{A.18}$$

Hence,

$$\|\boldsymbol{h}_{\mathcal{S}}\|_1 \leq \mu\big((K-1)\|\boldsymbol{h}_{\mathcal{S}}\|_1 + K\|\boldsymbol{h}_{\mathcal{S}^c}\|_1\big). \tag{A.19}$$

Rearranging,

$$\big(1 - (K-1)\mu\big)\|\boldsymbol{h}_{\mathcal{S}}\|_1 \leq K\mu \|\boldsymbol{h}_{\mathcal{S}^c}\|_1. \tag{A.20}$$

Dividing through by the positive factor $1 - (K-1)\mu$, define

$$\alpha := \frac{K\mu}{1 - (K-1)\mu}. \tag{A.21}$$

When $\mu < \frac{1}{2K-1}$, we have $\alpha < 1$. Since $\boldsymbol{h} \neq 0$ and $\tilde{D}\boldsymbol{h} = 0$, it is impossible for $\|\boldsymbol{h}_{\mathcal{S}^c}\|_1 = 0$ (otherwise both terms would vanish, forcing $\boldsymbol{h} = 0$, a contradiction). Therefore,

$$\|\boldsymbol{h}_{\mathcal{S}}\|_1 \leq \alpha \|\boldsymbol{h}_{\mathcal{S}^c}\|_1 < \|\boldsymbol{h}_{\mathcal{S}^c}\|_1. \tag{A.22}$$

Take any $\mathcal{S}_0$ with $|\mathcal{S}_0| = k \leq K$, and extend it to a superset $\mathcal{S} \supseteq \mathcal{S}_0$ such that $|\mathcal{S}| = K$. Then,

$$\|\boldsymbol{h}_{\mathcal{S}_0}\|_1 \leq \|\boldsymbol{h}_{\mathcal{S}}\|_1, \qquad \|\boldsymbol{h}_{\mathcal{S}_0^c}\|_1 \geq \|\boldsymbol{h}_{\mathcal{S}^c}\|_1. \tag{A.23}$$

If we know that $\|\boldsymbol{h}_{\mathcal{S}}\|_1 < \|\boldsymbol{h}_{\mathcal{S}^c}\|_1$ holds for all $\mathcal{S}$ of size $K$, then it follows that

$$\|\boldsymbol{h}_{\mathcal{S}_0}\|_1 \leq \|\boldsymbol{h}_{\mathcal{S}}\|_1 < \|\boldsymbol{h}_{\mathcal{S}^c}\|_1 \leq \|\boldsymbol{h}_{\mathcal{S}_0^c}\|_1. \tag{A.24}$$

Thus, the inequality also holds for any $\mathcal{S}_0$ with $|\mathcal{S}_0| \leq K$, which establishes NSP$_K$.

**Step (ii): Equivalence between NSP$_K$ and $\ell_1$ recovery.**

**NSP$_K$ $\Rightarrow$ unique $\ell_1$ recovery:** Suppose $\hat{\boldsymbol{x}}$ is another feasible solution such that $\tilde{D}\hat{\boldsymbol{x}} = \tilde{D}\boldsymbol{\delta}$. Let $\boldsymbol{h} = \hat{\boldsymbol{x}} - \boldsymbol{\delta} \in \ker(\tilde{D}) \setminus \{\boldsymbol{0}\}$, and let $\mathcal{S} = \mathrm{supp}(\boldsymbol{\delta})$ with $|\mathcal{S}| \leq K$. Then

$$\|\hat{\boldsymbol{x}}\|_1 = \|\boldsymbol{\delta} + \boldsymbol{h}\|_1 = \|\boldsymbol{\delta}_{\mathcal{S}} + \boldsymbol{h}_{\mathcal{S}}\|_1 + \|\boldsymbol{h}_{\mathcal{S}^c}\|_1 \geq \|\boldsymbol{\delta}_{\mathcal{S}}\|_1 - \|\boldsymbol{h}_{\mathcal{S}}\|_1 + \|\boldsymbol{h}_{\mathcal{S}^c}\|_1 > \|\boldsymbol{\delta}_{\mathcal{S}}\|_1 = \|\boldsymbol{\delta}\|_1, \tag{A.25}$$

where the strict inequality follows from NSP$_K$. Hence, $\boldsymbol{\delta}$ is the unique minimizer of the $\ell_1$ problem.

**Unique $\ell_1$ recovery $\Rightarrow$ NSP$_K$:** We argue by contradiction. If NSP$_K$ does not hold, then there exists $\boldsymbol{h} \in \ker(\tilde{D}) \setminus \{\boldsymbol{0}\}$ and some $\mathcal{S}$ with $|\mathcal{S}| \leq K$ such that $\|\boldsymbol{h}_{\mathcal{S}}\|_1 \geq \|\boldsymbol{h}_{\mathcal{S}^c}\|_1$. Take any nonzero $K$-sparse $\boldsymbol{\delta}$ with $\mathrm{supp}(\boldsymbol{\delta}) = \mathcal{S}$, and choose $\delta_j = \alpha_j \, \mathrm{sgn}(h_j)$ with $\alpha_j \geq |h_j|$ coordinate-wise. Consider $\hat{\boldsymbol{x}} = \boldsymbol{\delta} - \boldsymbol{h}$. Since $\tilde{D}\boldsymbol{h} = \boldsymbol{0}$, both $\boldsymbol{\delta}$ and $\hat{\boldsymbol{x}}$ are feasible, and

$$\|\hat{\boldsymbol{x}}\|_1 = \|\boldsymbol{\delta}_{\mathcal{S}} - \boldsymbol{h}_{\mathcal{S}}\|_1 + \|\boldsymbol{h}_{\mathcal{S}^c}\|_1 = \|\boldsymbol{\delta}_{\mathcal{S}}\|_1 - \|\boldsymbol{h}_{\mathcal{S}}\|_1 + \|\boldsymbol{h}_{\mathcal{S}^c}\|_1 \ \leq \ \|\boldsymbol{\delta}_{\mathcal{S}}\|_1 = \|\boldsymbol{\delta}\|_1. \tag{A.26}$$

Thus, $\boldsymbol{\delta}$ is not the unique minimizer of the $\ell_1$ problem (and may even fail to be a minimizer). This contradicts the uniqueness assumption. Therefore, NSP$_K$ must hold. $\qquad\square$

### A.6 PROOF OF THEOREM 10

**Theorem 10** (Identifiability of SAEs with Threshold Activation). *Let $M = \{\boldsymbol{m}_i\}_{i=1}^{|M|} \subset \mathbb{R}^H$ with $\boldsymbol{m}_i = D\boldsymbol{\delta}_i$, where $D = [\boldsymbol{d}_1, \cdots, \boldsymbol{d}_{|D|}] \in \mathbb{R}^{H \times |D|}$ and satisfies $|\langle \tilde{\boldsymbol{d}}_i, \tilde{\boldsymbol{d}}_j \rangle| \leq \epsilon$ for all $i \neq j$. Suppose each $\boldsymbol{\delta}_i \in \mathbb{R}^{|D|}$ is $K$-sparse, i.e. $\|\boldsymbol{\delta}_i\|_0 \leq K$. Consider the threshold activation function*

$$\sigma_\tau(x) = \begin{cases} 0 & x < \tau, \\ x & x \geq \tau, \end{cases} \tag{3.7}$$

*with threshold $\tau > 0$. Assume there exist constants $0 < \delta_{\min} \leq \delta_{\max} < \infty$ such that, for each $i$ and support $\mathcal{S}_i = \mathrm{supp}(\boldsymbol{\delta}_i)$, $\min_{j \in \mathcal{S}_i} \delta_{ij} \geq \delta_{\min}$ and $\max_{j \in \mathcal{S}_i} \delta_{ij} \leq \delta_{\max}$. If the amplitude gap and threshold satisfy $\varepsilon K \delta_{\max} < \tau < \delta_{\min} - \varepsilon(K-1)\delta_{\max}$, which is feasible whenever $\delta_{\min} > \varepsilon(2K-1)\delta_{\max}$, then setting $W_{dec} = D$ and $W_{enc} = D^\top \tilde{S}$ yields, in a probabilistic sense,*

$$\forall i, \quad W_{dec}\, \sigma_\tau(W_{enc}\boldsymbol{m}_i) = \boldsymbol{m}_i. \tag{3.8}$$

*Hence, under this parameterization, the SAE can identify the target atom set $D$.*

*Proof.* Consider a single-layer linear–nonlinear encoder of the form $W_{\mathrm{dec}}\, \sigma_\tau(W_{\mathrm{enc}}\boldsymbol{m}_i)$, with training objective

$$W_{\mathrm{dec}}\, \sigma_\tau(W_{\mathrm{enc}}\boldsymbol{m}_i) = \boldsymbol{m}_i, \quad \forall i. \tag{A.27}$$

Set $W_{\mathrm{dec}} = D$ and $W_{\mathrm{enc}} = D^\top \tilde{S}$. Denote $\mathcal{S}_i = \mathrm{supp}(\boldsymbol{\delta}_i)$. Then

$$D^\top \tilde{S}\boldsymbol{m}_i = \begin{bmatrix} \boldsymbol{d}_1^\top \\ \boldsymbol{d}_2^\top \\ \vdots \\ \boldsymbol{d}_{|D|}^\top \end{bmatrix} \tilde{S} \begin{bmatrix} \boldsymbol{d}_1 & \boldsymbol{d}_2 & \cdots & \boldsymbol{d}_{|D|} \end{bmatrix} \boldsymbol{\delta}_i \tag{A.28}$$

$$= \begin{bmatrix} \boldsymbol{d}_1^\top \tilde{S}\boldsymbol{d}_1 & \cdots & \boldsymbol{d}_1^\top \tilde{S}\boldsymbol{d}_{|D|} \\ \boldsymbol{d}_2^\top \tilde{S}\boldsymbol{d}_1 & \cdots & \boldsymbol{d}_2^\top \tilde{S}\boldsymbol{d}_{|D|} \\ \vdots & \ddots & \vdots \\ \boldsymbol{d}_{|D|}^\top \tilde{S}\boldsymbol{d}_1 & \cdots & \boldsymbol{d}_{|D|}^\top \tilde{S}\boldsymbol{d}_{|D|} \end{bmatrix} \boldsymbol{\delta}_i \tag{A.29}$$

$$= G\boldsymbol{\delta}_i, \quad G := D^\top \tilde{S}D. \tag{A.30}$$

By NAIP, we have $G_{kk} = 1$ and for $k \neq j$, $|G_{kj}| = |\langle \tilde{\boldsymbol{d}}_k, \tilde{\boldsymbol{d}}_j \rangle| \leq \varepsilon$. Thus, for any index $k$,

$$(G\boldsymbol{\delta}_i)_k = \begin{cases} \delta_{ik} + \underbrace{\sum\limits_{j \in \mathcal{S}_i \setminus \{k\}} \delta_{ij} G_{kj}}_{=:e_{ik}}, & k \in \mathcal{S}_i, \\[2ex] \underbrace{\sum\limits_{j \in \mathcal{S}_i} \delta_{ij} G_{kj}}_{=:e_{ik}}, & k \notin \mathcal{S}_i. \end{cases} \tag{A.31}$$

Using the coherence bound, we obtain the deterministic perturbation estimate

$$\begin{cases} (G\boldsymbol{\delta}_i)_k \geq \delta_{ik} - \varepsilon(K-1)\delta_{\max}, & k \in \mathcal{S}_i, \\ (G\boldsymbol{\delta}_i)_k \leq \varepsilon K \delta_{\max}, & k \notin \mathcal{S}_i. \end{cases} \tag{A.32}$$

Choose a threshold $\tau$ such that

$$\varepsilon K \delta_{\max} < \tau < \delta_{\min} - \varepsilon(K-1)\delta_{\max}. \tag{A.33}$$

This ensures support separation

$$\begin{cases} (G\boldsymbol{\delta}_i)_k > \tau, & k \in \mathcal{S}_i, \\ (G\boldsymbol{\delta}_i)_k < \tau, & k \notin \mathcal{S}_i. \end{cases} \tag{A.34}$$

Therefore, the coordinate-wise nonlinearity

$$\sigma_\tau(x) := \begin{cases} 0 & x < \tau, \\ x & x \geq \tau \end{cases} \tag{A.35}$$

produces activations $\boldsymbol{z}_i := \sigma_\tau(G\boldsymbol{\delta}_i)$, with $\operatorname{supp}(\boldsymbol{z}_i) = \mathcal{S}_i$. For $k \in \mathcal{S}_i$,

$$z_{ik} = (G\boldsymbol{\delta}_i)_k = \delta_{ik} + e_{ik}. \tag{A.36}$$

Since for any $j \neq k$, $G_{kj} = \boldsymbol{d}_k^\top \tilde{S} \boldsymbol{d}_j$ is distributed approximately as $\mathcal{N}(0, s^2)$ with small variance $s$, it follows that

$$\mathbb{E}[e_{ik}] = \mathbb{E}\left[\sum_{j \in \mathcal{S}_i \setminus \{k\}} \delta_{ij} G_{kj}\right] = \sum_{j \in \mathcal{S}_i \setminus \{k\}} \delta_{ij} \mathbb{E}[G_{kj}] = 0. \tag{A.37}$$

Since $\mathbb{E}[e_{ik}] = 0$ for all $i, k$, the law of large numbers implies that, in probability,

$$D\, \sigma_\tau(D^\top \tilde{S} \boldsymbol{m}_i) = \boldsymbol{m}_i, \quad \forall i. \tag{A.38}$$

Thus, under this parametrization, the SAE recovers the target atom set $D$. $\qquad\square$

## B  REPRESENTATION SHIFTING

In this section, we describe the experimental setup for studying **Representation Shifting**. Knowledge activations are extracted using the CounterFact dataset. Although CounterFact is typically employed for counterfactual knowledge-editing tasks, our objective differs: we analyze only the activations associated with subject entities, excluding counterfactual objects.

Specifically, we randomly sample 128 subject entities, a quantity previously shown to be sufficient (Hu et al., 2025), yielding $128 \times 128 = 16{,}384$ inner-product pairs. For each entity, we extract activations at the position of its final token, empirically identified through causal tracing as the critical position of knowledge extraction in language models (Meng et al., 2022). The activations at this position are therefore referred to as knowledge activations or knowledge representations.

Next, we compute the pairwise angles between these knowledge activations in Euclidean space using the standard Euclidean inner product. The results show a pronounced representation-shifting effect, an observation that we further confirm across a broad range of models:

- GPT2-Small (Figure 9), GPT2-Medium, GPT2-Large (Figure 10), GPT-J-6B (Figure 11);
- Pythia-1B (Figure 12), Pythia-1.4B, Pythia-2.8B (Figure 13), Pythia-6.9B (Figure 14);
- Llama2-7B (Figure 15), Llama2-13B (Figure 16), Llama3-8B (Figure 17), Llama3.1-8B (Figure 18);
- Gemma2-2B (Figure 19), Gemma2-9B (Figure 20).

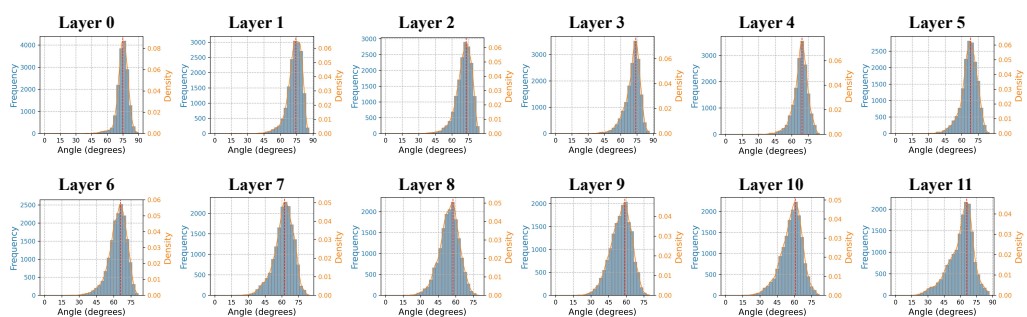

Figure 9: Representation shifting of GPT2-Small.

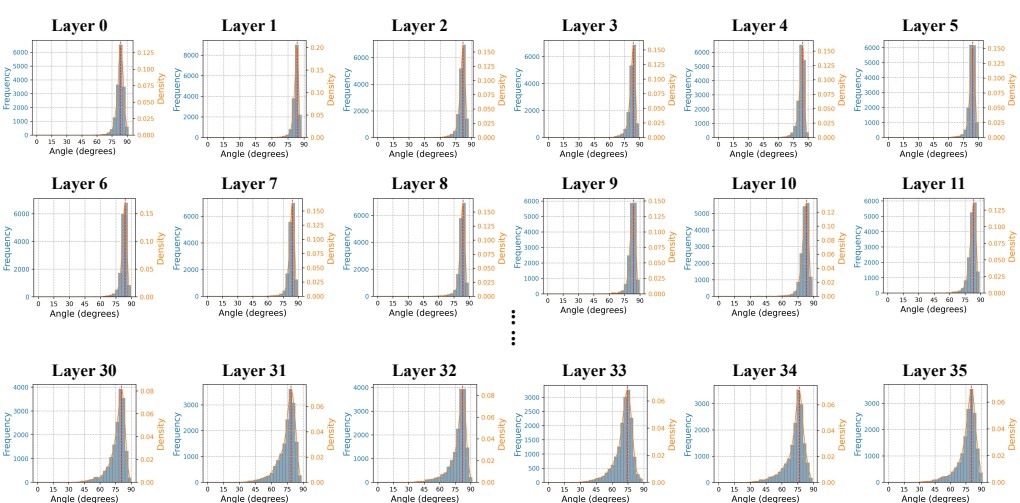

Figure 10: Representation shifting of GPT2-Large.

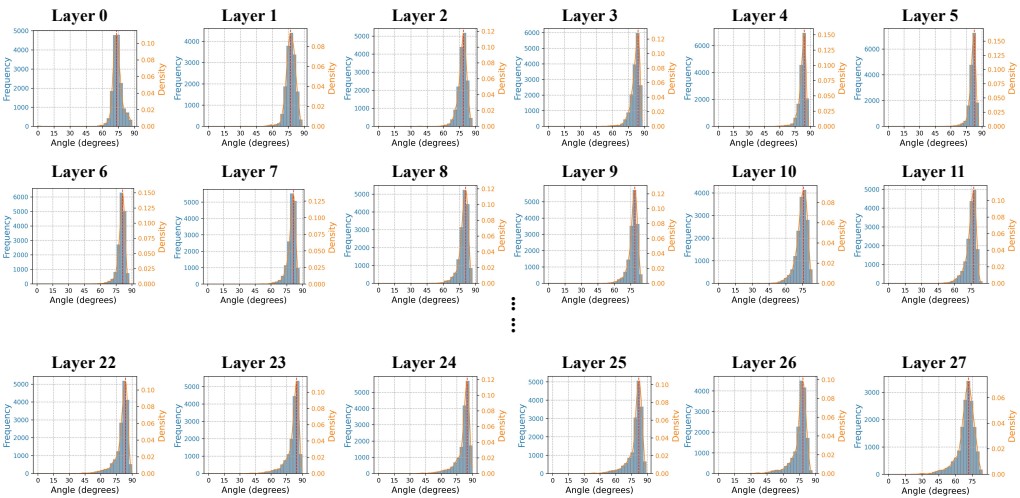

Figure 11: Representation shifting of GPT-J-6B.

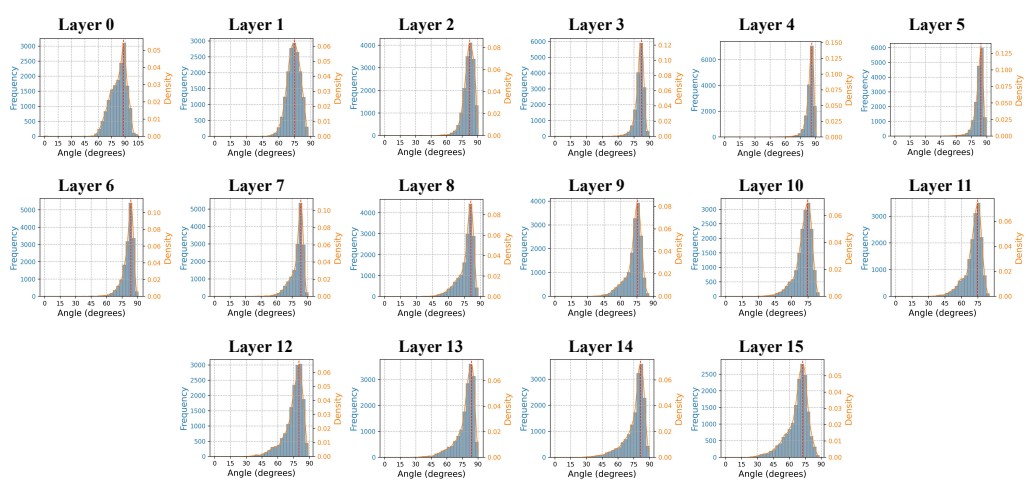

Figure 12: Representation shifting of Pythia-1B.

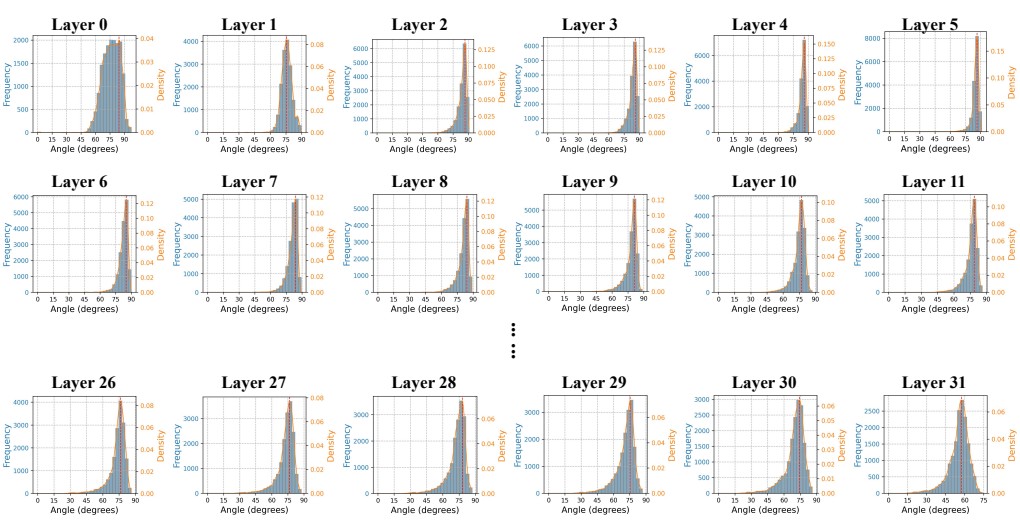

Figure 13: Representation shifting of Pythia-2.8B.

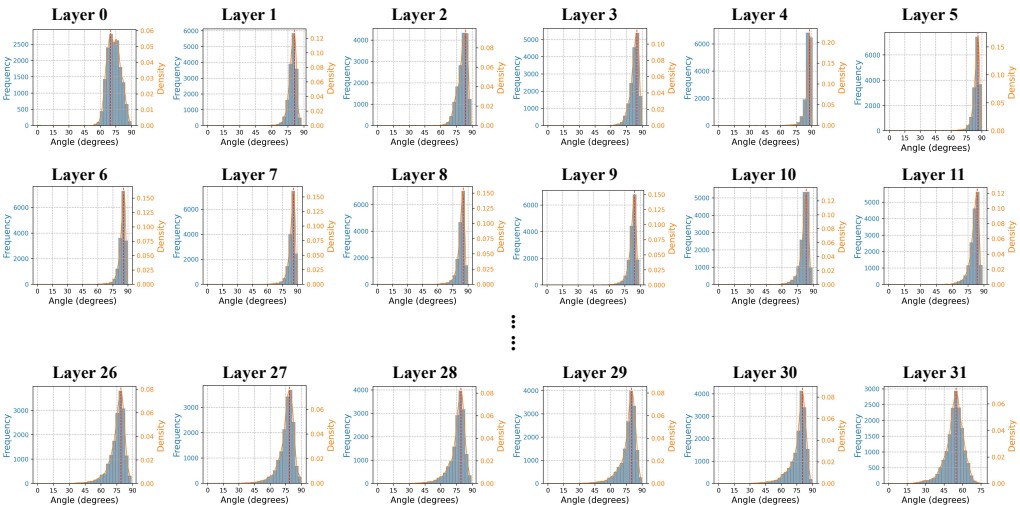

Figure 14: Representation shifting of Pythia-6.9B.

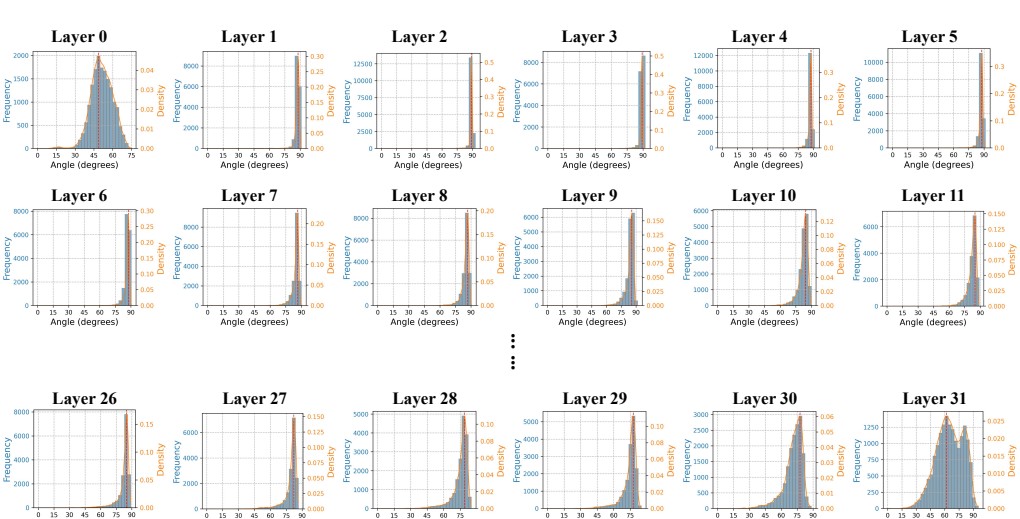

Figure 15: Representation shifting of Llama2-7B.

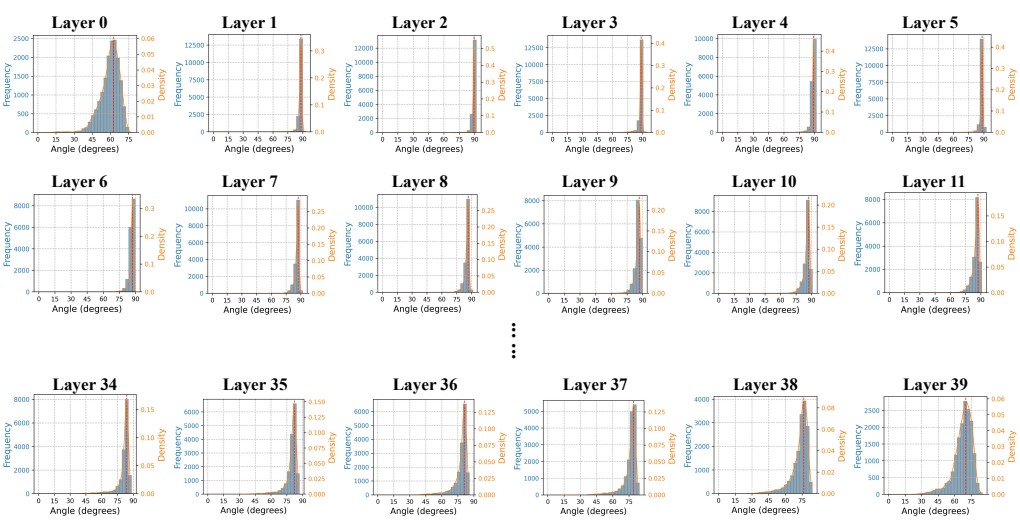

Figure 16: Representation shifting of Llama2-13B.

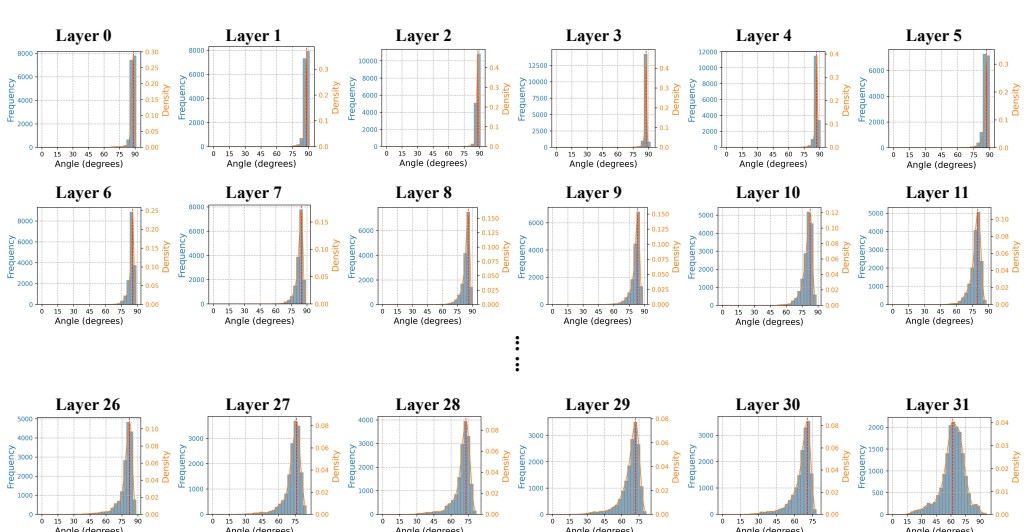

Figure 17: Representation shifting of Llama3-8B.

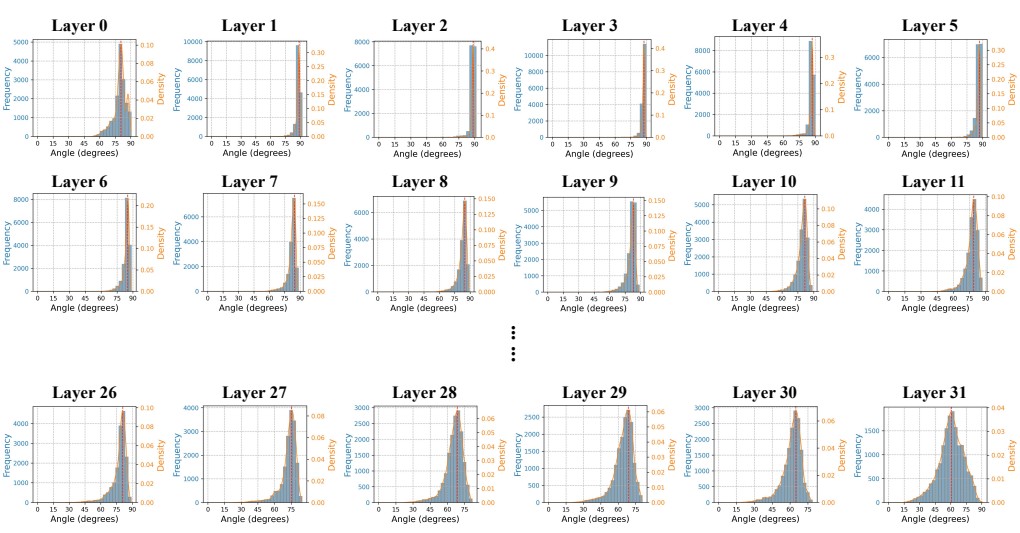

Figure 18: Representation shifting of Llama3.1-8B.

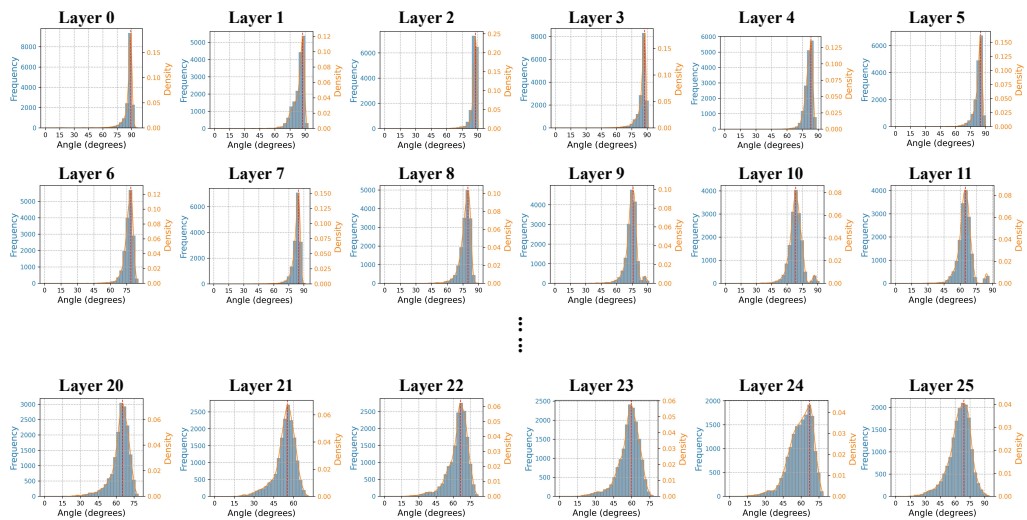

Figure 19: Representation shifting of Gemma2-2B.

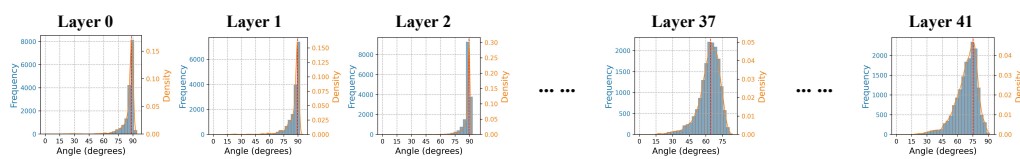

Figure 20: Representation shifting of Gemma2-9B.

This effect is observed across knowledge activations in all layers, indicating that representation shifting is pervasive in language models. To address it, we estimate an appropriate inner product as prescribed by Theorem 2 and Corollary 3 using 100,000 Wikipedia activation samples (Meng et al., 2022). Recomputing the angle distribution yields a centroid concentrated around $90°$, which accords closely with the theoretical expectation. This phenomenon is observed across all layers of all examined models (Figure 21 - 32), providing strong evidence for Atoms Theory: the activation spaces of these language models exhibit a well-defined geometric structure, with inner-product distributions centered near $90°$, indicating strong separation among activations. Moreover, some models exhibit partial orthogonality under the Euclidean inner product in certain layers, particularly earlier ones, likely because the dot-product operations of the attention mechanism promote Euclidean orthogonality. Nevertheless, our method consistently unifies representations across all layers, revealing a coherent geometric structure.

Although we identify the correct inner product for knowledge activations and confirm that their overall angle distribution aligns with expectations, substantial superposition (Hu et al., 2025) per-

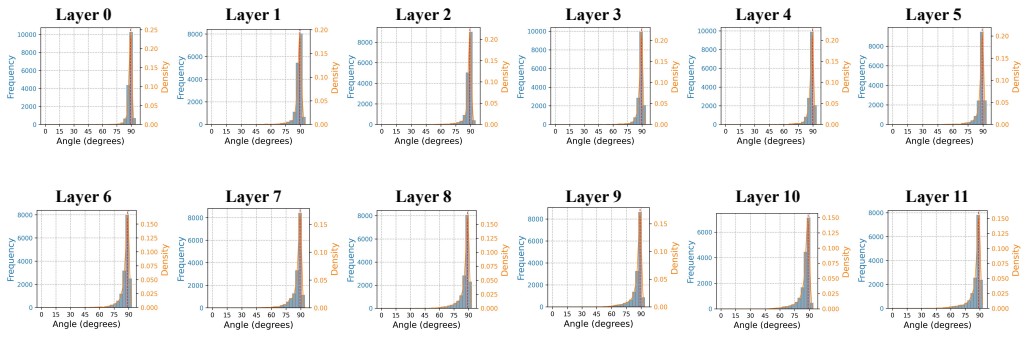

Figure 21: Correcting representation shifting on GPT2-Small.

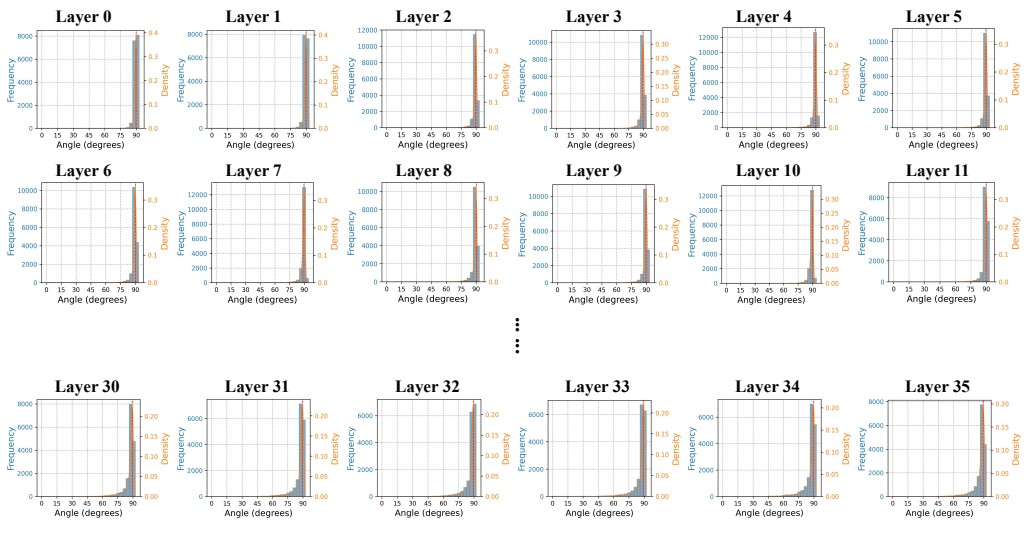

Figure 22: Correcting representation shifting on GPT2-Large.

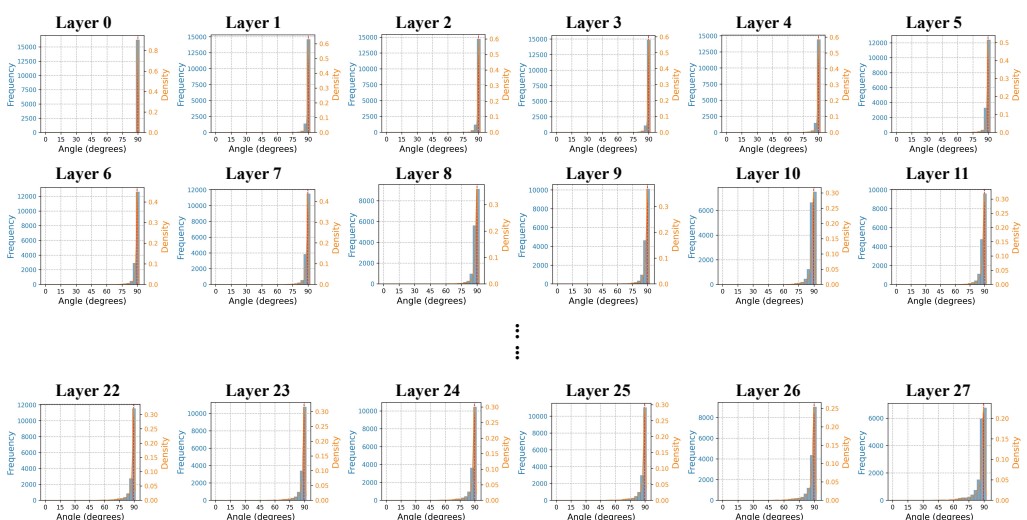

Figure 23: Correcting representation shifting on GPT-J-6B.

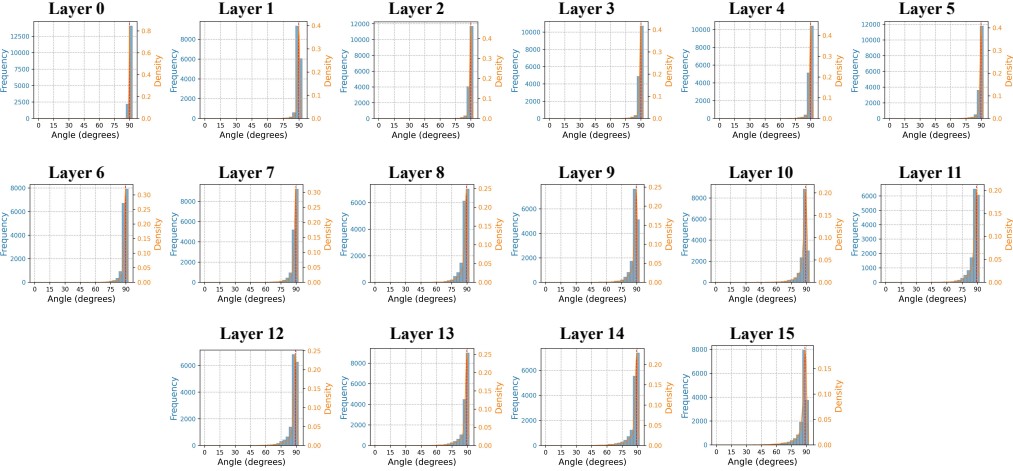

Figure 24: Correcting representation shifting on Pythia-1B.

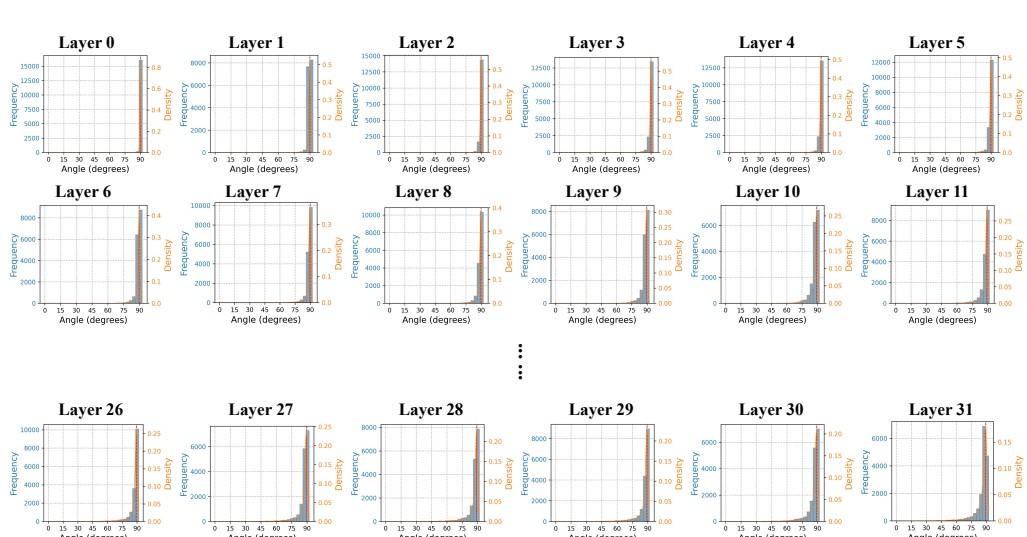

Figure 25: Correcting representation shifting on Pythia-2.8B.

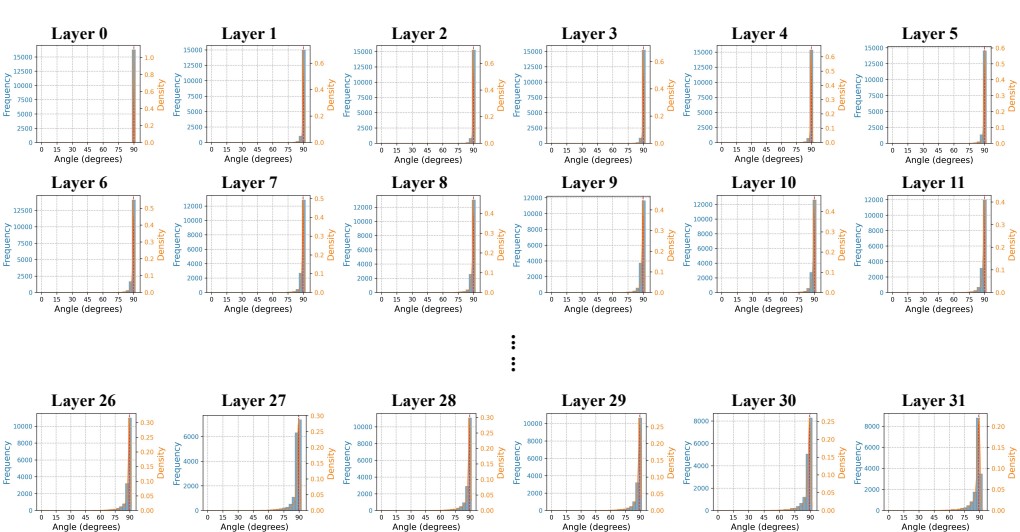

Figure 26: Correcting representation shifting on Pythia-6.9B.

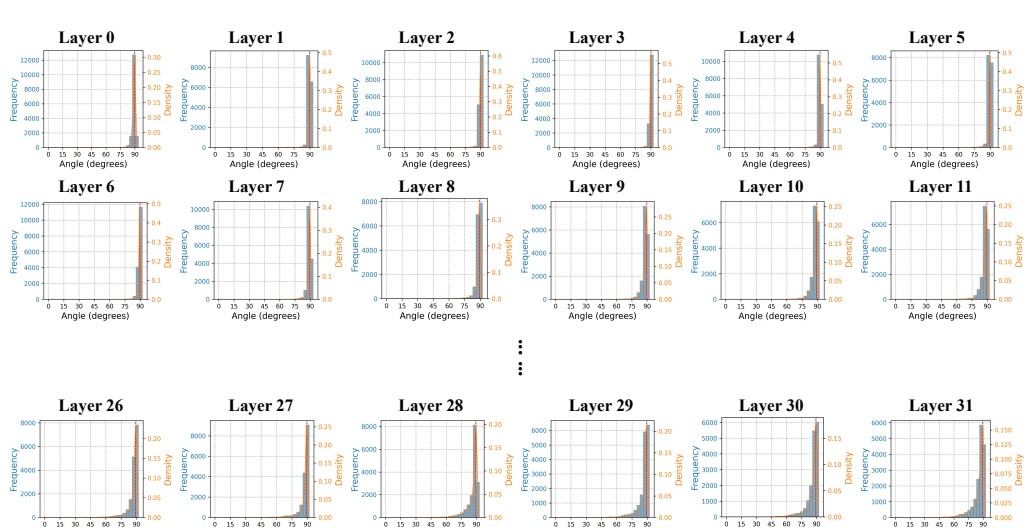

Figure 27: Correcting representation shifting on Llama2-7B.

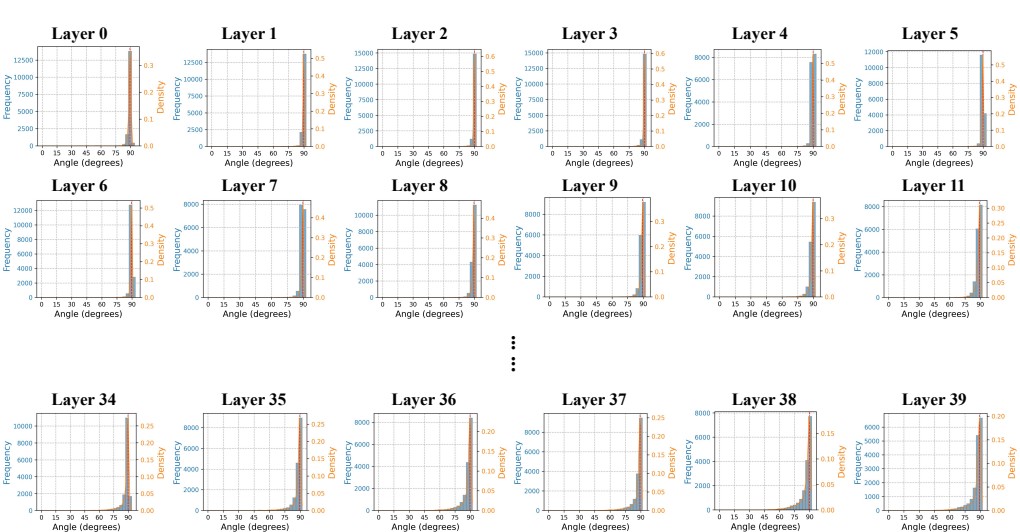

Figure 28: Correcting representation shifting on Llama2-13B.

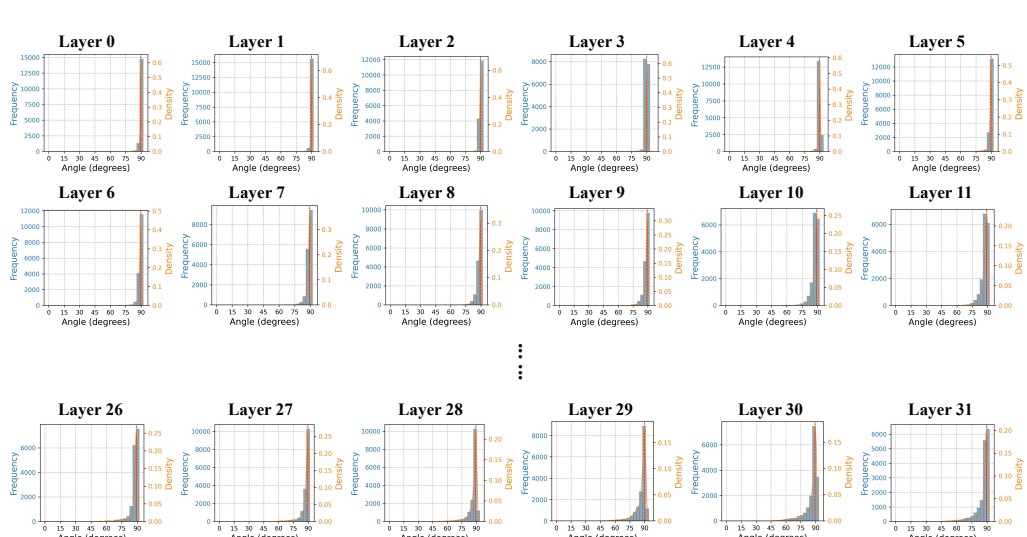

Figure 29: Correcting representation shifting on Llama3-8B.

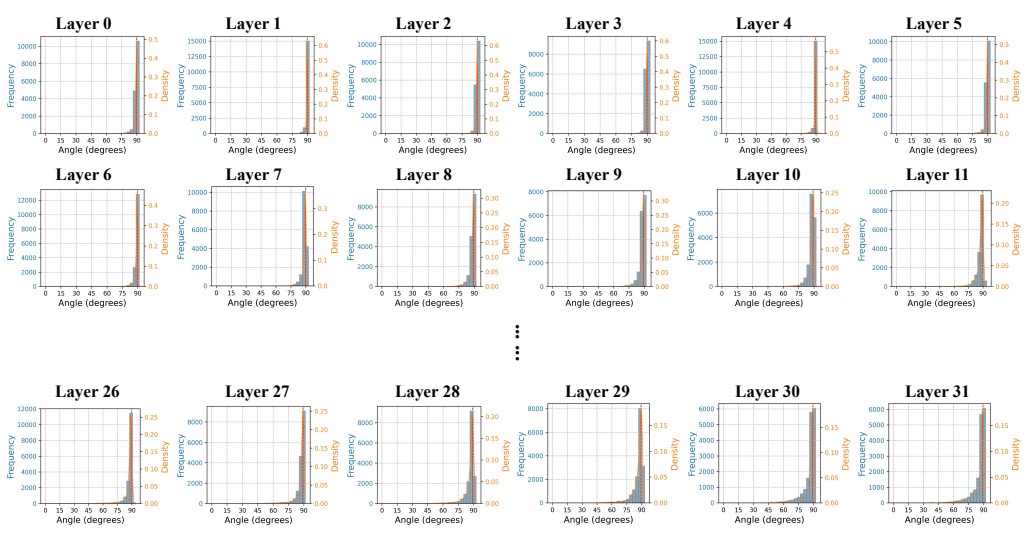

Figure 30: Correcting representation shifting on Llama3.1-8B.

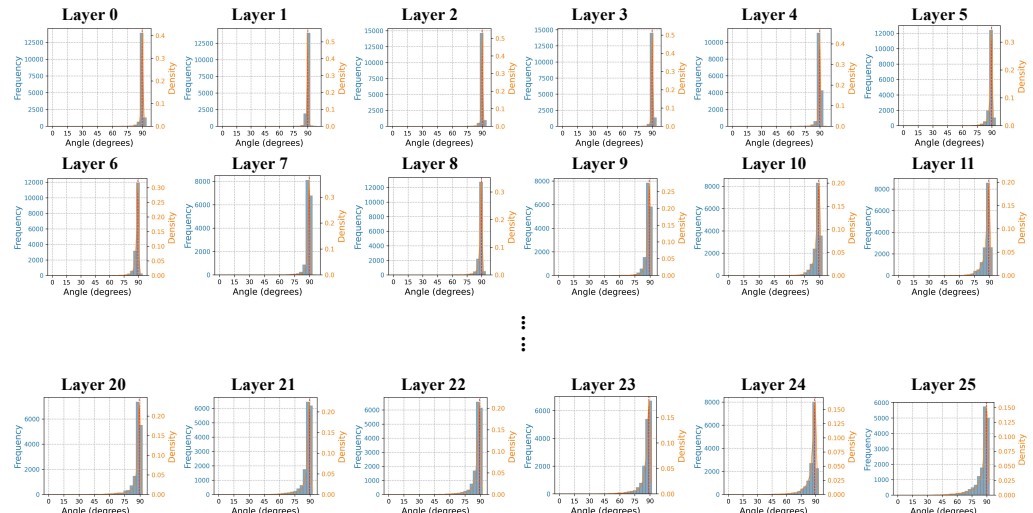

Figure 31: Correcting representation shifting on Gemma2-2B.

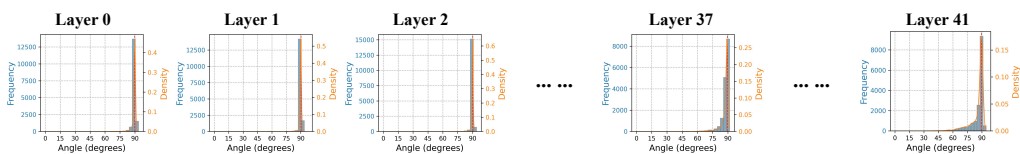

Figure 32: Correcting representation shifting on Gemma2-9B.

sists. Figure 33, shown for Gemma2-2B as an example, demonstrates that knowledge activations still exhibit widespread superposition, indicating that the representations are not fully disentangled.

The persistence of superposition indicates that knowledge activations are not the most appropriate choice as the fundamental unit in language models. We therefore propose Atoms Theory, which decomposes high-dimensional representations into atoms that more faithfully satisfy the criteria for fundamental units. The resulting decompositionshown in Figure 34, demonstrates that these atoms effectively resolve the superposition observed in knowledge activations.

## C    EXPERIMENTAL DETAILS

### C.1    KNOWLEDGE ATOMIZATION TRAINING PARADIGM

Unlike the common practice of fitting SAEs to full activations from natural corpora, we train them on a knowledge-atomization task, using activations elicited by entity knowledge. For each subject entity, we collect the corresponding activations at every layer to form the training set. This design offers three key advantages:

1. Higher effective rank. Entity-induced activations span a broader range of dimensions, providing richer information for learning, as shown in Figure 35;

2. Scalability. The entity set can be readily expanded to match different model and dataset sizes, enabling systematic study of the scale–recoverability relationship;

3. Sparsity. This setup aligns with the prior of Atoms Theory that entity knowledge is largely formed by combining a small number of "atoms," satisfying the sparsity assumption.

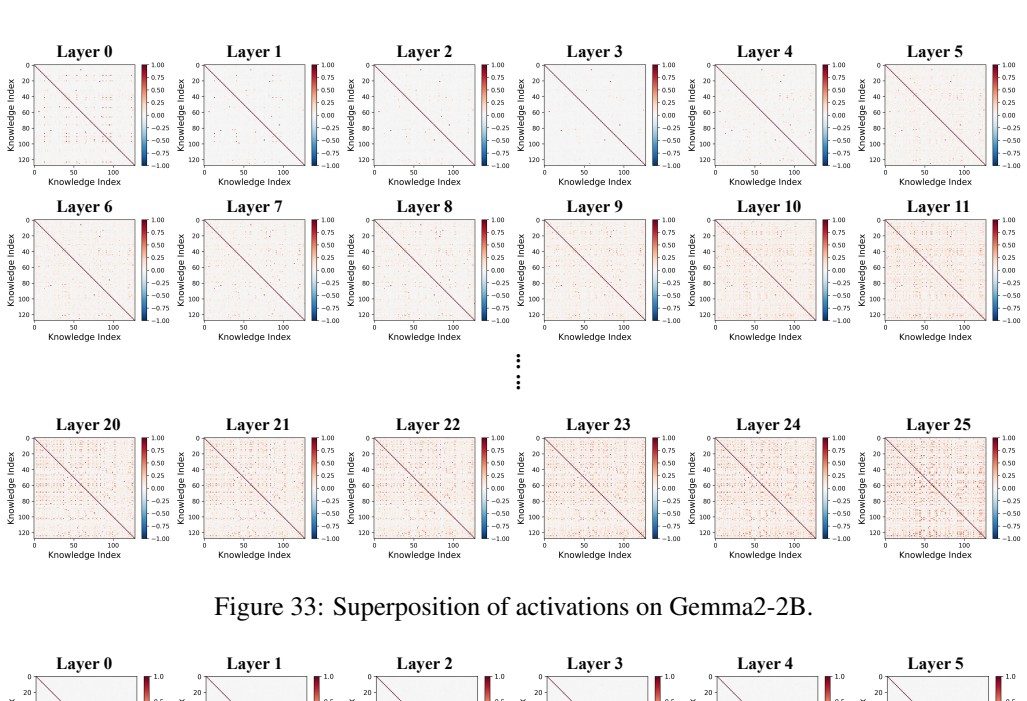

Figure 33: Superposition of activations on Gemma2-2B.

Figure 34: Solving superposition on Gemma2-2B.

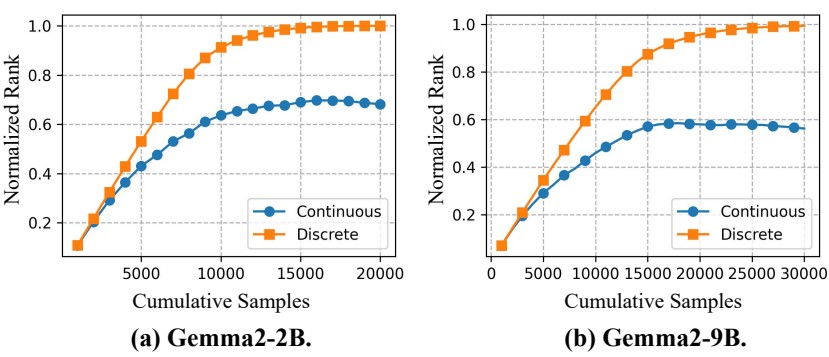

**(a) Gemma2-2B.**    **(b) Gemma2-9B.**

Figure 35: Cumulative normalized rank of (a) Gemma2-2B and (b) Gemma2-9B. Each data point corresponds to the ratio between the rank of the accumulated activation matrix (formed by stacking samples up to that point) and the total dimensionality (i.e., the theoretical maximum rank). Here we illustrate this for randomly selected early layers of Gemma2-2B and Gemma2-9B.

## C.2    DATA COLLECTION

The training data consist of activation representations collected from each layer under the subject prompts of all entities in Counterfact (Meng et al., 2022) and WikiData (Vrandečić & Krötzsch, 2014).

Specifically, we collect activations of subject entities from every layer of Gemma2-2B, Gemma2-9B, and Llama3.1-8B using nearly all WikiData subjects (CounterFact is itself a WikiData subset), including examples such as "Danielle Darrieux", "Edwin of Northumbria", and "Toko Yasuda". In the CounterFact setting this yields 530,166 activations for Gemma2-2B (26 layers × 20,391 entities), 856,422 for Gemma2-9B (42 layers × 20,391 entities), and 652,512 for Llama3.1-8B (32 layers × 20,391 entities). For the scaling experiment we obtain up to 73,728 activations from the first layer of Gemma2-2B using a larger WikiData subject set. Activations are gathered in a uniform manner: each subject name is used as a prompt to the model, and hooks record the activations at the last token of the subject mention, a site previously identified as critical for knowledge representation (Meng et al., 2022). The collected activations are then aggregated as static training data.

## C.3    TRAINING DETAILS

Unless otherwise noted, we employ no special training techniques and do not perform hyperparameter grid search, in order to assess reproducibility and robustness under relaxed settings.

The key hyperparameters are the sparsity coefficient $\lambda$ in the loss function 4.1 and the threshold initialization. We fix $\lambda = 0.1$ (later shown to be insensitive) and set the threshold initialization to 0.001 (or 0.0001), which offers a good balance between training speed and effectiveness: smaller initial thresholds make it easier to locate a value that satisfies the support-separation condition but lengthen training, while 0.001 provides a reliable default in our experiments. Moreover, the straight-through estimator (Rajamanoharan et al., 2024b) is used to approximate gradients at the non-differentiable threshold points during training. Practically, this can be viewed as a differentiable surrogate for $\ell_0$ support selection, while the $\ell_1$ regularization term encourages sparsity in the coefficient magnitudes.

During training, we select the checkpoint on the Pareto front that optimally balances reconstruction and sparsity losses as the final model, and Figure 36 illustrates the Pareto front for Gemma2-2B.

It is noteworthy that the reconstruction performance is largely insensitive to hyperparameters, as training with $\lambda \in \{0.01, 0.1, 1\}$ yields nearly identical learning curves (Figure 37), demonstrating robustness to the sparsity coefficient. This indicates that high-fidelity reconstruction reflects the inherent sparsifiability of the representations rather than an artifact of meticulous tuning.

A minor training issue was observed in layers 30 and 31 of Llama 3.1-8B, where unusually large activations caused optimization to fail; consequently, these layers are omitted from the reported results. This behavior is likely related to their proximity to the output, where activations may drive next-token prediction rather than encode entity-specific information. By contrast, Gemma 2-2B and Gemma 2-9B did not exhibit this problem, possibly because their extensive use of RMSNorm mitigates such activation outliers.

## C.4    BASELINE DETAILS

The baselines used in this paper include **GemmaScope**, comprising SAEs of widths 16k and 65k trained on the MLP layers of Gemma2-2B and SAEs of widths 16k and 131k trained on the MLP layers of Gemma2-9B, and **LlamaScope**, which provides SAEs with 8× and 32× expansion factors trained on the MLP layers of Llama3.1-8B. Both GemmaScope and LlamaScope are generally regarded as open-source tools for extracting features.

It is important to note that, although these models are trained on activations obtained from continuous text corpora, our use of them as baselines is not intended to show that our SAEs outperform GemmaScope or LlamaScope. Rather, the purpose is to highlight that feature-based reconstructions of raw activations remain unreliable, whereas our experiments demonstrate that entity-specific knowledge activations can indeed be reconstructed with high fidelity.

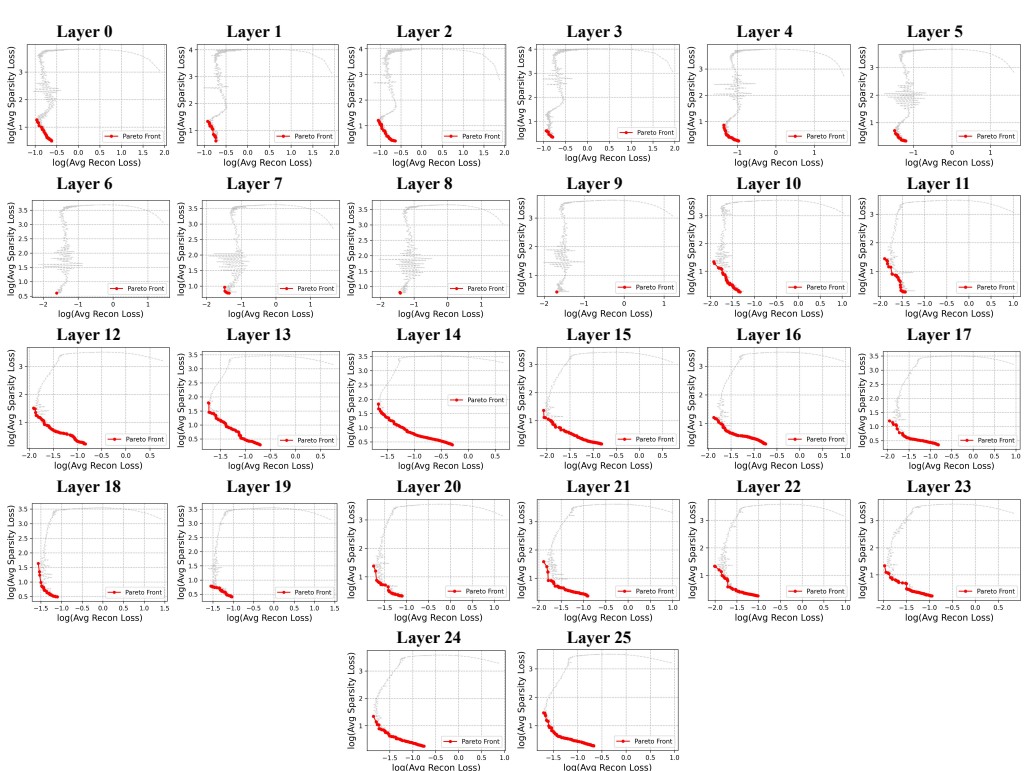

Figure 36: Pareto front during training on Gemma2-2B.

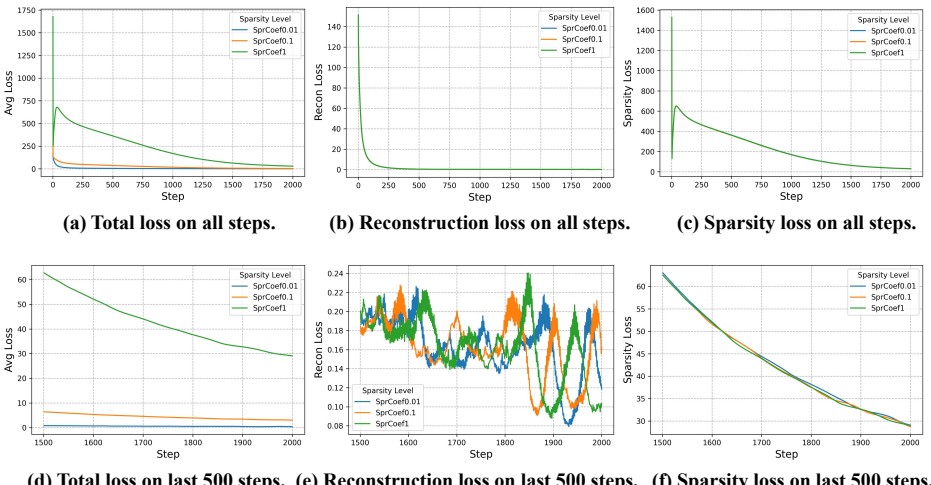

(a) Total loss on all steps.  (b) Reconstruction loss on all steps.  (c) Sparsity loss on all steps.

(d) Total loss on last 500 steps.  (e) Reconstruction loss on last 500 steps.  (f) Sparsity loss on last 500 steps.

Figure 37: Training loss is robust to hyperparameter selection on $\lambda$, maintaining stable performance across different configurations.

## C.5 Evaluation Details

To ensure robustness in real activations, especially in the presence of outliers or bad values, we introduce quantile statistics to correspond to the prior conditions in Theorem 9. Specifically, we define two important statistics:

- **Quantile sparsity** $K_q$: The quantile of sparsity $K_q$ is defined as

$$K_q = \inf \{k \in \mathbb{N} : \mathbb{P}_{\boldsymbol{\delta} \sim \mathcal{P}_\Delta} (K \leq k) \geq q\}, \tag{C.1}$$

where $\boldsymbol{\delta}$ is a coefficient vector sampled from the distribution $\mathcal{P}_\Delta$, and the random variable $K := \|\boldsymbol{\delta}\|_0$ represents the sparsity of the sampled coefficient vector. In simple terms, the quantile sparsity $K_q$ indicates that at least $q$ of the samples have sparsity no greater than $K_q$.

- **Quantile coherence** $\mu_q$: Similarly, the quantile of coherence $\mu_q$ is defined as

$$\mu_q = \inf \left\{\mu \geq 0 : \mathbb{P}_{(\mathcal{I},\mathcal{J})} \left(C \leq \mu | \tilde{D}\right) \geq q\right\}, \tag{C.2}$$

where $(\mathcal{I}, \mathcal{J})$ is uniformly sampled from all unordered pairs of indices $(\mathcal{I} \neq \mathcal{J})$, and the random variable $C := |\langle \tilde{\boldsymbol{d}}_\mathcal{I}, \tilde{\boldsymbol{d}}_\mathcal{J}\rangle|$ represents the coherence between two atoms. In simple terms, this means that the probability of randomly selecting a pair of different atoms with coherence no greater than $\mu_q$ is at least $q$.

Based on these definitions, for the supports of most samples, if the condition $\mu_q < \frac{1}{2K_q-1}$ holds, we can conclude that at least $q$ proportion of the samples satisfy the sufficient conditions for uniqueness and recoverability.

To determine the maximal quantile $q$ satisfying the theoretical criterion, we perform a binary search over the interval $[0, 0.999999]$ for the quantile parameter $\alpha$. At each iteration we compute the linear quantiles

$$\mu_\alpha := \text{Quantile}(\{\mu\}, \alpha), \quad K_\alpha := \text{Quantile}(\{K\}, \alpha), \tag{C.3}$$

and test whether $\mu_\alpha < \frac{1}{2K_\alpha-1}$ holds. If the condition is satisfied, the lower bound of the search interval is updated to $\alpha$; otherwise the upper bound is reduced. Upon convergence, the maximal $\alpha$ obtained is taken as the desired quantile $q$, together with the corresponding values of $\mu_\alpha$ and $K_\alpha$.

Note that verifying Theorem 9 requires the equality $\tilde{D}\boldsymbol{x} = \tilde{\boldsymbol{m}}$. However, as shown in Figure 4, features generally fail to achieve reliable reconstruction, so the quantile $q$ obtained from the condition $\mu_q < \frac{1}{2K_q-1}$ serves only as an ideal upper bound. In contrast, the learned atoms satisfy reliable reconstruction, and over 99.5% of atoms meet $\mu_q < \frac{1}{2K_q-1}$, confirming their favorable properties.

The experimental results are shown in Figure 7. For further detail, Table 1 reports the corresponding values of $q$, $\mu_q$, and $K_q$ for atoms of Gemma2-2B as an illustrative example. As shown in Figure 3, and Figure 21 - 32, the coherence among knowledge activations often reaches 0.9–1.0, whereas the coherence between learned atoms remains below 0.05, reinforcing that activations contain more fundamental, intrinsically sparse fundamental units.

## C.6 Case Studies

Before attempting a direct semantic interpretation of individual atoms, it is useful to establish a rigorous mathematical foundation.

The core of Atoms Theory is the atomic inner product (AIP), a weighted inner product defined by

$$\langle \boldsymbol{x}, \boldsymbol{y}\rangle_{\tilde{S}} := \boldsymbol{x}^\top \tilde{S} \boldsymbol{y}, \quad \tilde{S} = (DD^\top)^{-1}. \tag{C.4}$$

Therefore, we hope to use AIP as a basis to understand atoms without imposing too many artificial priors. Then we define the map

$$\phi : \mathbb{R}^n \to \ell^2, \quad \phi(\boldsymbol{x}) := D^\top \tilde{S} \boldsymbol{x} = \left(\langle \boldsymbol{d}_1, \boldsymbol{x}\rangle_{\tilde{S}}, \langle \boldsymbol{d}_2, \boldsymbol{x}\rangle_{\tilde{S}}, \ldots\right) \in \ell^2, \tag{C.5}$$

Table 1: Quantile statistics of atoms across layers on Gemma2-2B.

| Layer | $q$ | $\mu_q$ | $K_q$ |
|---|---|---|---|
| 0 | 0.997398 | 0.03447 | 14.947 |
| 1 | 0.997750 | 0.03469 | 15.000 |
| 2 | 0.997796 | 0.03461 | 15.000 |
| 3 | 0.998810 | 0.05250 | 10.000 |
| 4 | 0.999149 | 0.05886 | 9.000 |
| 5 | 0.998326 | 0.04020 | 13.000 |
| 6 | 0.998333 | 0.03712 | 14.000 |
| 7 | 0.999398 | 0.06650 | 8.000 |
| 8 | 0.997638 | 0.03455 | 15.000 |
| 9 | 0.998654 | 0.04366 | 12.000 |
| 10 | 0.998269 | 0.03698 | 14.000 |
| 11 | 0.997890 | 0.03442 | 15.000 |
| 12 | 0.991136 | 0.02438 | 21.000 |
| 13 | 0.995978 | 0.02705 | 19.000 |
| 14 | 0.999289 | 0.05252 | 10.000 |
| 15 | 0.998883 | 0.04248 | 12.231 |
| 16 | 0.997157 | 0.03225 | 16.000 |
| 17 | 0.999079 | 0.04717 | 11.000 |
| 18 | 0.994497 | 0.02701 | 19.000 |
| 19 | 0.998019 | 0.03706 | 14.000 |
| 20 | 0.998890 | 0.04749 | 11.000 |
| 21 | 0.996427 | 0.03032 | 17.000 |
| 22 | 0.992200 | 0.02563 | 20.000 |
| 23 | 0.997886 | 0.03691 | 14.000 |
| 24 | 0.998158 | 0.03993 | 13.000 |
| 25 | 0.994317 | 0.02860 | 18.000 |

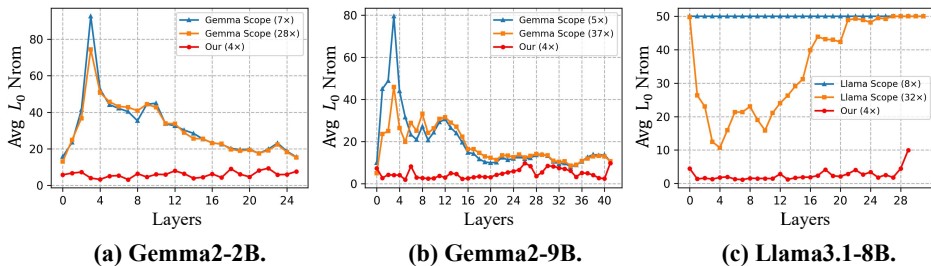

Figure 38: Average $\ell_0$ norm after sparse reconstruction cross models.

where $\ell^2$ is the Hilbert space of square–summable sequences, ensuring the completeness required by kernel methods. Intuitively, $\phi(\boldsymbol{x})$ represents the coordinates of $\boldsymbol{x}$ in the "atoms" basis. This leads to the kernel

$$k(\boldsymbol{x}, \boldsymbol{y}) = \langle \phi(\boldsymbol{x}), \phi(\boldsymbol{y}) \rangle_{\ell^2} \tag{C.6}$$

$$= \langle D^\top \tilde{S} \boldsymbol{x}, D^\top \tilde{S} \boldsymbol{y} \rangle_{\ell^2} \tag{C.7}$$

$$= (\tilde{S}\boldsymbol{x})^\top D D^\top (\tilde{S}\boldsymbol{y}) \tag{C.8}$$

$$= \boldsymbol{x}^\top \tilde{S} \boldsymbol{y} = \langle \boldsymbol{x}, \boldsymbol{y} \rangle_{\tilde{S}}. \tag{C.9}$$

Thus taking the Euclidean inner product after mapping through $\phi$ is exactly equivalent to applying the AIP in the original space.

This kernel perspective is not meant to assert that any individual atom directly corresponds to a human–interpretable concept. Rather, each atom appears as a coordinate of the map $(\langle \boldsymbol{d}_1, \boldsymbol{x} \rangle_{\tilde{S}}, \langle \boldsymbol{d}_2, \boldsymbol{x} \rangle_{\tilde{S}}, \ldots)$, so examining the activation of the $i$-th atom amounts to studying the $i$-th coordinate of $\phi(\boldsymbol{x})$. That is, we can quantify the match between an atom and a given input $\boldsymbol{x}$ through the AIP $\langle \boldsymbol{d}_i, \boldsymbol{x} \rangle_{\tilde{S}}$ and further examine the shape of the induced function $k(\cdot, \boldsymbol{d}_i)$ over the input space, thereby providing a more systematic characterization of the concept region represented by the atom.

This provides the mathematical foundation for post-hoc interpretability methods. However, Atoms Theory itself only identifies the concept region associated with each atom; interpreting that region inevitably involves human judgment to assess potential semantic coherence. Accordingly, in this work we present concept regions for atoms solely to demonstrate the consistent properties of atoms, without offering further subjective interpretation.

For example, we present the atoms activated by "Mac OS" (Table 2 - 7) and "Beijing" (Table 8-13) in layers 1–6 of Gemma2-2B and examine, for each layer, all entities that activate these atoms, thereby delineating the concept regions associated with each atom.

### C.7 SUPPLEMENTARY EXPERIMENTS

We present here supplementary experimental results that could not be included in the main text owing to space limitations.

Figure 38 reports the average $L_0$ norm of sparse reconstruction (i.e., the average realized sparsity of each entity activation) complementing Figure 4.

Figures 39 and 40 present the spontaneous alignment results for Gemma2-9B and Llama3.1-8B in the same format as Figure 5 for Gemma2-2B.

Figures 41, 42 and 43 provide the complete NAIP distributions for Gemma2-2B, Gemma2-9B, and Llama3.1-8B, extending the results of Figure 6.

Figure 44 reports the complete sparsity statistics for the scaling experiments corresponding to Figure 8.

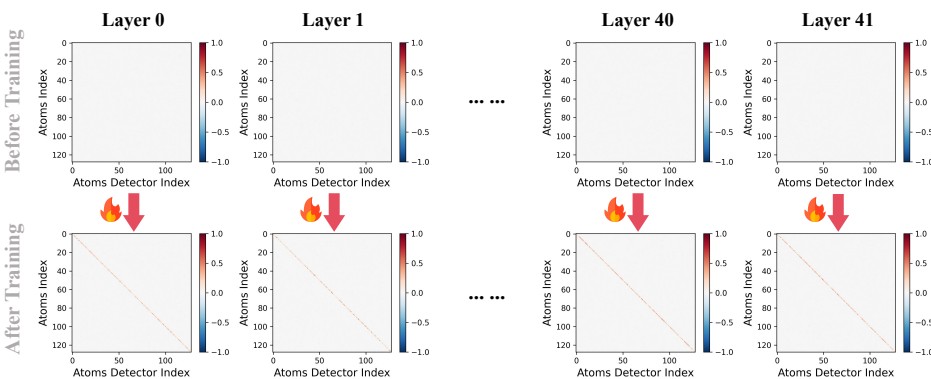

Figure 39: Spontaneous alignment between the encoder and decoder during training on Gemma2-9B.

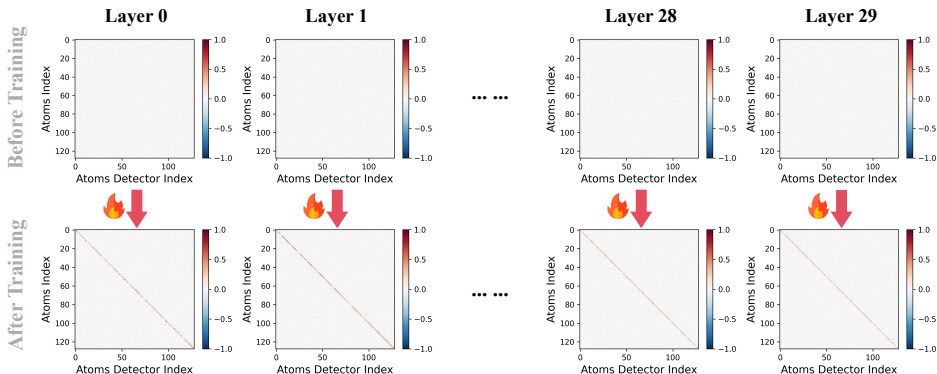

Figure 40: Spontaneous alignment between the encoder and decoder during training on Llama3.1-8B.

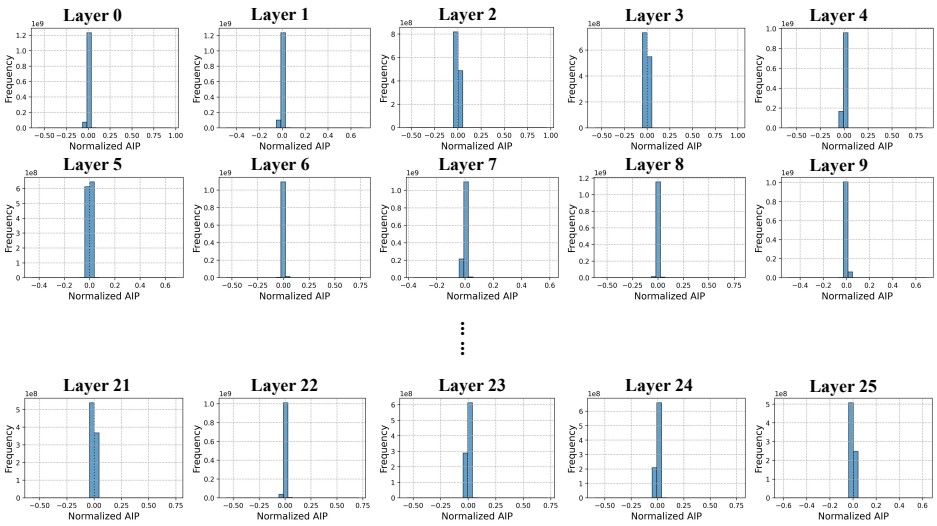

Figure 41: NAIP distribution of atoms across all layers of the Gemma2-2B.

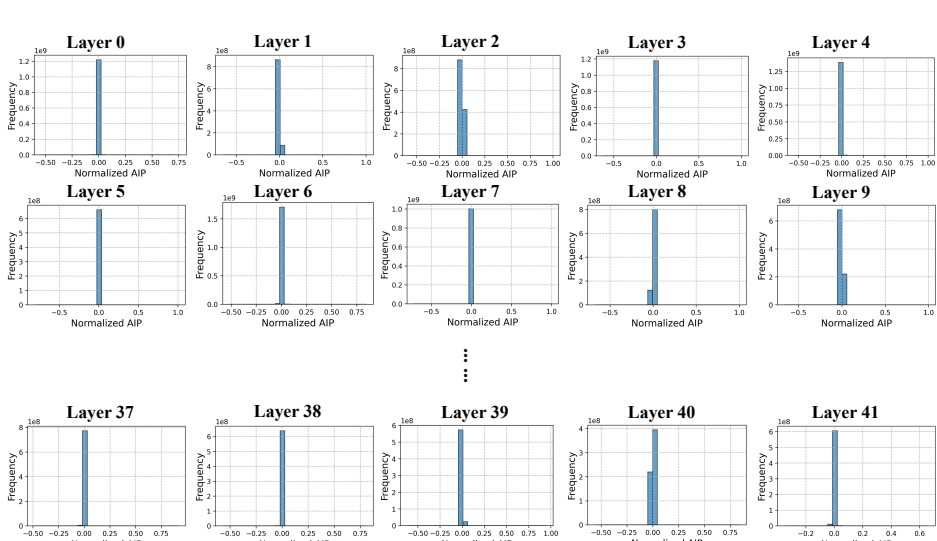

Figure 42: NAIP distribution of atoms across all layers of the Gemma2-9B.

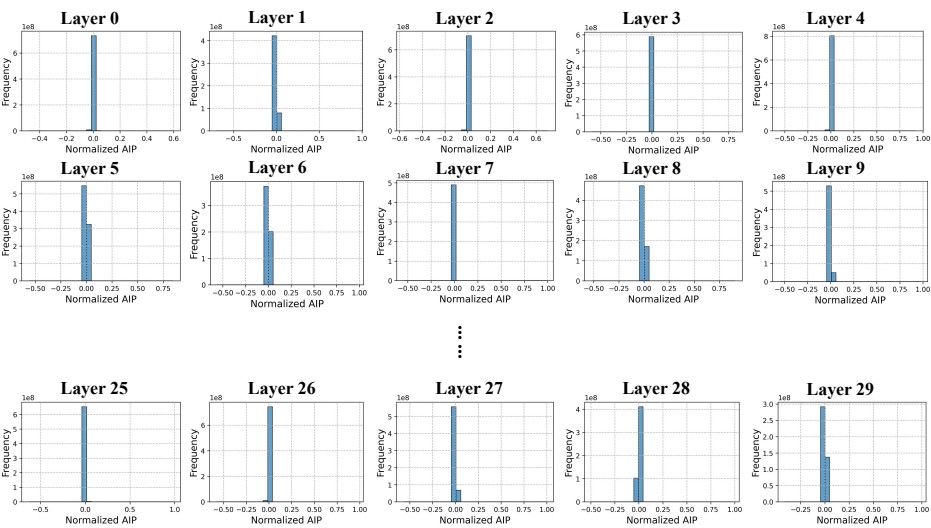

Figure 43: NAIP distribution of atoms across all layers of the Llama3.1-8B.

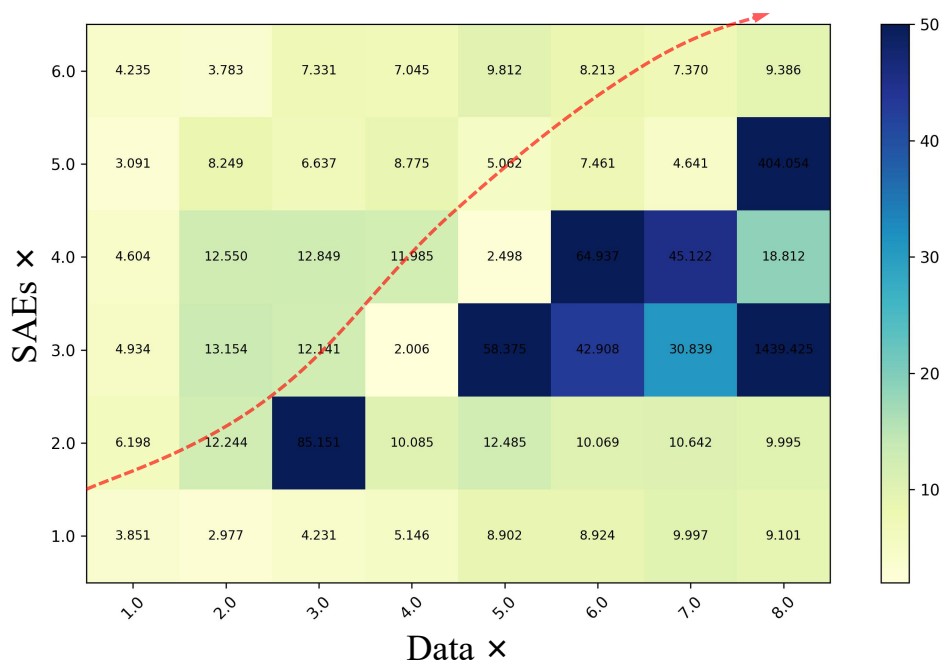

Figure 44: Average $\ell_0$ norm of scaling experiments on Gemma2-2B.

Table 2: Entities grouped by atoms ID for *Mac OS* on layer 1 of Gemma2-2B.

| Atoms ID | Entities |
|---|---|
| 10352 | Mac OS X Lion, Mac OS X Panther, Mac OS, Mac OS X Leopard, MacBook Air, Windows Media Encoder, Mac OS X Tiger |
| 11039 | OS X Mavericks, Ike Ekweremadu, Rhea Chakraborty, Mac OS, Apple Watch, Lucerne, Elf Aquitaine, Microsoft Entourage |
| 11259 | Mac OS X Lion, Indore, Mac OS |
| 13741 | Johnny Yune, Apple II, Tarnobrzeg Voivodeship, Salentin IX of Isenburg-Grenzau, Mac OS |
| 16379 | Chrome OS, IBM 4690 OS, IBM Workplace OS, Mac OS |
| 16618 | Serpent Column, Track Record, Gilera, Mac OS, Jean Reno, Gary Burton, Chevrolet Captiva |
| 20855 | Singapore Bus Service, Jeff Fager, Bruce County, Logan Verrett, Mac OS, Anders Fager |
| 21712 | Aceh, Facit, Mac OS, Cas Haley, Chelmsford 123 |
| 22737 | John Treacy, Mac OS, Alejandro Bustillo, Test Drive Le Mans, Simon Louvish |
| 26809 | Chrome OS, Windows Virtual PC, MacApp, iOS, macOS, Mac OS |
| 34234 | Edmund Neupert, CNN Heroes, Mac OS, George Frideric Handel, OS X Yosemite |

Table 3: Entities grouped by atoms ID for *Mac OS* on layer 2 of Gemma2-2B.

| Atoms ID | Entities |
|---|---|
| 1974 | Chrome OS, Mac OS, Mona Lisa, Wear OS |
| 2321 | Partners HealthCare, Mac OS, Johnny Smith, Nicole Oresme |
| 4532 | Saladin, Varkaus, Mac OS, Always Greener, Wear OS |
| 4879 | The Truce, Nazi Germany, Windows XP Media Center Edition, Mac OS |
| 5978 | Mac OS, Mahmud Hussain, Brigitte Bardot |
| 6218 | Tom Atkins, Orivesi, El Diario de Hoy, Michael Jackson, Mac OS |
| 8984 | Zeev Rechter, Mac OS, Dmitry Puchkov, Walter Gay |
| 14156 | Chrome OS, IBM 4690 OS, Gyllene Tider, IBM Workplace OS, Mac OS, Carl Orff, Orfeu, Alexander Wittek |
| 14208 | Bengkulu, Bola Sete, Mac OS, Watch My Chops, Julius Erving, Astaldi, John Landy |

Table 4: Entities grouped by atoms ID for *Mac OS* on layer 3 of Gemma2-2B.

| Atoms ID | Entities |
|---|---|
| 5542 | Mac OS, Hartwall |
| 6708 | Bourg-la-Reine, Mac OS, Malabo |
| 10489 | The Mentalist, Mac OS, Pretzel, Boris Karloff |
| 19938 | Mac OS, Altera Enigma, Robert Riefling |
| 27693 | macOS, Mac OS, Bertold Hummel |

Table 5: Entities grouped by atoms ID for *Mac OS* on layer 4 of Gemma2-2B.

| Atoms ID | Entities |
|---|---|
| 17076 | Symbian, iOS, Windows 10, macOS, IBM PC DOS, Mac OS, Apple DOS, Windows 1.0, Atari DOS, MSX-DOS, MS-DOS |
| 25709 | Brief Encounter, Mac OS, Le comte Ory |
| 28634 | Chrome OS, IBM 4690 OS, Windows Phone 8.1, Windows NT, IBM Workplace OS, Windows 2000, macOS, Mac OS, IBM AIX, Newton OS |
| 29286 | SFJAZZ Collective, Mac OS |

Table 6: Entities grouped by atoms ID for *Mac OS* on layer 5 of Gemma2-2B.

| Atoms ID | Entities |
|---|---|
| 1377 | Mac OS |
| 10852 | Chrome OS, Windows NT, Mac OS, Apple Remote Desktop, Google Chrome, Chromecast, Chromebook, Wear OS |
| 10900 | Mac OS, Toyota Vios |
| 22511 | Manila Light Rail Transit System, Wii U system software, Windows XP Media Center Edition, UNIX System Services, Mac OS, OS X Mountain Lion |
| 23957 | Lin Huiyin, Mac OS |
| 24148 | Mac OS 8, Mac OS 9, Mac OS, Golden Axe, Super Monaco GP, Gitarzan, System 7 |
| 28986 | Logic Pro, Adobe Encore, Final Cut Pro, Adobe Soundbooth, Mac OS, Android TV, Adobe Premiere Pro, Final Cut Pro X |
| 33604 | Like Father, Like Daughter, Mac OS, Supporters Range |

Table 7: Entities grouped by atoms ID for *Mac OS* on layer 6 of Gemma2-2B.

| Atoms ID | Entities |
|---|---|
| 2847 | Naan Potta Savaal, Mac OS, Rajakokila, Solva Saal |
| 8265 | European Union, Clarke Stadium, Mac OS |
| 18850 | M5 motorway, Mac OS, Doublemoon |
| 19901 | BBC One, Nintendo DS, Xbox 360, Mac OS |
| 25337 | Mac OS 8, IBM 4690 OS, IBM Workplace OS, Mac OS 9, macOS, Mac OS, Xenix, IBM AIX, XNU, Newton OS, Microsoft Windows, System 7 |
| 27739 | Windows NT, Greenpeace, Shigeru Miyamoto, Mac OS |

Table 8: Entities grouped by atoms ID for *Beijing* on layer 1 of Gemma2-2B.

| Atoms ID | Entities |
|---|---|
| 15264 | Beijing, Seoul, 1 Maccabees, Ulysses Dove |
| 15982 | Beijing, Siikainen, 36 China Town, Jim Allchin |
| 23987 | Beijing, Swann Memorial Fountain, Charles Chilton, Otto Neurath |
| 31322 | Shanghai, Beijing |
| 35951 | Beijing, Russia, Arkansas, Paris |
| 36035 | Beijing, Meiert Avis, Aviation Industry Corporation of China |

Table 9: Entities grouped by atoms ID for *Beijing* on layer 2 of Gemma2-2B.

| Atoms ID | Entities |
|---|---|
| 620 | Shanghai, Beijing, Hanoi, Tokyo, Adam Maida |
| 6258 | Beijing, Majorca, Thailand, Greg Dyke |
| 7540 | 1300 Oslo, Beijing, Miami Horror, Lille |
| 10761 | Moscow, Beijing, Canberra, Pyongyang |
| 11519 | Karl Polanyi, Beijing, Cevdet Sunay, Mary Gaunt, Cyd Hayman, Les diamants de la couronne |
| 13418 | Beijing, Tarnobrzeg Voivodeship, Yakuza, Longs Peak, Jeep Wrangler |
| 15585 | Beijing, Ivan Koloff, Olinto Cristina |
| 22622 | Shanghai, Cleveland, Beijing, Delhi, Saint Lucia, St Lucia, Venice |
| 26002 | Beijing, Alte Oper, Intimate Stories, Seventeen, Five Star Krishna |
| 27116 | Ankara, Mandarin Oriental, Bangkok, Cairo, Beijing, Dublin, Jakarta, Amsterdam, Bratislava, Toronto, Sydney, Edinburgh, London, Honolulu, Auckland, Bali, Tokyo, Manila, Queens Gardens, Brisbane, Budapest, Montreal, Perth, Kolkata, Dubai, Melbourne, Copenhagen, Nairobi, Bangkok, Bangalore |

Table 10: Entities grouped by atoms ID for *Beijing* on layer 3 of Gemma2-2B.

| Atoms ID | Entities |
|---|---|
| 9444 | Shanghai, Beijing |
| 24724 | Moscow, Beijing, Russia |
| 30463 | Beijing, Thailand |
| 32854 | Beijing, Madrid, Mariano Gonzalvo |

Table 11: Entities grouped by atoms ID for *Beijing* on layer 4 of Gemma2-2B.

| Atoms ID | Entities |
|---|---|
| 1578 | Beijing, Cadbury |
| 11098 | Beijing, Jakarta |
| 11158 | Beijing |
| 15601 | Oslo, Moscow, Stockholm, Berlin, Athens, Helsinki, Beijing, Vienna, Geneva, Amsterdam, Seoul, Prague, Madrid, London, Warsaw, Kyoto, Naples, Tokyo, Budapest, Paris, Rome, Bangkok |
| 25755 | Stockholm, Helsinki, Beijing, Minneapolis, Minecraft, Copenhagen, Nairobi |
| 33322 | Shanghai, Beijing, Guangzhou, Macau, Hong Kong, Chongqing, Shenzhen, Wuhan |

Table 12: Entities grouped by atoms ID for *Beijing* on layer 5 of Gemma2-2B.

| Atoms ID | Entities |
|---|---|
| 11453 | Beijing, The Great Citizen |
| 12661 | Beijing, Holycross-Ballycahill GAA |
| 19018 | Beijing, Registro, 4th of August Regime, Witnesses |
| 23750 | Moscow, Ankara, Beijing, Jakarta, Madrid |

Table 13: Entities grouped by atoms ID for *Beijing* on layer 6 of Gemma2-2B.

| Atoms ID | Entities |
|---|---|
| 7533 | Johannesburg, Shanghai, Beijing, Colombo, Prafulla Chandra Ghosh |
| 16414 | Shenyang, Shanghai, Beijing, Guangzhou, Yangtze, Google China, Taobao, Tianjin, Chongqing, National Development and Reform Commission, Shenzhen, Qing dynasty, Aviation Industry Corporation of China, Qzone, Youku, Wuhan, People's Republic of China |
| 22386 | Beijing |
| 33958 | Carol Zhao, Shenyang, Shanghai, Beijing, Guangzhou, Seoul, Yangtze, Macau, Hanoi, Taipei, Hong Kong, Kaohsiung, South Korea, Busan, United States Army Military Government in Korea, Tianjin, Pyongyang, Incheon, Chongqing, Vietnam, Dennis Hwang, Shenzhen, Daejeon, North Korea, Wuhan |

# D  USAGE OF LARGE LANGUAGE MODELS

Large language models (LLMs) are employed solely as auxiliary tools during the preparation of this manuscript. Specifically, we use LLM-based services to assist with language refinement and grammar checking of text drafted by the authors. The conceptual development of the research, the design and execution of all experiments, the analysis of results, and the formulation of conclusions are performed entirely by the authors without automated content generation. All scientific claims, theoretical arguments, and experimental findings presented in this paper are the sole responsibility of the authors.

