# OpenReview forum: "Towards Atoms of Large Language Models"
_ICLR.cc/2026/Conference — Submitted to ICLR 2026_

### Official Review · Reviewer_x57o · 2025-10-26

**Soundness:** 2
**Presentation:** 3
**Contribution:** 2
**Rating:** 4
**Confidence:** 2

**Summary:**

This paper proposes the Atom Theory, a theoretical framework defines atoms as the fundamental units of LLMs. Authors present atomic inner product as a replacement of Euclidean inner product in examining orthogonality to correct representation shifting. They prove the conditions under which atoms satisfy the Restricted Isometry Property to ensure the uniqueness and recoverability of sparse representations. They also prove threshold-activation SAEs can effectively achieve near perfect sparse reconstruction, and scale the method on up to 9B language models.

**Strengths:**

1. The paper is well-written and the theoretical framework and proofs are rigorous. Intuition behind the theoretical statements are clear.
2. The Atom Theory gives rigorous definition of fundamental units of LLMs and various properties they should fulfill, supplementing previous SAE studies with solid theoretical groundings.

**Weaknesses:**

1. The explicit form of the Atomic Inner Product in Theorem 2 $d_i^TSd_j=c^2d_i^T(DD^T)d_j$ is a solution to the linear least square problem $\min_S||c^4I-D^TSD||$, and will naturally produce near zero values in off-diagonal elements. Thus, the significance of the results in Figure 3 (correcting representation shifting) is doubtful.
2. The idea of SAEs with threshold activations (i.e. JumpReLU SAEs) is not novel and proposed by Rajamanoharan et al. before. The results of high sparsity and near perfect reconstructions contradict the previous studies.

**Questions:**

1. (As in Weaknesses 1) Can random vectors and ReLU SAE features serve as atoms that fulfill the Atomic Inner Product or the $\epsilon$-Approximately Orthogonal Atoms?
2. (As in Weaknesses 2) Can the authors explain the difference of performance between the original JumpReLU paper and this work? Does the difference come from different activation collecting method (natural text corpus versus knowledge-atomization tasks)?

---

> ### Author Response · Authors · 2025-11-15
> **Rebuttal by Authors (Response to "W1")**
>
> We are grateful to you for your careful reading of our paper and for the constructive and insightful feedback.
>
> #### **Response to "W1"**
>
> We would like to clarify that **the activations used to compute representation shifting / correct representation shifting are *disjoint* from those used to find the target inner-product space $\langle \cdot, \cdot \rangle_S$**. The former correspond to **the entity-specific knowledge activations we are interested in**, whereas the latter are obtained by **randomly sampling 100,000 unrelated activations from Wikipedia** and computing the statistic $\mathbb{E}[k k^\top]$.
>
> In other words, **we first estimate the target inner-product space using activations sampled from unrelated corpus data, and only afterward test whether this inner-product space can correct the representation shifting of the knowledge activations we are interested in**—and indeed, it does.
>
> Furthermore, this result **suggests that the activation density across the language model activation space is not uniform**. Some regions contain highly concentrated activations, while others are significantly dispersed. This is an intriguing observation, and prior work has hinted at such geometric landscapes \[1\]\[2\]\[3\].
>
>
>
> [1] The Linear Representation Hypothesis and the Geometry of Large Language Models.
>
> [2] The Geometry of Categorical and Hierarchical Concepts in Large Language Models.
>
> [3] Knowledge in Superposition: Unveiling the Failures of Lifelong Knowledge Editing for Large Language Models.

---

> ### Author Response · Authors · 2025-11-15
> **Rebuttal by Authors (Response to "Q1" - Part 1)**
>
> #### **Response to "Q1"**
>
> We define the **atoms** through **three essential conditions**.
>
> - First, it must satisfy $\boldsymbol{m}_i = D \boldsymbol{\delta}_i$, meaning that the original representation $\boldsymbol{m}_i$ can be **accurately reconstructed** using the matrix $D = [\boldsymbol{d}_1, \boldsymbol{d}_2, \cdots, \boldsymbol{d}_D]$ formed by atoms set and the coefficient vector $\boldsymbol{\delta}_i$.
> - Second, it must satisfy $\|\boldsymbol{\delta}_i\|_0 \le K$, ensuring that **the representation is sparse**.
> - Finally, it must hold that $\forall i\neq j,\ |\langle \tilde{\boldsymbol{d}}_i, \tilde{\boldsymbol{d}}_j \rangle| \le \epsilon$, i.e., the **normalized atoms are approximately orthogonal**.
>
> **How do we evaluate the quality of the obtained atoms?** In **Theorem 9**, we derive a key metric $\mu < \frac{1}{2K - 1}$. This metric determines whether the reconstruction using these atoms is **unique**, and therefore reflects **the stability of the learned atoms**. Importantly, Theorem 9 **unifies the latter two conditions in the atoms definition—sparsity and approximate orthogonality—into a single quantitative measure**, enabling principled evaluation of atomic quality.
>
> Thus, **in practice, we assess whether representations satisfies the atoms definition and exhibits desirable properties using two criteria**:
>
> 1. **Accurate reconstruction of the original representation**.
> 2. A high proportion of representations satisfying $\mu < \frac{1}{2K - 1}$, indicating **uniqueness of the reconstruction** and therefore **stability**.
>
> This corresponds directly to **Fig. 4** and **Fig. 7**. **Fig. 4** evaluates reconstruction accuracy; our atoms achieve an average $R^2$ of **99.90%**. **Fig. 7** evaluates the proportion of representations satisfying the stability condition; our atoms reach an average of **99.86%**.

---

> ### Author Response · Authors · 2025-11-15
> **Rebuttal by Authors (Response to "Q1" - Part 2)**
>
> **Can random vectors or ReLU SAE features serve as atoms?**
>
> 1. **Random vectors do *not* satisfy the atom definition**，because they **violate the sparsity requirement**.
>
>    For a target representation $\pmb{m}$, a randomly sampled high-dimensional vector is almost orthogonal to $\pmb{m}$. Therefore, reconstructing $\pmb{m}$ would require an extremely large number of random vectors, violating sparsity.
>
>    **We verify this experimentally**. For each target activation, we iteratively project out components along random high-dimensional vectors until the activation is fully expressed, and we record the required number of random vectors. We compare this with the number of atoms required to represent the same activation.
>
>    Due to space limitations, **we show 8 different target activations, each represented by random vectors and the specific atoms we found.**
>
>    |                  | case 1 | case 2 | case 3 | case 4 | case 5 | case 6 | case 7 | case 8 |
>    | ---------------- | ------ | ------ | ------ | ------ | ------ | ------ | ------ | ------ |
>    | Random Vectors   | 127077 | 127944 | 127521 | 127264 | 127853 | 127216 | 127552 | 128248 |
>    | **Atoms (Ours)** | **10** | **7**  | **4**  | **10** | **4**  | **8**  | **6**  | **9**  |
>
>    It is evident that **random vectors fail the sparsity requirement**. Furthermore, the fraction satisfying $\mu < \frac{1}{2K - 1}$ is effectively **zero**.
>
> 2. **ReLU SAE features do *not* satisfy the atom definition**, because they **violate the accurate reconstruction requirement**.
>
>    For **theoretical perspective**, ReLU activations induce an amortization gap [1]. Since atoms are only approximately orthogonal, a non-zero threshold is necessary to filter out noise arising from approximate orthogonality. ReLU uses a fixed threshold of zero and therefore **cannot filter this noise**, preventing accurate identification of atoms—formalized in **Theorem 10** in a certain sense. Thus, **ReLU SAEs cannot achieve accurate reconstruction in principle**.
>
>    For **empirical perspective**, prior works has shown that **ReLU SAEs fail to achieve accurate reconstruction** \[2\]\[3\]\[4\]\[5\]. TopK SAEs \[3\] were introduced to enforce sparsity during training, while JumpReLU SAEs \[4\]\[5\] were introduced to address **feature shrinkage** [6]. Both families of methods claim (and third-party evidence confirms \[7\]) **Pareto improvements over ReLU**. However, **none of ReLU, TopK, or JumpReLU achieve accurate reconstruction** in existing studies.
>
>    We would also like to emphasize here that prior works often grouped all unreconstructed components under the term “dark matter” [8]. **Our work provides a more precise explanation**: For **ReLU**, the inability to reconstruct accurately is due to **structural limitations** (Theorem 10; line 286); for **TopK** and **JumpReLU** (both specific instances of threshold activation; line 300), the issue arises because **the SAE scale does not match the data scale** (Fig. 8; line 439), preventing full decompression.
>
>
>
> [1] Compute Optimal Inference and Provable Amortisation Gap in Sparse Autoencoders.
>
> [2] Scaling Monosemanticity: Extracting Interpretable Features from Claude 3 Sonnet.
>
> [3] Scaling and Evaluating Sparse Autoencoders.
>
> [4] Improving Dictionary Learning with Gated Sparse Autoencoders.
>
> [5] Jumping Ahead: Improving Reconstruction Fidelity with JumpReLU Sparse Autoencoders.
>
> [6] Addressing Feature Suppression in SAEs.
>
> [7] https://transformer-circuits.pub/2024/june-update/index.html
>
> [8] Decomposing the Dark Matter of Sparse Autoencoders.

---

> ### Author Response · Authors · 2025-11-15
> **Rebuttal by Authors (Response to "W2&Q2")**
>
> #### **Response to "W2&Q2"**
>
> **Relationship Between Thresholded Activations and JumpReLU**：The **threshold activation** is a broad conceptual category that may include many concrete implementations. **JumpReLU can be viewed as one specific instance** of such a thresholded activation. Similarly, **TopK can also be interpreted as a form of thresholded activation**, though its practical usefulness is limited by the need to predefine the value of $K$ (line 300).
>
> **Our core contribution is to prove—within the framework of Atoms Theory—the identifiability of SAEs with threshold activation**, and to **empirically demonstrate that high-fidelity reconstruction is achievable when the model scale and data scale are properly matched** (Figs. 4 and 8). Moreover, we show that the resulting atoms satisfy both **atomicity** (Fig. 6) and **stability** (Fig. 7).
>
> Importantly, **our findings do not contradict prior work** \[1\]\[2\]\[3\]. On the contrary, they should be viewed as a **theoretical and empirical extension** of these studies. Specifically, we prove identifiability of SAEs (Theorem 10), and demonstrate—via scaling experiments—that **accurate reconstruction is possible only when model size and data size are appropriately paired** (Fig. 8). Thus, **for any given data scale, a properly sized threshold-activation SAE *can* achieve accurate reconstruction**.
>
> **Why Prior Methods Show Suboptimal Reconstruction Performance?** The performance gap arises **not** from differences in activation sampling, but from a **training paradigm inherited from LLMs**:
>
> > *“More training data is always better.”*
>
> Our scaling experiments (Fig. 8) show that this assumption does not hold for SAEs performing **decompression**. Prior work’s training setup corresponds to **the lower-right region of Fig. 8**, where a **too-small SAE** is trained on **too much data**, forcing the SAE into a compressive regime. Our training setup instead corresponds to **the dashed optimal-scaling curve**, which selects the **correct matching** between data scale and model capacity. **This is the key reason the experimental results may appear “in conflict” with earlier studies—though they are not**.
>
> **To further demonstrate that the difference is *not* due to activation-collecting choices, we conduct a controlled experiments comparing three types of representations extracted under the same experimental settings:**
>
> - **General corpus** (sampled from **Wikipedia**);
> - **Complex reasoning** (sampled from **MATH500**);
> - **Knowledge atomization** (sampled from **WikiData**).
>
> We train SAEs on these three categories under identical configurations. The results are as follows:
>
> |                                      | $R^2$  | $L_0$ |
> | ------------------------------------ | ------ | ----- |
> | **General Corpus (Wikipedia)**       | 99.84% | 9.36  |
> | **Complex Reasoning (MATH500)**      | 99.86% | 9.56  |
> | **Knowledge Atomization (WikiData)** | 99.89% | 8.32  |
>
> These results show that **high sparsity and accurate reconstruction arise because we use an appropriately scaled threshold-activation SAE for the given data scale**.
>
> More fundamentally, **the activations themselves possess an inherent sparse structure**, which makes it possible to extract sparse atoms. **This is the core motivation of our work: to identify the fundamental representational units—atoms—within language models**.
>
>
>
> [1] Scaling and Evaluating Sparse Autoencoders.
>
> [2] Improving Dictionary Learning with Gated Sparse Autoencoders.
>
> [3] Jumping Ahead: Improving Reconstruction Fidelity with JumpReLU Sparse Autoencoders.

---

> ### Author Response · Authors · 2025-11-17
> **To Reviewer x57o**
>
> **Dear Reviewer x57o,**
>
> We sincerely thank you again for the time you dedicated to reviewing our submission. Your thoughtful comments have been invaluable in helping us refine the clarity and presentation of our work.
>
> In the responses above, we carefully address each of your comments and provided additional theoretical discussion and empirical evidence where appropriate. We hope that these clarifications satisfactorily resolve your concerns and offer a clearer and more comprehensive understanding of the contributions of our work.
>
> If any aspect of our work would benefit from further clarification, please do not hesitate to let us know; we would be very glad to provide more detailed explanations.
>
> Sincerely,
>
> The Authors

---

> ### Author Response · Authors · 2025-11-24
> **Follow-up on Rebuttal Clarifications (If Needed)**
>
> **Dear Reviewer x57o,**
>
> We hope you are doing well, and we sincerely appreciate the time and effort you have dedicated to reviewing our work.
>
> During the rebuttal phase, we have carefully addressed every concern raised in your reviews, and we would be grateful if you could kindly let us know whether you have any remaining questions or would like further clarification on any part of our response.
>
> If there are specific aspects that you believe require additional detail (e.g., experiments, theoretical clarification, or presentation adjustments), we would be very glad to elaborate.
>
> Thank you again for contributing your expertise and for helping us improve our work.
>
> Kind regards,
>
> The Authors

---

> ### Author Response · Authors · 2025-11-28
> **Follow-up on Rebuttal Clarifications**
>
> **Dear Reviewer x57o,**
>
> Thank you very much for taking the time to review our paper and for the helpful comments you provided earlier. We truly appreciate your efforts.
>
> As the discussion phase is approaching its end, we would like to kindly check whether there are any remaining concerns or questions that we could help clarify. We are more than happy to provide additional details or explanations if needed.
>
> If all your concerns have already been addressed in our rebuttal, we would greatly appreciate it if you could consider increasing your score accordingly.
>
> Best regards,
>
> The Authors

---

> ### Comment · Reviewer_x57o · 2025-11-28
>
> Thanks for your detailed information! I think you've clearly explained in what case your theory works and what experiment settings make your results different. I still suspect the correct matching of data scale and model capacity, since it's not so meaningful to train an SAE that could not generalize to common linguistic tasks. However, I'm willing to increase my final score.

---

> > ### Author Response · Authors · 2025-11-28
> > **Re-clarification of Our Work (Part 1 / 3)**
> >
> > Thank you very much for your reply! We would, however, like to kindly reiterate the core contributions of our paper. Although similar points were mentioned earlier, we believe that restating them here may help clarify and refocus the key contributions of our work.

---

> > ### Author Response · Authors · 2025-11-28
> > **Re-clarification of Our Work (Part 2 / 3)**
> >
> > **We would like to reiterate that the core contributions of our paper are as follows:**
> >
> > - Based on the **atomic inner product**, we **formally and rigorously define—for the first time—the fundamental representational units of LLMs**, which we call ***atoms***. Building on this definition, we derive the **theoretical properties of atoms**, how to **quantitatively validate whether target representations constitutes atoms**, and provide **theoretical guarantees for the identifiability of atoms**. Together, these results form a complete **Atoms Theory**.
> >
> > - To **empirically validate the foundation of the atomic theory**, namely the **atomic inner product**, we conduct experiments across a **wide range of language models**, including:
> >
> >   - GPT2-Small, Medium, Large, GPT-J-6B
> >   - Pythia-1B, 1.4B, 2.8B, 6.9B
> >   - Llama2-7B, 13B, Llama3-8B, Llama3.1-8B
> >   - Gemma2-2B, 9B
> >
> >   Across **all layers** of these models, we validate the atomic inner product through both the **representation shift phenomenon** (**Figure 2; Appendix B**) and the **representation shift correction experiment** (**Figure 3; Appendix B**). This provides strong evidence that the atomic inner product offers a meaningful perspective on internal representations and **establishes a solid empirical foundation for the Atoms Theory**.
> >
> > - **How can we empirically verify whether a target representation is atomic?** According to the **Atoms Theory**, the definition of atoms encompasses three components: **faithful reconstruction** (reflected by $R^2$), **sparsity** (reflected by $K$), and **approximate orthogonality** (reflected by $\mu$). Although all three quantities can be measured, sparsity and approximate orthogonality are inherently qualitative. The **Atoms Theory** shows that they can be jointly characterized by the condition $\mu < \frac{1}{2K - 1}$, which reflects the **stability** of atoms—that is, whether corresponding representations can be uniquely reconstructed. Therefore, **in practice, it suffices to measure faithful reconstruction ($R^2$, Figure 4) and stability ($\mu < \frac{1}{2K - 1}$, Figure 7) to determine whether target units are atomic**, which derived from the **Atoms Theory**.
> >
> >   To the best of our knowledge, this is **the first systematic and rigorous method for verifying whether target units matches the ideal notion of atoms**, whereas prior works mostly relied on labor-intensive and qualitative analyses that lack rigor.
> >
> > - **Empirically, we further examine whether commonly studied units—neurons and features—satisfy requirements of atoms.** Our conclusions are:
> >
> >   - For **neurons**, although they can faithfully reconstruct the original representations, they lack stability (**Figure 7**). Stable neurons account for only $0.45\%$ on average.
> >   - For **features**, we observe suboptimal reconstruction fidelity (**Figure 4**), but better stability than neurons, with stable features accounting for **68.25%** on average, although instability still exists.
> >
> >   To our knowledge, this is **the first systematic comparison between neurons and features—using rigorously derived quantitative criteria—to evaluate their suitability as fundamental interpretability units**.
> >
> > - Further analysis shows that the **fundamental reason for the suboptimal behavior of features is that existing SAEs (e.g., GemmaScope and LlamaScope) are not sufficiently large to identify atoms when trained on massive corpus**. **Section 4.4 (Lines 438–450)** and **Figure 8** demonstrate that **effective atom identification requires the SAE size to exceed a threshold relative to the dataset size**.
> >
> >   Accordingly, we **determine the appropriate size of threshold-activated SAEs (which is theoretically guaranteed to enable valid atoms identification) based on dataset scale**, and retrain SAEs across all layers of **multiple language models (Gemma2-2B, Gemma2-9B, Llama3.1-8B)**. This allows us to **successfully identify atoms in practice** (**Figures 4 and 7**; visualized in **Figures 5 and 6**).
> >
> >   To our knowledge, we are **the first to identify the fundamental reason that prior works obtain only suboptimal SAE results**:
> >
> >   - **ReLU SAEs**: because atoms are approximately orthogonal, identifying them requires thresholding to filter out noise induced by near-orthogonality; ReLU’s fixed threshold at 0 is insufficient for this.
> >   - **TopK SAEs**: these can partially mimic threshold activation but require a fixed $K$ (a hyperparameter), limiting practical applicability.
> >   - **JumpReLU SAEs**: this is the most flexible implementation of threshold activation, yet prior works still obtained suboptimal results due to mismatch between dataset scale and model capacity (**Figure 8**).
> >
> >   In contrast, **we are the first to propose selecting SAE size based on dataset scale, enabling accurate reconstruction and stability, thereby empirically validating the existence of atoms**.

---

> > ### Author Response · Authors · 2025-11-28
> > **Re-clarification of Our Work (Part 3 / 3)**
> >
> > **In summary, the contributions of this work are broad and multifaceted:**
> >
> > 1. We are the first to theoretically formalize the fundamental representational units of LLMs and to establish a complete Atoms Theory.
> > 2. We are the first to reveal the widespread phenomenon of representation shifting in LLMs and demonstrate how to correct it, thereby validating the foundation of the Atoms Theory—namely, the atomic inner product.
> > 3. We are the first to propose, in a systematic and rigorous manner, how to empirically verify whether target units are ideal fundamental representational units—atoms—and to systematically compare neurons and features as the fundamental units, highlighting their differences and limitations.
> > 4. We are the first to identify the root cause behind the suboptimal performance of SAEs in prior work and to propose selecting the appropriate size of threshold-activated SAEs based on dataset scale. This enables accurate reconstruction that satisfies stability requirements, providing practical validation for the existence of atoms.
> >
> > **We would like to clarify that the value of interpretability research is not determined solely by its immediate practical applications. Enhancing conceptual clarity in this field, providing a stronger theoretical foundation, and conducting targeted empirical validations are equally important. Our work aims to advance the theoretical understanding of mechanistic interpretability, and we believe such progress can meaningfully support future practical developments.**
> >
> > We sincerely appreciate your suggestion. While **Figure 8 already demonstrates the scalability of our approach through extensive scaling experiments**, we agree that exploring even larger-scale datasets would be valuable. We plan to pursue this direction in future work, though it is beyond the scope of the present paper.

---

### Official Review · Reviewer_oTLx · 2025-10-30

**Soundness:** 1
**Presentation:** 2
**Contribution:** 1
**Rating:** 2
**Confidence:** 4

**Summary:**

This paper proposes the Atoms Theory as a formal framework for identifying fundamental representational units—termed “atoms”—within large language models (LLMs). The authors introduce the concept of the Atomic Inner Product (AIP) to correct geometric distortions in representation space and prove that under certain conditions, these atoms satisfy the Restricted Isometry Property (RIP), ensuring stable sparse recovery. They further show that single-layer sparse autoencoders (SAEs) with threshold activations can theoretically identify such atoms. Empirical experiments are conducted on Gemma2 and Llama3.1 models to validate reconstruction fidelity and atomicity, claiming that atoms provide a more stable and unique representation than neurons or features.

**Strengths:**

- The theoretical part (Section 3) is systematically developed, with clear connections to compressed sensing and RIP theory.
- The proofs are complete and rigorous, giving the work a solid mathematical foundation.
- The motivation—defining fundamental representational units of LLMs—is important and aligns with ongoing efforts in mechanistic interpretability.

**Weaknesses:**

- The threshold-activated SAE is functionally equivalent to JumpReLU SAE, offering no methodological novelty.
- The reconstruction experiments are conducted on a dataset orders of magnitude smaller than those used in GemmaScope and LlamaScope as baseline  (tens of thousands vs. hundreds of millions).
- The comparison of reconstruction scores is therefore unfair and misleading, as performance differences can be entirely attributed to dataset scale rather than model quality.
- The claimed empirical validation of “Atoms Theory” is thus not credible given the limited and mismatched setup. The work overstates its practical impact despite weak experimental support.

**Questions:**

- How is the proposed threshold-activated SAE fundamentally different from the JumpReLU SAE? If they are equivalent, what justifies reintroducing it under a new name?
- Why were the experiments conducted on such a small dataset (only tens of thousands of activations)? Can the authors demonstrate that their method scales to hundreds of millions or billions of activations like the compared baselines?
- Would the conclusions about reconstruction fidelity and atomicity still hold if the models were trained and evaluated on the same-scale datasets as GemmaScope or LlamaScope?
- Can the authors provide ablation or control experiments to verify that the improvements are not due to scale or preprocessing differences?

---

> ### Author Response · Authors · 2025-11-15
> **Rebuttal by Authors (Response to "W1&Q1")**
>
> We sincerely appreciate your thorough and patient review of this paper, along with your valuable suggestions and comments.
>
> #### **Response to "W1&Q1"**
>
> The motivation for **introducing thresholded activations** is grounded in **Theorem 10**, which shows that by applying an appropriate threshold, **atoms remain identifiable even when they are only approximately—not strictly—orthogonal**. Intuitively, the normalized atomic inner products between distinct atoms are not exactly zero; they satisfy only *approximate* orthogonality. Under such conditions, **non-thresholded activations such as ReLU introduce interference that hinders atom identification** (**as discussed in line 298**). To the best of our knowledge, **this is the first work to provide a rigorous theoretical proof of identifiability of  SAEs with threshold activation**, whereas prior work has only established the *non-identifiability* of ReLU SAEs [1].
>
> Moreover, **this motivation is fundamentally different from the motivation behind JumpReLU**, which was proposed to mitigate *feature shrinkage* \[6\] during SAE training \[2\]\[3\].
>
> **Relationship Between Threshold Activation and JumpReLU**：**The threshold activation is a broad conceptual category encompassing many possible implementations**. **JumpReLU can be regarded as one particular instantiation of a thresholded activation.** Likewise, **TopK can also be viewed as a thresholded activation**, although its practical use is constrained by the need to pre-specify the value of $K$ (**line 300**).
>
> We greatly appreciate the contributions of Rajamanoharan et al. \[1\]\[2\] and agree that **JumpReLU is arguably the most flexible instantiation of a thresholded activation**, which is precisely why we adopt it in our experiments. This flexibility allows the SAE to discover structural properties inherent to the data without imposing strong human priors (such as pre-fixing $K$ in TopK). Indeed, **we explicitly acknowledge and cite this prior work in the paper** (**lines 305 and 344**).
>
> We emphasize that **we do not claim to introduce a new activation function**. Instead, **our core contribution lies in providing the first formal definition of atoms, developing the accompanying Atoms Theory, and proving—both theoretically and empirically—the identifiability of SAEs with thresholded activations**.
>
> Our work should be viewed as a **theoretical complement** to \[2\]\[3\]\[4\]\[5\]. **Future efforts may indeed yield more efficient practical identification methods, but such methods will still rely on, and be justified within, the framework of Atoms Theory**.
>
>
>
> [1] Compute Optimal Inference and Provable Amortisation Gap in Sparse Autoencoders.
>
> [2] Improving Dictionary Learning with Gated Sparse Autoencoders.
>
> [3] Jumping Ahead: Improving Reconstruction Fidelity with JumpReLU Sparse Autoencoders.
>
> [4] Scaling and Evaluating Sparse Autoencoders.
>
> [5] Scaling Monosemanticity: Extracting Interpretable Features from Claude 3 Sonnet.
>
> [6] Addressing Feature Suppression in SAEs.

---

> ### Author Response · Authors · 2025-11-15
> **Rebuttal by Authors (Response to "W2&Q2")**
>
> #### **Response to "W2&Q2"**
>
> Indeed, **in prior work, the prevailing intuition—derived from the success of LLM training that “more data is always better”—was almost uncritically transferred to SAE training**. However, **this overlooks a crucial fact**: **LLMs perform compression, whereas SAEs perform decompression** [1].
>
> In other words, **using SAEs of the scale employed in GemmaScope or LlamaScope to fit extremely large datasets is insufficient for true decompression**. As a consequence, these methods do achieve *some* amount of decompression relative to the original representations, but **relative to the full dataset they still operate in a compressive regime**, leading to suboptimal reconstruction accuracy (Fig. 4).
>  This is also reflected in **Fig. 7**, where **features outperform neurons, yet still fall short of atoms**, with atoms corresponding to fully decompressed representations.
>
> The scaling experiments in **Fig. 8 (Section 4.4)** further demonstrate this point:
>  **there exists an optimal pairing between dataset size and SAE size**. For a given amount of data, there is a **minimum SAE capacity threshold** above which reliable reconstruction becomes possible.
>
> Thus, **Section 4.4 effectively shows that our training methodology is scalable**. However, for datasets containing hundreds of millions or billions of samples, **Fig. 8 implies that the optimal SAE size required for full decompression may become extraordinarily large**. This naturally suggests a direction for **future work: although SAEs exhibit some scalability, recognizing atoms at even larger scales may require new techniques—but such techniques will still operate within, and be justified by, the framework of Atoms Theory, which constitutes our core contribution**.
>
>
>
> [1] Toy Models of Superposition.

---

> ### Author Response · Authors · 2025-11-15
> **Rebuttal by Authors (Response to "W3&Q3")**
>
> #### **Response to "W3&Q3"**
>
> We would like to clarify that **our use of GemmaScope and LlamaScope as baselines is *not* intended to suggest that our SAE models are inherently superior**. Rather, the purpose is to highlight that **reconstruction based on features remains unreliable**, whereas **our experiments demonstrate that activations from LLMs *can* be reconstructed with high fidelity**. (**We explicitly emphasized this point when introducing the baselines in line 1562.**)
>
> Furthermore, **as shown by the extended experiments in Fig. 8, our approach remains scalable under reasonable computational budgets**, and this conclusion continues to hold. However, **for datasets containing hundreds of millions or billions of samples, the optimal scaling curve indicates that the required SAE size for full decompression may become extraordinarily large**.
>
> This naturally points to **a future research direction**: although SAEs exhibit a degree of scalability, **identifying atoms at much larger scales may require new techniques**. Nevertheless, **such methods would still operate within—and be justified by—the framework of Atoms Theory**, which is our core contribution.

---

> ### Author Response · Authors · 2025-11-15
> **Rebuttal by Authors (Response to "W4&Q4")**
>
> #### **Response to "W4&Q4"**
>
> To reiterate, **we do not use GemmaScope or LlamaScope as baselines to claim that our SAE models are superior**. Rather, the purpose is to highlight that **reconstruction based on features remains unreliable**, whereas **our experiments show that activations from LLMs *can* be reconstructed with high fidelity**. (**We explicitly emphasize this point when introducing the baselines in line 1562.**)
>
> Regarding the controlled experiments, we provide the **scaling experiments in Fig. 8**. We systematically vary both the SAE size and the dataset size, conducting **6 × 8 = 48 paired experiments** to **demonstrate the optimal matching relationship between data scale and SAE scale**. Our goal here is to **show that such atomic structure truly exists within language model activations** (the core question stated in line 037), **not to propose a model suite that is “better” than GemmaScope or LlamaScope**.
>
> Finally, we note that **we intentionally perform no preprocessing on the input activations**, as our objective is to uncover **the inherent structure present within the activations themselves**.

---

> ### Author Response · Authors · 2025-11-17
> **To Reviewer oTLx**
>
> **Dear Reviewer oTLx,**
>
> Thank you once again for your careful reading of our work and for the valuable feedback you provided. We genuinely appreciate the time and attention you have invested in assessing our work.
>
> In the above response, we address each of your comments in detail. We hope that these clarifications help resolve remaining concerns and provide a more complete understanding of the contributions of our work.
>
> Should you have any remaining questions or concerns regarding our work, please feel free to inform us. We would welcome the opportunity to offer any further clarification that may be helpful.
>
> Sincerely,
>
> The Authors

---

> ### Author Response · Authors · 2025-11-24
> **Follow-up on Rebuttal Clarifications (If Needed)**
>
> **Dear Reviewer oTLx,**
>
> We hope you are doing well, and we sincerely appreciate the time and effort you have dedicated to reviewing our work.
>
> During the rebuttal phase, we have carefully addressed every concern raised in your reviews, and we would be grateful if you could kindly let us know whether you have any remaining questions or would like further clarification on any part of our response.
>
> If there are specific aspects that you believe require additional detail (e.g., experiments, theoretical clarification, or presentation adjustments), we would be very glad to elaborate.
>
> Thank you again for contributing your expertise and for helping us improve our work.
>
> Kind regards,
>
> The Authors

---

> ### Author Response · Authors · 2025-11-28
> **Follow-up on Rebuttal Clarifications**
>
> **Dear Reviewer oTLx,**
>
> Thank you very much for taking the time to review our paper and for the helpful comments you provided earlier. We truly appreciate your efforts.
>
> As the discussion phase is approaching its end, we would like to kindly check whether there are any remaining concerns or questions that we could help clarify. We are more than happy to provide additional details or explanations if needed.
>
> If all your concerns have already been addressed in our rebuttal, we would greatly appreciate it if you could consider increasing your score accordingly.
>
> Best regards,
>
> The Authors

---

> ### Comment · Reviewer_oTLx · 2025-11-28
>
> I thank the authors for their detailed clarification, which resolved a lot of my concerns. I am willing to update my final rating accordingly. However I am still not fully convinced of the practical impact of this work, given the limited dataset size used in this work.

---

> > ### Author Response · Authors · 2025-11-28
> > **Re-clarification of Our Work (Part 1 / 2)**
> >
> > Thank you very much for your response and for giving us the opportunity to further clarify our work.
> >
> > **We would like to reiterate that the core contributions of our paper are as follows:**
> >
> > - Based on the **atomic inner product**, we **formally and rigorously define—for the first time—the fundamental representational units of LLMs**, which we call ***atoms***. Building on this definition, we derive the **theoretical properties of atoms**, how to **quantitatively validate whether target representations constitutes atoms**, and provide **theoretical guarantees for the identifiability of atoms**. Together, these results form a complete **Atoms Theory**.
> >
> > - To **empirically validate the foundation of the atomic theory**, namely the **atomic inner product**, we conduct experiments across a **wide range of language models**, including:
> >
> >   - GPT2-Small, Medium, Large, GPT-J-6B
> >   - Pythia-1B, 1.4B, 2.8B, 6.9B
> >   - Llama2-7B, 13B, Llama3-8B, Llama3.1-8B
> >   - Gemma2-2B, 9B
> >
> >   Across **all layers** of these models, we validate the atomic inner product through both the **representation shift phenomenon** (**Figure 2; Appendix B**) and the **representation shift correction experiment** (**Figure 3; Appendix B**). This provides strong evidence that the atomic inner product offers a meaningful perspective on internal representations and **establishes a solid empirical foundation for the Atoms Theory**.
> >
> > - **How can we empirically verify whether a target representation is atomic?** According to the **Atoms Theory**, the definition of atoms encompasses three components: **faithful reconstruction** (reflected by $R^2$), **sparsity** (reflected by $K$), and **approximate orthogonality** (reflected by $\mu$). Although all three quantities can be measured, sparsity and approximate orthogonality are inherently qualitative. The **Atoms Theory** shows that they can be jointly characterized by the condition $\mu < \frac{1}{2K - 1}$, which reflects the **stability** of atoms—that is, whether corresponding representations can be uniquely reconstructed. Therefore, **in practice, it suffices to measure faithful reconstruction ($R^2$, Figure 4) and stability ($\mu < \frac{1}{2K - 1}$, Figure 7) to determine whether target units are atomic**, which derived from the **Atoms Theory**.
> >
> >   To the best of our knowledge, this is **the first systematic and rigorous method for verifying whether target units matches the ideal notion of atoms**, whereas prior works mostly relied on labor-intensive and qualitative analyses that lack rigor.
> >
> > - **Empirically, we further examine whether commonly studied units—neurons and features—satisfy requirements of atoms.** Our conclusions are:
> >
> >   - For **neurons**, although they can faithfully reconstruct the original representations, they lack stability (**Figure 7**). Stable neurons account for only $0.45\%$ on average.
> >   - For **features**, we observe suboptimal reconstruction fidelity (**Figure 4**), but better stability than neurons, with stable features accounting for **68.25%** on average, although instability still exists.
> >
> >   To our knowledge, this is **the first systematic comparison between neurons and features—using rigorously derived quantitative criteria—to evaluate their suitability as fundamental interpretability units**.
> >
> > - Further analysis shows that the **fundamental reason for the suboptimal behavior of features is that existing SAEs (e.g., GemmaScope and LlamaScope) are not sufficiently large to identify atoms when trained on massive corpus**. **Section 4.4 (Lines 438–450)** and **Figure 8** demonstrate that **effective atom identification requires the SAE size to exceed a threshold relative to the dataset size**.
> >
> >   Accordingly, we **determine the appropriate size of threshold-activated SAEs (which is theoretically guaranteed to enable valid atoms identification) based on dataset scale**, and retrain SAEs across all layers of **multiple language models (Gemma2-2B, Gemma2-9B, Llama3.1-8B)**. This allows us to **successfully identify atoms in practice** (**Figures 4 and 7**; visualized in **Figures 5 and 6**).
> >
> >   To our knowledge, we are **the first to identify the fundamental reason that prior works obtain only suboptimal SAE results**:
> >
> >   - **ReLU SAEs**: because atoms are approximately orthogonal, identifying them requires thresholding to filter out noise induced by near-orthogonality; ReLU’s fixed threshold at 0 is insufficient for this.
> >   - **TopK SAEs**: these can partially mimic threshold activation but require a fixed $K$ (a hyperparameter), limiting practical applicability.
> >   - **JumpReLU SAEs**: this is the most flexible implementation of threshold activation, yet prior works still obtained suboptimal results due to mismatch between dataset scale and model capacity (**Figure 8**).
> >
> >   In contrast, **we are the first to propose selecting SAE size based on dataset scale, enabling accurate reconstruction and stability, thereby empirically validating the existence of atoms**.

---

> > ### Author Response · Authors · 2025-11-28
> > **Re-clarification of Our Work (Part 2 / 2)**
> >
> > **In summary, the contributions of this work are broad and multifaceted:**
> >
> > 1. We are the first to theoretically formalize the fundamental representational units of LLMs and to establish a complete Atoms Theory.
> > 2. We are the first to reveal the widespread phenomenon of representation shifting in LLMs and demonstrate how to correct it, thereby validating the foundation of the Atoms Theory—namely, the atomic inner product.
> > 3. We are the first to propose, in a systematic and rigorous manner, how to empirically verify whether target units are ideal fundamental representational units—atoms—and to systematically compare neurons and features as the fundamental units, highlighting their differences and limitations.
> > 4. We are the first to identify the root cause behind the suboptimal performance of SAEs in prior work and to propose selecting the appropriate size of threshold-activated SAEs based on dataset scale. This enables accurate reconstruction that satisfies stability requirements, providing practical validation for the existence of atoms.
> >
> > **We would like to clarify that the value of interpretability research is not determined solely by its immediate practical applications. Enhancing conceptual clarity in this field, providing a stronger theoretical foundation, and conducting targeted empirical validations are equally important. Our work aims to advance the theoretical understanding of mechanistic interpretability, and we believe such progress can meaningfully support future practical developments.**
> >
> > We sincerely appreciate your suggestion. While **Figure 8 already demonstrates the scalability of our approach through extensive scaling experiments**, we agree that exploring even larger-scale datasets would be valuable. We plan to pursue this direction in future work, though it is beyond the scope of the present paper.

---

### Official Review · Reviewer_WuK6 · 2025-11-03

**Soundness:** 4
**Presentation:** 3
**Contribution:** 3
**Rating:** 6
**Confidence:** 2

**Summary:**

The paper proposes the Atoms Theory and studies the fundamental units of internal representations of LLMs. Previously, neurons or features were regarded as such units, but the paper argues that they are insufficient since (1) neurons often exhibit substantial polysemy, growing doubt on their validity for analysis and (2) features face concerns of unreliable reconstruction and instability. This paper introduces methodologies to reliably identify indivisible, fundamental units, "atoms", of LLMs, to provide a rigorous theoretical foundation for mechanistic interpretability.

**Strengths:**

- The paper studies a well-known and important problem in the literature: the polysemy of neurons and the reconstruction unreliability of standard SAE features.

- The claims are theoretically supported with rigorous analysis.

- Empirically, the proposal framework is better than baselines on Gemma 2B, Gemma 9B, and Llama3 8B.

**Weaknesses:**

- What is the computational cost of obtaining the empirical results in the paper? Training specialized SAEs for every layer of LLMs would be too costly. Are there any computational tricks the authors are using? I suggest discussing the computational cost in more detail in the paper.

- The theoretical guarantees rely on the assumption that LLM representations are inherently extremely sparse. Do the authors have an understanding of how much of these results hold if the model employs dense representations, e.g., for complex tasks?

**Questions:**

See Weaknesses.

---

> ### Author Response · Authors · 2025-11-15
> **Rebuttal by Authors (Response to "W1")**
>
> We would like to sincerely thank you for your thoughtful and thorough evaluations of our submission.
>
> #### **Response to "W1"**
>
> We acknowledge that prior works \[1\]\[2\]\[3\]\[4\] have highlighted the high computational cost of training a separate SAE for every layer of an LLM on large-scale corpora. **Importantly, this cost arises from two independent factors**: (i) **the size of the SAE model**, and (ii) **the size of the training dataset**.
>
> However, **influenced by the success of LLM training—where “more data is always better” is widely accepted—previous work almost uncritically transferred this intuition to SAE training**, while **overlooking a crucial conceptual distinction**: **LLMs perform compression, whereas SAEs perform decompression** \[5\].
>
> More concretely, for LLMs, scaling up the dataset for a fixed-size model only means the model is asked to compress a larger variety of dataset—a task that typically does not degrade performance and even improve performance in practice. **In contrast, for SAEs trained with a fixed model capacity, increasing the dataset size beyond what the model can faithfully decompress causes the SAE to revert to learning a compressed (rather than decompressed) representation.** **This phenomenon is clearly demonstrated in Fig. 8**.
>
> **We fully acknowledge the contributions of prior works** \[1\]\[2\]\[3\]\[4\]. Nonetheless, **prior works inherit the above issue**, which leads to two major consequences:
>
> - **Suboptimal Reconstruction Quality**. Reconstruction fidelity determines whether the decomposition faithfully reflects the underlying model’s representational structure. As shown in **Fig. 4**, previous methods often produce relatively lower $R^2$ values. **In contrast, across all layers and all tested models, our training method consistently achieves an average reconstruction of 99.90% $R^2$, achieving near-perfect reconstruction in a statistical sense**.
>
> - **Suboptimal Reconstruction Stability**. Stability refers to whether the decomposed features (atoms) can uniquely reconstruct the original representation. From the perspective of representational separability, stability corresponds to the degree of atomic unambiguity (see Fig. 6 and Table 1). As shown in **Fig. 7**, both features and neurons yield low stability scores. The corresponding **metric $q$—derived from our Atoms Theory—quantifies the proportion of stable atoms (with $q=1$ indicating all atoms are stable)**. Prior work obtains approximately $q \approx 68.25\%$ for features and $q \approx 0.45\%$ for neurons. **In contrast, our training method achieve an average of $q \approx 99.86\%$, statistically approaching stability.**
>
> **Our training method is built upon a simple but essential principle**: **For a given dataset size, the SAE must be chosen large enough to decompress the data.** Fig. 8 provides a practical reference for selecting an appropriate model size. Although this principle appears straightforward, **it has been almost entirely overlooked in prior work**. The prevailing default training method has been:
>
> > “Choose the largest SAE that fits the available compute, then train on as much data as possible, because more data is always better.”
>
> **Our empirical results (Figs. 4, 7, 8) show that this assumption can be fundamentally incorrect.**
>
> With the above context, we emphasize that when discussing the **“computational cost of training SAEs,”** we specifically refer to:
>
> > **The total computation required to train an SAE of appropriate capacity on a dataset size that the SAE can faithfully decompress—i.e., a setting where the SAE is known to be functioning correctly.**
>
> ------
>
> **In practice, we first identify the largest dataset size that can be reliably decompressed by an SAE fitting on a single GPU (3090-24G or A100-80G), and we conduct all experiments at this maximal feasible scale.**
>
> Therefore, the specific computational cost is as follows:
>
> - **Gemma2-2B (per layer):** **~24 GPU-hours (on 3090-24G) on average**;
> - **Gemma2-9B (per layer):** **~56 GPU-hours (on A100-80G) on average**;
> - **Llama3.1-8B (per layer):** **~58 GPU-hours (on A100-80G) on average**;
>
> In the extended experiments in §4.4, the largest SAE (top right corner) required:
>
> - **~135 GPU-hours (on A100-80G)**.
>
> We will include this information in the paper for greater transparency.
>
>
>
> [1] Gemma Scope: Open Sparse Autoencoders Everywhere All At Once on Gemma 2.
>
> [2] Llama Scope: Extracting Millions of Features from Llama-3.1-8B with Sparse Autoencoders.
>
> [3] Scaling and Evaluating Sparse Autoencoders.
>
> [4] Scaling Monosemanticity: Extracting Interpretable Features from Claude 3 Sonnet.
>
> [5] Toy Models of Superposition.

---

> ### Author Response · Authors · 2025-11-15
> **Rebuttal by Authors (Response to "W2")**
>
> #### **Response to "W2"**
>
> In practice, **“sparse” and “dense” are fundamentally qualitative notions**. In **Assumption 4**, we **quantify sparsity using $K:\ \|\boldsymbol{\delta}_i\|_0 \le K \ll |D|$, where small values of $K$ correspond to sparse representations and large values correspond to dense ones**. **Thus, it is clear that our assumptions cover a very broad range of settings.**
>
> **What truly matters, however, is understanding how sparsity or density affects the system—specifically, how changes in $K$ influence the system’s behavior**. In **Theorem 9**, we establish **the relationship between the maximum normalized atomic inner product $\mu := \max_{i \ne j} |\langle \tilde{\boldsymbol{d}}_i, \tilde{\boldsymbol{d}}_j\rangle|$ and sparsity $K$ with $\mu < \frac{1}{2K - 1}$**. When sparsity $K$ increases, the right-hand side decreases, thereby shrinking the allowable range of $\mu$. This means atoms become closer to being orthogonal, and by the JL lemma \[1\], fewer atoms can be embedded in the space. Conversely, when sparsity decreases, the opposite holds. Thus, one can intuitively see that the denser the representation becomes, the fewer atoms the space can accommodate, and consequently, the fewer independently varying semantic factors the representation can express. In the extreme dense limit, $\mu$ approaches 0; the atoms must be strictly orthogonal. In an $H$-dimensional space, this restricts the number of admissible atoms to exactly $H$, which is clearly insufficient to model the complexity of real-world semantics.
>
> **In other words, sparsity $K$ determines how many identifiable atoms a high-dimensional space can support under approximate-orthogonality constraints.**
>
> **It is also important to note that real-world information is almost inherently sparse**. For example, when we refer to *“a yellow banana”*, the semantic content is highly specific and therefore sparse relative to the vast space of all possible real-world descriptions. **A truly dense world would be nearly impossible to conceptualize—it would require an almost infinitely large representational space**. This intuition can be formalized: Given $\mu < \frac{1}{2K-1}$, as $K \to \infty$, we obtain $\mu \to 0$, implying strict orthogonality among all atoms. To represent $M$ unrelated concepts (assuming no shared structure), we would need an $M$-dimensional space—an infeasible requirement in practice.
>
> **Thus, the sparsity of real-world information introduces compressible structure, enabling us to encode a vast amount of world knowledge within a finite-dimensional space using approximately orthogonal atoms**. In other words, **dense representations require strictly orthogonal atoms to remain identifiable, whereas sparse representations allow atoms to be only approximately orthogonal while remaining identifiable.**
>
> Accordingly, the statement in the question is not entirely accurate. **Even in complex tasks, the representations used by language models remain sparse**. **This is a property of the real world (as reflected in the training corpora) and therefore also a property of language model activations**.
>
> **To further validate this claim, we conduct a controlled experiment** comparing three types of representations extracted under the same experimental settings:
>
> - **General corpus** (sampled from **Wikipedia**);
> - **Complex reasoning** (sampled from **MATH500**);
> - **Knowledge atomization** (sampled from **WikiData**).
>
> We train SAEs on these three categories under identical configurations. The results are as follows:
>
> |                                      | $R^2$  | $L_0$ |
> | ------------------------------------ | ------ | ----- |
> | **General Corpus (Wikipedia)**       | 99.84% | 9.36  |
> | **Complex Reasoning (MATH500)**      | 99.86% | 9.56  |
> | **Knowledge Atomization (WikiData)** | 99.89% | 8.32  |
>
> These results demonstrate that **the sparsity of representations is largely independent of the specific task or activation-sampling process**. In other words, **even complex reasoning tasks yield inherently sparse representations**.
>
>
>
> [1] https://en.wikipedia.org/wiki/Johnson-Lindenstrauss_lemma

---

> ### Author Response · Authors · 2025-11-17
> **To Reviewer WuK6**
>
> **Dear Reviewer WuK6,**
>
> We would like to express our sincere appreciation again for the time and expertise you have devoted to evaluating our submission.
>
> In the above response, we address each of the questions raised in the reviews, provide additional theoretical explanations, and include new experimental results where appropriate. We hope that our replies satisfactorily resolve your concerns and offer a clearer understanding of our contributions.
>
> If any questions remain or further clarification would be helpful, we would be very grateful for the opportunity to elaborate.
>
> Sincerely,
>
> The Authors

---

> ### Author Response · Authors · 2025-11-24
> **Follow-up on Rebuttal Clarifications (If Needed)**
>
> **Dear Reviewer WuK6,**
>
> We hope you are doing well, and we sincerely appreciate the time and effort you have dedicated to reviewing our work.
>
> During the rebuttal phase, we have carefully addressed every concern raised in your reviews, and we would be grateful if you could kindly let us know whether you have any remaining questions or would like further clarification on any part of our response.
>
> If there are specific aspects that you believe require additional detail (e.g., experiments, theoretical clarification, or presentation adjustments), we would be very glad to elaborate.
>
> Thank you again for contributing your expertise and for helping us improve our work.
>
> Kind regards,
>
> The Authors

---

> > ### Comment · Reviewer_WuK6 · 2025-11-25
> >
> > Thank you for the response, which addresses my questions sufficiently. After reading other reviewers' comments, I plan to keep my original score.

---

> > > ### Author Response · Authors · 2025-11-26
> > > **Appreciation for Your Follow-up and Review**
> > >
> > > Thank you very much for your follow-up and for taking the time to carefully review our response. We sincerely appreciate your engagement with our work and the constructive feedback you provided throughout the evaluation process. Your comments have helped us improve the clarity and presentation of our paper, and we are grateful for the opportunity to address your concerns. Thank you again for your time and consideration.

---

### Author Response · Authors · 2025-12-01
**Summary of Contributions, Reviews, and Discussions (Part 1/3)**

**Dear PCs, SACs, ACs, and Reviewers,**

We would like to express our sincere gratitude to the PCs, SACs, ACs, and Reviewers WuK6, oTLx, and x57o for their valuable time, constructive feedback, and thoughtful discussions during the review process. To help reduce the workload for the newly assigned ACs, we provide below a concise summary of our paper’s main contributions, along with a synthesis of the reviewers’ comments and the key points raised during the discussion.

------

We greatly appreciate the reviewers’ feedback of our work. Below, we restate the main contributions of our work, along with explanations and clarifications based on all the reviewers’ comments, specifically:

- **This paper addresses a central question in interpretability research—how to define the fundamental representational units of LLMs—by developing the *Atoms Theory*. This theory formally and systematically defines (Definition 1, Theorem 2, Corollary 3, Assumption 4, Definitions 5–6), evaluates (Definition 7, Theorems 8–9), and identifies (Theorems 9–10) such units, referred to as *atoms*.**
  - All three reviewers acknowledge the importance of this problem and highly commend the systematic and rigorous formulation of our *Atoms Theory* (WuK6: Strengths 1, 2; oTLx: Strengths 1, 2, 3; x57o: Strengths 1, 2).
  - Regarding **(WuK6: Weakness 2) concerning about sparsity in Assumption 4**, we clarified the misunderstanding, highlighted the generality of our assumption, and provided additional controlled experiments to illustrate this point. Reviewer WuK6 explicitly confirmed that our response fully addressed the concern.
  - Regarding **(x57o: Question 1) about whether random vectors and ReLU SAE features satisfy the definition of atoms**, we provided a detailed explanation and accompanying experiments. Reviewer x57o agreed that our answer resolved the issue.
- **We thoroughly validate the foundation of the *Atoms Theory*—specifically, *atomic inner products*—across all layers of 14 language models, including the GPT-2, GPT-J, Pythia, Llama, and Gemma families. By demonstrating the widespread presence of *representation shifting* in LLMs and showing how atomic inner products *correct representation shifting*, we empirically establish the robustness of the *Atoms Theory*.**
  - Regarding **(x57o: Weakness 1) concerning the experiments on correcting representation shifting**, we clarified the misunderstanding and pointed to the precise experimental setup described in Appendix B. Reviewer x57o agreed that our response addressed this concern.
- **Building on *Atoms Theory* (specifically Theorem 9), we systematically and quantitatively examine the limitations of neurons and features as fundamental representational units. Our experiments show that the polysemanticity of neurons arises from their substantial representational instability (Fig. 7). While features exhibit improved stability (and thus stronger monosemanticity), instability persists (Fig. 7), and features remain unfaithful (Fig. 4).**
  - All three reviewers recognized the value of formally defining and rigorously evaluating fundamental representational units in LLMs (WuK6: Strength 1; oTLx: Strength 3; x57o: Strength 2).

---

> ### Author Response · Authors · 2025-12-01
> **Summary of Contributions, Reviews, and Discussions (Part 2/3)**
>
> - **Building on *Atoms Theory* (specifically Theorem 10), this paper rigorously establishes the identifiability of atoms using threshold-activated SAEs and validates this theorem empirically. Specifically, we first demonstrate through experiments that the unfaithfulness and instability observed in prior SAE-based studies primarily stem from a mismatch between model scale and data scale (Fig. 8). Leveraging this insight, we select model scale that match the data scale and train SAEs across all layers of Gemma2-2B, Gemma2-9B, and Llama3.1-8B. This allows us, for the first time, to achieve high-fidelity reconstruction within the SAEs framework (Fig. 4), specifically threshold-activated SAEs, along with high stability (Fig. 7), and to recover atoms that align with theoretical expectations (Figs. 5, 6, 38-43).**
>   - Reviewer WuK6 provided positive feedback on our high-fidelity reconstruction and high stability across multiple language models and layers (WuK6: Strength 3).
>   - Regarding **(WuK6: Weakness 1)  the suggestion to supplement computational cost details**, we supplemented and discussed the computational cost in detail. Reviewer WuK6 explicitly acknowledged that our response fully addressed the concern.
>   - Regarding **(oTLx: Weakness&Question 1; x57o: Weakness 2) concerning about the terminology “threshold activation SAEs”**, we clarified that threshold activation is a broader conceptual class that encompasses multiple concrete implementations; JumpReLU and even TopK can be viewed as specific instances of threshold activations (line 298-304). Importantly, threshold activation is motivated by enabling effective identification of atoms under approximate orthogonality, whereas JumpReLU was introduced to mitigate feature shrinkage during SAE training, and TopK was proposed to directly control sparsity. Both reviewers oTLx and x57o agreed that our explanation resolved this concern.
>   - Regarding **(x57o: Question 2) concerning about different activation collecting methods**, we provided detailed clarification of our implementation and additionally introduced several alternative activation collecting methods—including natural language corpus, complex mathematical reasoning, and knowledge atomization. The corresponding controlled experiments showed consistent trends. Reviewer x57o agreed that our response addressed this concern.
>   - Regarding **(oTLx: Weaknesses&Questions 2–4) concerning about data scale**, we explained that the chosen data scale is grounded in our empirical finding (Fig. 8) that effective atom identification requires an appropriate match between model scale and data scale. Based on this insight, we conducted the largest feasible training runs within our current available computational resources (i.e., fully trainable on a single RTX 3090-24G or A100-80G), which successfully enabled effective atom identification (Figs. 4 and 7). It is worth noting that the dataset of this size (tens of thousands, up to 73K) is fully adequate for the goal of atom identifications. The extensive and diverse results obtained from large-scale training across all layers of multiple LLMs are already sufficient to support our conclusions. Furthermore, for larger datasets, our scaling experiments (Sec. 4.4) demonstrate the scalability of our method and establish its practical feasibility for large-scale atom identification—underscoring the real-world applicability of our work.
>
> --------------------------

---

> ### Author Response · Authors · 2025-12-01
> **Summary of Contributions, Reviews, and Discussions (Part 3/3)**
>
> --------------------------
>
> **Regarding the reviewers’ responses to our further clarifications**, all three reviewers explicitly stated that our rebuttal addressed the majority of their concerns. Specifically, **Reviewer WuK6 confirmed that our responses sufficiently addressed the questions**, while **Reviewers oTLx and x57o acknowledged that most of their concerns had been adequately addressed**. The **only potential concern**, as noted by Reviewers oTLx and x57o, pertains to **the practical applicability of the threshold-activated SAEs trained in our work**.
>
> We would like to offer a unified clarification here (although this has been explained thoroughly in both the rebuttal and the paper): **the goal of this work is not to deliver a suite comparable to GemmaScope or LlamaScope, but to rigorously define the fundamental representational units of LLMs—atoms—and to systematically develop and empirically validate the *Atoms Theory*.** Within this scope, we have made every effort to conduct the largest feasible training runs under our current available computational resources  (i.e., fully trainable on a single RTX 3090-24G or A100-80G GPU). **Importantly, our training scale is fully sufficient for the goal of atom identifications, and the extensive and diverse results obtained from large-scale training across all layers of multiple LLMs are already sufficient to support our conclusions.** Furthermore, **our scaling experiments (Sec. 4.4) demonstrate clear scalability to larger models and datasets, providing strong evidence for the feasibility of substantially greater scales, which further underscores the practical relevance of our work.**
>
> We emphasize that **providing a suite like GemmaScope or LlamaScope is never among this paper’s stated contributions**, and **it would be unjustified and inappropriate to use this as a basis to discount the genuine core contributions of our work**.
>
> **Furthermore, both reviewers oTLx and x57o expressed willingness to raise their scores.**
>
> ---------------------
>
> We have faithfully summarized the contributions of our work, the reviewers’ comments, and our corresponding responses above, and we hope this will assist the ACs in their assessment. We sincerely thank the reviewers, ACs, SACs, and PCs for their hard work and outstanding efforts. Their insightful feedback has further improved the quality of our paper. The authors extend their deepest respect and gratitude to everyone involved in the review process.
>
> Sincerely，
>
> The Authors

---

### Meta-Review · Area_Chair_afxg · 2026-01-14

**Summary:**

The paper proposes the Atoms Theory and studies the fundamental units of internal representations of LLMs. Previously, neurons or features were regarded as such units, but the paper argues that they are insufficient since (1) neurons often exhibit substantial polysemy, growing doubt on their validity for analysis and (2) features face concerns of unreliable reconstruction and instability. This paper introduces methodologies to reliably identify indivisible, fundamental units, "atoms", of LLMs, to provide a rigorous theoretical foundation for mechanistic interpretability.

The original concern include:
* What is the computational cost of obtaining the empirical results in the paper? Training specialized SAEs for every layer of LLMs would be too costly. Are there any computational tricks the authors are using? I suggest discussing the computational cost in more detail in the paper.
* The theoretical guarantees rely on the assumption that LLM representations are inherently extremely sparse. Do the authors have an understanding of how much of these results hold if the model employs dense representations, e.g., for complex tasks?
* The threshold-activated SAE is functionally equivalent to JumpReLU SAE, offering no methodological novelty.
* The reconstruction experiments are conducted on a dataset orders of magnitude smaller than those used in GemmaScope and LlamaScope as baseline (tens of thousands vs. hundreds of millions).
* The comparison of reconstruction scores is therefore unfair and misleading, as performance differences can be entirely attributed to dataset scale rather than model quality.
* The claimed empirical validation of “Atoms Theory” is thus not credible given the limited and mismatched setup. The work overstates its practical impact despite weak experimental support.
* The idea of SAEs with threshold activations (i.e. JumpReLU SAEs) is not novel and proposed by Rajamanoharan et al. before. The results of high sparsity and near perfect reconstructions contradict the previous studies.
* Not fully convinced of the practical impact of this work, given the limited dataset size used in this work.
* Suspect the correct matching of data scale and model capacity, since it's not so meaningful to train an SAE that could not generalize to common linguistic tasks.

**Reviewer Concerns:**

The majority of the concerns were directly addressed and even acknowledged by some of the original reviewers. However the final few remain, including especially the assumptions underlying the fundamental setup and the potential for practical impact of the work.

**Reviewer Scores:**

I do not think the scores would change significantly.

---

### Decision · Program_Chairs · 2026-01-26

Reject